# Quantitative Bounds for Length Generalization in Transformers

**Zachary Izzo** [*]
NEC Labs America
zle.izzo@gmail.com

**Eshaan Nichani** [*]
Princeton University
eshnich@princeton.edu

**Jason D. Lee**
Princeton University
jasondlee88@gmail.com

## Abstract

We study the problem of length generalization (LG) in transformers: the ability of a model trained on shorter sequences to maintain performance when evaluated on much longer, previously unseen inputs. Prior work by Huang et al. (2025) established that transformers eventually achieve length generalization once the training sequence length exceeds some finite threshold, but left open the question of how large it must be. In this work, we provide the first quantitative bounds on the required training length for length generalization to occur. Motivated by previous empirical and theoretical work, we analyze LG in several distinct problem settings: $\ell_\infty$ error control vs. average error control over an input distribution, infinite-precision softmax attention vs. finite-precision attention (which reduces to an argmax) in the transformer, as well as for one- or two-layer transformers. In all scenarios, we prove that LG occurs when the internal behavior of the transformer on longer sequences can be "simulated" by its behavior on shorter sequences seen during training. Our bounds give qualitative estimates for the required length of training data required for a transformer to generalize, and we verify these insights empirically. These results sharpen our theoretical understanding of the mechanisms underlying extrapolation in transformers, and formalize the intuition that richer training data is required for generalization on more complex tasks.

## 1 Introduction

An important problem in the training of large language models (LLMs) is length generalization (LG), which is the ability of a model to generalize to input sequences longer than those encountered during training. Prior works have studied the ability of transformers to length generalize on simple testbed tasks (Anil et al., 2022; Kazemnejad et al., 2023), yet the success of LG varies widely from task to task. Recent theoretical work has thus sought to characterize which tasks admit LG. In particular, Zhou et al. (2023) introduced the RASP-L conjecture, which states that transformers can length generalize on tasks which are expressible by a "simple" RASP-L program (a variant of the RASP language introduced in Weiss et al. (2021)). Huang et al. (2025) later formalized and partially proved this conjecture, showing that tasks expressible by a limiting object called a "limit transformer," which includes tasks expressible by a C-RASP program (Yang & Chiang, 2024), admit LG at some finite training length. These results, however, are asymptotic in nature and rely on "identification in the limit" (Gold, 1967; Angluin, 1980) style arguments, where the inference procedure can eventually rule out all hypotheses except for the ground truth. In particular, for a fixed task $f$ on which LG is possible, it is not specified what the minimum training length is for LG to occur.

Our goal in this paper is to characterize how long training sequences need to be in order for a transformer to generalize to sequences of arbitrary length. Specifically, we adopt the limit transformer formulation from Huang et al. (2025), and aim to provide quantitative bounds on the minimum $N$ such that two limit transformers $f, g$ which agree on inputs of length $\leq N$ approximately agree on inputs of arbitrary length.

We study this question in two distinct regimes. In Section 4, we consider limit transformers operating at finite-precision, which matches the setting of Huang et al. (2025). This results in a hard attention

---

[*]Equal contribution, order determined by coin flip.

pattern for sequences of a certain length. Our main results are that for one-layer limit transformers, for both worst-case error control (Theorem 4.1) and average error control over a distribution (Theorem 4.2) the minimum such $N$ scales monotonically with the parameter norms of the transformer, the positional embedding periodicity $\Delta$, "locality" parameter $\tau$, token vocabulary size $|\Sigma|$, and inverse error $\varepsilon^{-1}$. In Section 5, we additionally study the setting where the parameters and forward pass are computed at infinite precision. This allows us to establish results independent of the model precision, and is a more suitable model for multi-layer transformers where the inputs to later layers "mix" the first-layer inputs, and hence can be treated as continuous. In Theorem 5.2, we establish a quantitative LG bound for two-layer transformers, which scales with the transformer weight norms.

The proofs of our main results in both the finite- and infinite-precision settings rely on the following high-level "simulation argument." Given two limit transformers $f, g$ and a long input string $x$, we construct a string $z$ of length at most $N$ such that $f(x) \approx f(z)$ and $g(x) \approx g(z)$; if $f$ and $g$ agree on all inputs of length $\leq N$, then they must satisfy $f(x) \approx g(x)$. The key step in this simulation argument is to construct $z$ which approximately preserves various sufficient statistics which are necessary for computing the forward pass of the model. The proof of Theorem 4.1 does this explicitly, by ensuring that $z$ approximates the empirical frequencies of each token in the hard attention pattern, while the proof of Theorem 5.2 does this randomly, sampling $z$ from a specially defined distribution and invoking the probabilistic method. Nevertheless, the unifying principle in both settings is that LG is possible whenever the internal behavior of a transformer on a larger sequence can be simulated by its behavior on a shorter sequence.

Altogether, our results make progress towards both characterizing a natural hierarchy of "difficulty" amongst length-generalizable tasks, and more practically speaking, developing a better understanding of how to scale training context length for LLMs.

## 2 RELATED WORK

A number of works have empirically studied the ability of transformers to length generalize on various tasks. Bhattamishra et al. (2020) studies the ability of transformers to length generalize on various formal language tasks. Anil et al. (2022) show that transformers fail to generalize on certain reasoning tasks, unless certain scratchpad prompting techniques are used. Kazemnejad et al. (2023) study the role of various positional encoding schemes on LG. Zhou et al. (2023) study LG on various algorithmic tasks, and observe that tasks with a short RASP program (Weiss et al., 2021) have better LG, leading to their RASP-L conjecture. This is supported by works such as Jelassi et al. (2024), who observe that for the string copying task, transformers can length generalize when there are no repeated tokens, but fail once the string has repeats. LG has also been studied outside the context of transformers. For instance, Nerem et al. (2025) showed that trained graph neural networks can learn the Bellman-Ford algorithm which generalizes to shortest paths of arbitrary length. Buitrago & Gu (2025) studied LG in the context of recurrent models such as state-space models or linear attention.

In light of these LG challenges, recent works have designed specific positional encoding schemes, such as Alibi (Press et al., 2021) or Abacus (McLeish et al., 2024) to improve LG. Other works have also considered modifying the input with a scratchpad, extra positional information, or alternative training techniques to improve LG on arithmetic tasks (Lee et al., 2023; Shen et al., 2023; Cho et al., 2025; Lee et al., 2025; Cai et al., 2025). Most recently, architectural modifications such as looping (Fan et al., 2024) or recurrence (McLeish et al., 2024) have led to LG improvements. Other approaches by Li et al. (2025); Anson et al. (2025); Hashemi et al. (2025) have considered making modifications to the attention mechanism to improve LG.

On the theoretical front, Huang et al. (2025) partially resolves the RASP-L conjecture for tasks expressible by limit transformers. Yang et al. (2025) shows the equivalence of a class of transformers to the C-RASP programming language and provide empirical evidence that their theory predicts the depth of a transformer which is required for LG to occur in practice. Wang et al. (2024) proves that 1-layer transformers trained with gradient descent length generalize on a sparse token selection task. Ahuja & Mansouri (2024) show that a model resembling a self-attention head can length generalize. Golowich et al. (2025) show that an abstraction of the self-attention head can length generalize on tasks which depend on a sparse subset of input tokens. Veitsman et al. (2025) studied transformer LG related to copy and retrieval operations, and find that theoretical limitations do indeed transfer to practice. The work of Chen et al. (2025) is at first glance the most similar to ours, as the authors

give nonasymptotic bounds for LG. However, they focus on general models of computation with variable-length input rather than on transformers, offering complementary insights.

## 3 PROBLEM FORMULATION

### 3.1 LIMIT TRANSFORMERS

We are interested in the ability of transformers to generalize to sequences of arbitrary length, but real transformer architectures are limited by a bounded context length. To address this issue, Huang et al. (2025) introduced the concept of a *limit transformer*. These objects have an infinite context length and generalized positional embeddings, allowing them to distinguish between arbitrarily many positions in their context. The computation of a limit transformer proceeds as follows:

$$\boldsymbol{y}_i^{(0)} = \boldsymbol{E}_{x_i} + \boldsymbol{p}_i, \quad i = 1, \ldots, |x|,$$

$$a_{i,j}^{(l,h)} = (\boldsymbol{y}_j^{(l-1)})^\top \boldsymbol{K}_{l,h}^\top \boldsymbol{Q}_{l,h} \boldsymbol{y}_i^{(l-1)} + \phi_{l,h}(j, i),$$

$$\boldsymbol{Y}_i^{(l)} = \boldsymbol{y}_i^{(l-1)} + \sum_{h=1}^H \frac{\sum_{j=1}^i \exp\left(\log|x| \cdot a_{i,j}^{(l,h)}\right) \boldsymbol{V}_{l,h} \boldsymbol{y}_j^{(l-1)}}{\sum_{j=1}^i \exp\left(\log|x| \cdot a_{i,j}^{(l,h)}\right)},$$

$$\boldsymbol{y}_i^{(l)} = \boldsymbol{Y}_i^{(l)} + \boldsymbol{B}_l \cdot \psi_l(\boldsymbol{A}_l \boldsymbol{Y}_i^{(l)} + \boldsymbol{b}_l),$$

$$T(x)_i = \boldsymbol{U} \boldsymbol{y}_i^{(L)}.$$

Here $x$ is the input sequence with token $x_i \in \Sigma$ in the $i$-th position, $\boldsymbol{E}_{x_i} \in \mathbb{R}^d$ is the embedding of the $i$-th token, $\boldsymbol{p}_i$ is the $i$-th (absolute) positional embedding vector. The super- and sub-scripts $(l, h)$ denote the $l$-th layer of the transformer and the $h$-th attention head. $a_{i,j}^{(l,h)}$ is the $(l, h)$ attention logit between token $i$ and $j$, $\boldsymbol{K}_{l,h}$, $\boldsymbol{Q}_{l,h}$, and $\boldsymbol{V}_{l,h}$ are the the $(l, h)$ key, query, and value embedding matrices, respectively. The functions $\phi_{l,h}(j, i)$ do not allow for modifications to the attention pattern which cannot be captured by positional embedding vectors alone. $\boldsymbol{Y}_i^{(l)}$ denote the pre-activation features for layer $l$ at position $i$, and $\boldsymbol{y}_i^{(l)}$ denote the post-activation features which have been passed through a single-hidden-layer MLP with 1-Lipschitz activation $\psi_l$, plus a residual connection; $\boldsymbol{A}_l$ and $\boldsymbol{b}_l$ denote the hidden layer weights and bias term for this MLP, and $\boldsymbol{B}_l$ denotes the output layer weights. Finally, $T(x)_i$ denotes the output logits at position $i$ which are computed via the unembedding matrix $\boldsymbol{U}$.

Without additional constraints, a limit transformer cannot be recovered without seeing arbitrarily long input sequences. Thus, Huang et al. (2025) also make two additional assumptions. First, the limit transformers in question are assumed to be $\Delta$-*periodic*, defined as $\boldsymbol{p}_i = \boldsymbol{p}_{i+\Delta}$ for all $i$. Second, the limit transformers are also *translation-invariant*, defined as $\phi_{l,h}(j, i) = \phi_{l,h}(j + t, i + t)$ for all $t$, and $\tau$-*local*, defined as $\phi_{l,h}(j, i) = 0$ whenever $i > j + \tau$.

### 3.2 FINITE-PRECISION ATTENTION

Huang et al. (2025) assume that all of the transformer parameters, as well as the softmax attention, are computed at $p$ finite bits of precision. This is motivated by Merrill & Sabharwal (2023), and indeed, finite precision is a real constraint when LLMs are implemented in practice.

For our analysis, the precise instantiation of this assumption is that we will assume that all quantities of absolute value $\leq 2^{-p}$ are rounded to 0 during each intermediate computation of the limit transformer. Even this definition requires further clarification, particularly for the computation of the softmax. This is because the softmax (at infinite precision) is invariant to a constant shift in all of the logits; thus, in principal, the softmax may be computed as a collection of terms each of which has absolute value less than $2^{-p}$, in which case it is unclear what to do. To avoid this problem, we take the usual step for improving the numerical stability of softmax and perform computations with the largest logit shifted to 0. Equivalently, we subtract the largest logit from every logit in the softmax. After this standardization, all terms in the softmax (post exponentiation) with absolute value at most $2^{-p}$ are rounded to 0, then the computation proceeds as usual.

The impact of this assumption is as follows. Let $f$ be a single-layer limit transformer which is $\tau$-local, $\Delta$-periodic, and translation invariant as defined above. We can define the attention matrix $A \in \mathbb{R}^{\Delta|\Sigma| \times \Delta|\Sigma|}$ indexed by pairs $(y, i)$ for $y \in \Sigma$ and $i \in \mathbb{Z}/\Delta$, where $A_{(y,i),(z,j)} :=$ $(\boldsymbol{E}_z + \boldsymbol{p}_i)^\top K^\top Q(\boldsymbol{E}_y + \boldsymbol{p}_j)$. For $y \in \Sigma$ and $i \in \mathbb{Z}/\Delta$, define

$$\mathcal{A}_{(y,i)} := \{A_{(y,i),(z,i-k)} + \phi(1, k+1) \mid z \in \Sigma, \, k = 0, \dots, \tau\}.$$

Note that $\mathcal{A}_y$ contains all of the possible attention logits that we can observe when processing a token $x_i = y$. We then define the *logit margin* $\gamma(f)$ of $f$ by

$$\gamma(f) := \min_{\substack{y \in \Sigma \\ i \in \mathbb{Z}/\Delta}} \min_{\substack{a, a' \in \mathcal{A}_{(y,i)} \\ a - a' > 0}} a - a',$$

where the minimum over an empty set is defined as $+\infty$. The quantity $\gamma(f)$ is the smallest nonzero gap we can observe between a maximal attention logit and any non-maximal logit.

Now let $x$ be any input sequence and suppose that $N = |x| \geq 2^{p/\gamma(f)}$. Consider an individual term in the softmax, post-exponentiation but before the rounding procedure. These have the form

$$s_j = \exp\Big(\log N \cdot [(A_{(x_N,N),(x_j,j)} + \phi(j, N)) - (A_{(x_N,N),(x_{j^*},j^*)} + \phi(j^*, N))]\Big)$$

$$= \exp\Big(\log N \cdot [(A_{(x_N,N),(x_j,j)} + \phi(1, N-j+1)) - (A_{(x_N,N),(x_{j^*},j^*)} + \phi(1, N-j^*+1))]\Big)$$

$$= \exp(\log N \cdot (a - a^*)),$$

where $j^* \in \operatorname{argmax}_{j'=1,\dots,i} A_{(x_N,N),(x_{j'},j')} + \phi(j', N)$ is an index with the largest attention logit and $a, a^* \in \mathcal{A}_{(x_N,N)}$ are simply a renaming of the logits to emphasize that these are quantities in $\mathcal{A}_{(x_N,N)}$. The second equation follows by the translation invariance of $\phi$.

There are now two cases. If $a = a^*$ (i.e., the $j$-th position attains maximal attention for the input sequence), then $s_j = \exp(0) = 1$ and this contribution to the softmax will not be affected by the rounding procedure. On the other hand, if $a \neq a^*$ (i.e., the $j$-th position attains strictly sub-maximal attention for the input sequence), then by definition of $\gamma(f)$, $a - a^* \leq -\gamma(f)$ and we have

$$s_j = \exp(\log N \cdot (a - a^*)) \leq \exp\left(-\frac{p \log 2}{\gamma(f)} \gamma(f)\right) = 2^{-p}.$$

Thus, this term will be rounded to 0. It follows that for sequences $x$ of length $N \geq 2^{p/\gamma(f)}$, *softmax attention acts as a hardmax* and the computation is performed as a uniform average over the tokens with argmax attention.

As can be seen from this analysis, while these design choices may seem like minutiae, they have outsized effects on the analysis, and this fact has been observed in previous work (Jerad et al., 2025). There is also empirical evidence that attention does indeed concentrate on only a few tokens (Bietti et al., 2023; Rogers et al., 2021) and that finite precision does have a noticeable impact on LLM behaviors (He & Lab, 2025).

### 3.3 Infinite-Precision Attention

Deviating from previous works, we also provide results when the transformer's attention computations (and indeed, all internal computations) are performed at infinite precision. In this case, we do not need to make careful assumptions about rounding. Instead, however, there is an additional subtlety about the scaling of the attention logits. In particular, given infinite precision and bounded weight matrices, the effect of the $\tau$-suffix on the LT's computation must *always* decay to 0 as the length of the input sequence diverges to infinity. This is undesirable as it precludes important functions which transformers are empirically capable of learning, e.g., the induction head. To alleviate this shortcoming, we propose scaling *only the $\tau$-suffix logits* by a logarithmic factor:

$$a_{i,j}^{(l,h)} = (\boldsymbol{y}_j^{(l-1)})^\top \boldsymbol{K}_{l,h}^\top \boldsymbol{Q}_{l,h} \boldsymbol{y}_i^{(l-1)} + \log i \cdot \phi_{l,h}(j, i),$$

$$\boldsymbol{Y}_i^{(l)} = \boldsymbol{y}_i^{(l-1)} + \sum_{h=1}^{H} \frac{\sum_{j=1}^{i} \exp\left(a_{i,j}^{(l,h)}\right) \boldsymbol{V}_{l,h} \boldsymbol{y}_j^{(l-1)}}{\sum_{j=1}^{i} \exp\left(a_{i,j}^{(l,h)}\right)}. \tag{1}$$

Depending on the size of the $\tau$-suffix positional embeddings, this scaling increases the expressivity of LTs to give three different possible behaviors. Consider the computation of the $h$th attention head in the first layer. For $j < i - \tau$, the contribution of the $j$th token will be proportional to $\exp\left(\boldsymbol{E}_{x_j}^\top \boldsymbol{K}_{1,h}^\top \boldsymbol{Q}_{1,h} \boldsymbol{E}_{x_i}\right)$. Therefore for any $s \in \Sigma$, the total contribution of all tokens equal to $s$ will be $\exp\left(\boldsymbol{E}_s^\top \boldsymbol{K}_{1,h}^\top \boldsymbol{Q}_{1,h} \boldsymbol{E}_{x_i}\right) \cdot \mu(x_{\leq i})_s i$, where $\mu(x_{\leq i})_s = \frac{1}{i} \sum_{j=1}^i \mathbf{1}(x_j = s)$ is the empirical frequency of $s$ in the first $i$ tokens of $x$. On the other hand, the contribution of the $j$th token for $i - \tau \leq j \leq i$ is $\exp\left(\boldsymbol{E}_{x_j}^\top \boldsymbol{K}_{1,h}^\top \boldsymbol{Q}_{1,h} \boldsymbol{E}_{x_i}\right) \cdot i^{\phi_{1,h}(j,i)}$. This yields the following three regimes:

1. **Token Dominant** ($\max_{t \leq \tau} \phi_{1,h}(i - t, i) < 1$). For typical sequences, the empirical frequences $\mu(x_{\leq i})_s$ will be $\Theta(1)$ for large $i$. Therefore if $\phi$ is bounded below 1, as $i \to \infty$ the contribution of the $\tau$-suffix will grow negligible.

2. **Balanced** ($\max_{t \leq \tau} \phi_{1,h}(i - t, i) = 1$). Both the total contribution of all tokens equal to $s$, as well as tokens in the $\tau$-suffix with $\phi_{1,h}(j,i) = 1$, will be proportional to $i$, and thus affect the output of self-attention in a constant fashion as $i \to \infty$.

3. **Position Dominant** ($\max_{t \leq \tau} \phi_{1,h}(i - t, i) > 1$). The contribution of the $\tau$-suffix dominates that of the rest of the sequence, with the self-attention weights concentrating on those tokens $j$ which maximize $\phi_{1,h}(j,i)$.

Thus, the proposed scaling allows us to consider the full range of possible relative importance for the local information (found in the $\tau$-suffix) and global information (found in the $\tau$-prefix) of the input. In spite of these three qualitatively different regimes, we are able to provide a unified analysis which addresses LG in all three scenarios simultaneously.

We will operate in the infinite precision setting for our results on multi-layer transformers (Theorem 5.2). Intuitively, in the first layer, the tokens are in fact discrete and hard attention to a subset of these tokens may be desirable. Beyond the first layer, however, the token representations become continuous mixtures of these discrete objects and are in some sense more "inherently" continuous. This makes the infinite precision setup more suitable for this setting.

Lastly, we remark that the while the details of the finite- and infinite-precision analysis are quite different, the fundamental analysis technique is the same. Namely, we show that it is possible to *simulate* the behavior of the transformer on longer sequences using strings of bounded length. The implications of the theory for the data requirement vs. various parameters of the target function also align qualitatively for both precision regimes, and these insights match with our empirical results.

## 4  LENGTH GENERALIZATION WITH FINITE PRECISION

In this section, we give upper bounds on the length of training data required for a single-layer limit transformer to generalize to sequences of arbitrary length in the finite precision setting.

Let $f$ and $g$ be two single-layer limit transformers which are $\tau$-local, $\Delta$-periodic, translation invariant, and operate at $p$ finite bits of precision as described in Section 3. Let $\boldsymbol{V}_f, \boldsymbol{E}_s^f, (\boldsymbol{A}_f, \boldsymbol{B}_f)$ be the value matrix, token embedding, and MLP weights for $f$ (and analogously defined for $g$), and define

$$L_f = \max\{\|\boldsymbol{U}_f\|(1 + \|\boldsymbol{A}_f\|\|\boldsymbol{B}_f\|\|\psi_f\|)(\|\boldsymbol{V}_f(\boldsymbol{E}_s^f + \boldsymbol{p}_i)\| + \|\boldsymbol{E}_s^f + \boldsymbol{p}_i\| + \|b_f\|) \ : \ s \in \Sigma\},$$

$L_g$ similarly for $g$, and $L = L_f + L_g$. Finally, let $\gamma = \min\{\gamma(f), \gamma(g)\}$, with $\gamma(f)$ and $\gamma(g)$ as defined in Section 3.2. We first establish LG for single-layer transformers in an $\ell_\infty$ setting.

**Theorem 4.1.** *There exists an* $N = O\left(\max\left\{2^{p/\gamma}, \frac{L^2 \Delta^7 |\Sigma|^6 \tau^2}{\varepsilon^2}\right\}\right)$ *such that* $\|f(x) - g(x)\| \leq \varepsilon$ *for all* $|x| \leq N$ *implies that* $\|f(x) - g(x)\| = O(\varepsilon)$ *for any* sequence $x$.

*Proof sketch.* As discussed in Section 3.2, the output of each limit transformer depends roughly on the ratios between each token type entering hardmax attention. We construct a "simulation map" from a string $x$ of arbitrary length to a string $z$ of length $|z| \leq N$ which preserves these ratios up to $O(\varepsilon)$ error *simultaneously* for the tokens in attention in both $f$ and $g$. Since $f(z) \approx g(z)$ by assumption, this in turn implies that $f(x) \approx g(x)$. The complete proof is given in Appendix A.2. □

**Remarks.** Theorem 4.1 shows that, assuming that the input sequences are sufficiently long ($N \gtrsim 2^{p/\gamma}$), the desired training length scales polynomially in the periodicity parameter $\Delta$, the parameter norms $L$, the vocabulary size $|\Sigma|$, and the inverse accuracy $\varepsilon^{-1}$. The $N \gtrsim 2^{p/\gamma}$ constraint ensures that the softmax attention behaves as a hardmax as discussed in Section 3.2. Indeed, it is possible for this hardmax behavior to occur at smaller training lengths, implying that the training length $N$ need only scale with $\tau \Delta L^2 / \varepsilon^2$. See Section 6 for empirical support of this claim.

Theorem 4.1 bounds the test error when we have an $\ell_\infty$ bound on the error, i.e., when the error on *every sequence* of length at least $N$ is bounded by $\varepsilon$. In practice, it is more common to have a bound on the *average* error. The following theorem establishes that we can still achieve LG with respect to average error for a certain class of sequence distributions.

**Theorem 4.2.** *For any probability distribution $\mathcal{P} = (p_s)_{s \in \Sigma}$ over the token vocabulary $\Sigma$, define*

$$\|f - g\|_{n,\mathcal{P}} = \sum_{|x|=n} \mathbb{P}_{\mathcal{P}}(x) \|f(x) - g(x)\|,$$

*where $\mathbb{P}_{\mathcal{P}}(x) = \prod_{i=1}^{|x|} p_{x_i}$ is the probability of the sequence $x$ when the tokens are drawn i.i.d. from $\mathcal{P}$. Let $\mathcal{P} = (p_s)_{s \in \Sigma} \sim \mathrm{Dir}((\alpha_s)_{s \in \Sigma})$ be drawn from a Dirichlet distribution, and define*

$$\|f - g\|_n = \mathbb{E}_{\mathcal{P} \sim \mathrm{Dir}((\alpha_s)_{s \in \Sigma})}[\|f - g\|_{n,\mathcal{P}}].$$

*Let $\alpha_0 = \min_{s \in \Sigma} \alpha_s$. Then there exists*

$$N_0 = O\left(\max\left\{2^{p/\gamma}, \frac{16^{\frac{\alpha^*}{\alpha_0}} L^{2+2\alpha_0^{-1}} |\Sigma|^{4+2\alpha_0^{-1}} \Delta^5}{\alpha_0^{2\alpha_0^{-1}} \varepsilon^{2+2\alpha_0^{-1}}} \log \frac{|\Sigma| \Delta L}{\varepsilon}\right\}\right) = \widetilde{O}(\varepsilon^{-2-2\alpha_0^{-1}})$$

*such that if $\|f - g\|_N \leq \varepsilon$ for all $N \leq N_0$, we have that $\|f - g\|_T = O(\varepsilon^{1/2})$ for any $T$.*

*Proof sketch.* We show that with high probability over the draw of $(p_s)_{s \in \Sigma}$ and the resulting sequence $x$, the fraction of (token, positional embedding) pairs is close to its mean. For such sequences, we further show that the output of the limit transformer is approximately constant. This allows us to define a simulation map $\mathrm{sim} : \Sigma^T \to \Sigma^N$ from longer sequences to shorter ones which (1) satisfies $f(x) \approx f(\mathrm{sim}(x))$ and (2) does not transfer a large probability mass of long sequences in $\Sigma^T$ to a low-probability subset of short sequences in $\Sigma^N$. These two features of the simulation map allow us to control $\|f - g\|_T$ in terms of $\|f - g\|_N$ for any $T \geq N$. The full proof is given in Appendix A.3. $\square$

**Remarks.** We make two remarks on this result. First, the form of the sequence distribution is meant to ensure some regularity between sequences of longer and shorter lengths. The need for some such regularity assumption is inevitable. For instance, an obvious example would be where the distribution over shorter sequences has support only on sequences with tokens in $\Sigma_{\mathrm{short}} \subsetneq \Sigma$, while the distribution over longer sequences has support $\Sigma \setminus \Sigma_{\mathrm{short}}$. The switch can occur at an arbitrarily large sequence length, so a bound on the required training length cannot exist in such a setting. This counterexample can also be approximated without requiring the probability of certain sequences to be exactly equal to 0. We expect a similar result to hold for sequences with some form of regularity in terms of token ratios between shorter and longer sequences; e.g., if the sequences are drawn from a Markov chain, concentration of the token ratios to the stationary distribution may be sufficient. It is an interesting direction for future work to establish minimal conditions on the sequence distribution for LG to occur in the average case. Second, as a corollary to our proof technique, we can strengthen the error dependence of our bound when $\|f - g\|_{N,\mathcal{P}}$ is controlled *conditional* on $\min_{s \in \Sigma} p_s = \Omega(1)$. In this case, the LG error does not suffer from the quadratic increase from $\varepsilon$ to $\varepsilon^{1/2}$ as in Theorem 4.2, but the required training length to achieve $O(\varepsilon)$ error is longer. The proof for this setting can also easily be extended to the case where the tokens are drawn from a *fixed* categorical distribution and the probability $p_s$ for each token is at least a constant.

## 5 SOFT ATTENTION TRANSFORMERS WITH INFINITE PRECISION

In this section, we provide upper bounds on the length of training sequences required for two-layer limit transformers operating at infinite precision to generalize to sequences of arbitrary length. Recall

that we have made the assumption that transformers which operate at infinite precision only have the $\tau$-suffix logits scaled by $\log(\text{token index})$, and thus have forward pass given by (1). The key quantity which governs the minimum training length is the following complexity measure.

**Definition 5.1** (Complexity and positional margin). Let $\mathcal{F}_\tau$ be the class of depth 2 transformers transformers which are $\tau$-local, translation invariant, operate at infinite precision, use no positional information in the second layer, and have nonnegative $\phi_{l,h}$[1]. For a transformer $f \in \mathcal{F}_\tau$, with key, query and value matrices $\{(\boldsymbol{K}_{1,h}, \boldsymbol{Q}_{1,h}, \boldsymbol{V}_{1,h})\}_{h \in [H]} \cup \{(\boldsymbol{K}_{2,1}, \boldsymbol{Q}_{2,1}, \boldsymbol{V}_{2,1})\}$, MLP weights $\{(\boldsymbol{A}_l, \boldsymbol{B}_l)\}_{l \in \{1,2\}}$, embeddings $\|\boldsymbol{E}_s\| \leq 1$, and unembedding $\boldsymbol{U}$, define the *complexity* $C(f)$ as

$$C(f) := \exp\left(\text{poly}\left(\left\{\|\boldsymbol{V}_{1,h}\|_{op}, \left\|\boldsymbol{K}_{1,h}^\top \boldsymbol{Q}_{1,h}\right\|_{op}\right\}_{h \in [H]}, \|\boldsymbol{A}_1\|_{op}, \|\boldsymbol{B}_1\|_{op}, \left\|\boldsymbol{K}_{2,1}^\top \boldsymbol{Q}_{2,1}\right\|_{op}\right)\right)$$
$$\cdot \text{poly}\left(\|\boldsymbol{V}_2\|_{op}, \|\boldsymbol{A}_2\|_{op}, \|\boldsymbol{B}_2\|_{op}, \|\boldsymbol{U}\|_{op}, \tau, |\Sigma|\right)$$

Moreover, define the *positional margin* $\gamma(f)$ by

$$\gamma(f) := \min_{h \in H}\left(\max \mathcal{P}_h - \max\{p \in \mathcal{P}_h : p \neq \max \mathcal{P}_h\}\right)$$

where $\mathcal{P}_h := \{\phi_{1,h}(i - t, i)\}_{0 \leq t \leq \tau} \cup \{1\}$ is the set of positional embedding values in the $h$th head.

**Theorem 5.2.** *Let $f, g \in \mathcal{F}_\tau$. There exists $N \lesssim \left(\max(C(f), C(g))\varepsilon^{-1}\right)^{\max(\gamma(f)^{-1}, \gamma(g)^{-1}, 3)}$ such that $\|f(x) - g(x)\| \leq \varepsilon$ for all $|x| \leq N$ implies that $\|f(x) - g(x)\| = O(\varepsilon)$ for* any *sequence $x$.*

*Proof sketch.* Similar to before, our goal is to, given an arbitrary string $x$, construct a simulation $z$ which satisfies $f(x) \approx f(z)$ and $g(x) \approx g(z)$. Our first observation is that $\boldsymbol{Y}_i^{(1)}$, the output of the first layer of the transformer in the $i$th position, can be written as a Lipschitz and bounded function of both the $\tau$-suffix $x_{i-\tau:i}$ and the empirical histogram up to token $i$, $\mu(x_{\leq i}) := \frac{1}{i}\sum_{j=1}^i \mathbf{e}_{x_j}$.[2] As such, the output of the two-layer transformer depends continuously on the empirical joint distribution of $\{(x_{i-\tau:i}, \mu(x_{\leq i})\}_{i \in [|x|]}$. We would thus like for the simulation $z$ to approximately preserve this distribution. To do so, we construct a *random* simulation $z$ by randomly sampling a subset of the tokens in $x$, show in expectation that the outputs are preserved, and invoke the probabilistic method. In particular, the following "key simulation lemma" shows that such a subset does indeed exist.

**Lemma 5.3.** *Let $p : [S]^{\tau+1} \times \Delta^S \to \mathbb{R}^m$ be a fixed function, which is $L$ Lipschitz in its second argument and uniformly bounded by $G$. Then, there exists a subset $\mathcal{I} \subset [T]$ such that, if $z = x_\mathcal{I}$, then $||\mathcal{I}| - n| \leq \tau + 1 + n^{1/3}$ and*

$$\left\|\frac{1}{T}\sum_{t=1}^T p(x_{t-\tau:t}, \mu(x_{\leq t})) - \frac{1}{|z|}\sum_{t=1}^{|z|} p(z_{t-\tau:t}, \mu(z_{\leq t}))\right\| \lesssim \frac{(G+L)(\tau+1)}{n^{1/3}}.$$

The proof of Lemma 5.3 proceeds as follows. In order to preserve the empirical distribution over $\tau$-suffices, we would like for the simulation $z$ to include large (i.e., $\omega(1)$ in size) contiguous blocks of $x$. To do so, we consider a *Markov chain* $(i_1, \ldots, i_T)$ on the state space $\{0, 1\}$, with stationary distribution $\mathbb{P}(i_j = 1) = n/T$ and transition $\mathbb{P}(i_{j+1} = 0 \mid i_j = 1) \ll 1$. Letting $\mathcal{I} = \{j : i_j = 1\}$, one can show that the choice $z = x_\mathcal{I}$ yields a good simulation in expectation. The proof of Lemma 5.3, as well as the full proof of Theorem 5.2, are deferred to Appendix B. $\quad\square$

**Remarks.** The complexity measure in Definition 5.1 scales exponentially in the first layer weight norms. This is unavoidable, as the Lipschitz constant of the first layer softmax scales exponentially in $\|\boldsymbol{K}_{1,h}^\top \boldsymbol{Q}_{1,h}\|_{op}$ when considered as a function on the space of probability measures, which is necessary when considering LG as we must have a unified bound independent of the sequence length $T$. Other works (Kim et al., 2021; Wang et al., 2023; Castin et al., 2024) achieve bounds on the Lipschitz constant of attention which do not scale exponentially in the model weights, but which

---

[1]As per the discussion in Section 3.3, all $\phi_{l,h} < 1$ yield the same "token-dominant" regime, and hence assuming $\phi_{l,h} \geq 0$ does not affect expressivity.

[2]Infinite precision attention is necessary here to show that this function is indeed Lipschitz.

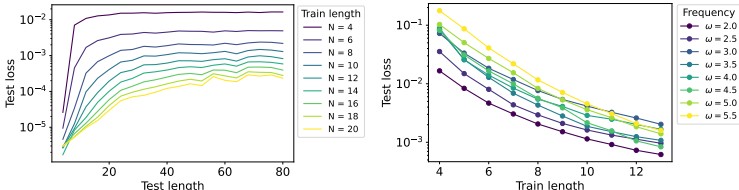

Figure 1: Experiments on SimpleTask. **Left:** Test loss as a function of test length and train length, for fixed $\omega$. For each fixed train length, as test length increases, the test loss plateaus at a finite value. **Right:** Final test loss as a function of train length and $\omega$. The value the test loss plateaus at decreases monotonically with train length, and increases monotonically with $\omega$.

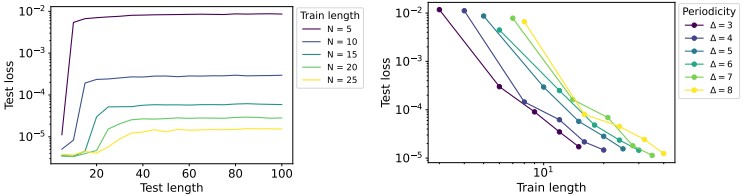

Figure 2: Experiments on ModPTask. **Left:** Test loss as a function of test length and train length, for fixed $\Delta$. For each fixed train length, as test length increases, the test loss plateaus at a finite value. **Right:** Final test loss as a function of train length and $\Delta$. The value the test loss plateaus at decreases monotonically with train length, and increases monotonically with $\Delta$.

depend on the sequence length and which are therefore not applicable to our setting. Moreover, for certain tasks which can be naturally expressed by a two-layer transformer, the complexity is mild. Consider the following *in-context k-gram* task, which is a generalization of the induction head (Olsson et al., 2022):

**Definition 5.4.** Let $\Sigma = [S]$. We say that $f^*$ is an *in-context k-gram* estimator if its output on a sequence $x$ is the empirical distribution of the token following all occurrences of $x_{T-k+1:T}$[3] i.e

$$f^*(x_{1:T}) = \frac{\sum_{t=k+1}^{T} \mathbf{1}(x_{t-k:t-1} = x_{T-k+1:T}) \cdot \mathbf{e}_{x_t}}{\sum_{t=k+1}^{T} \mathbf{1}(x_{t-k:t-1} = x_{T-k+1:T})} \in \mathbb{R}^S.$$

Nichani et al. (2024) show that $f^*$ can be approximated by a depth two transformer with $k-1$ heads in the first layer, and local, translational invariant positional embeddings with $\tau = k - 1$. In Appendix B.3, we show heuristically that $f^*$ can be approximated up to error $\varepsilon$ by a transformer $f$ with complexity $C(f) = \varepsilon^{-\Theta(k^2)}$ and $\gamma(f) \geq 1$, so a training length of $\varepsilon^{-\Theta(k^2)}$ suffices for LG.

We also remark that in Theorem 5.2, the training length $N$ scales exponentially with the inverse margin $1/\gamma$. This mimics the bound in Theorem 4.1, which contains an $\exp(\gamma^{-1})$ dependence on the *logit*-margin. Whether these margins matter for LG empirically or are simply an artifact of our analysis is an interesting question for future work.

## 6 EXPERIMENTS

**Single-layer Transformers.** We next provide empirical support for the conclusions of Theorem 4.1 and 4.2. We consider the following two synthetic tasks:

SimpleTask: The vocabulary is $\Sigma = \{0, 1, 2\}$. Given an input sequence $x_{1:T} = (x_1, \ldots, x_T) \in \Sigma^T$, define $c_s(x) = \sum_{t=1}^{T} \mathbf{1}(x_t = s)$ to count the number of tokens equal to $s$. The output $f^*$ is given by $f^*(x_{1:T}) = \sigma\left(\frac{c_0(x) - c_1(x)}{c_0(x) + c_1(x)}\right)$, where $\sigma(z) = \sin(\omega z)$ for some $\omega \in \mathbb{R}$. One observes that $f^*$ is expressible by a one-layer limit transformer with no positional embeddings and $L = \Theta(\omega)$.

---

[3]If there is no such occurrence within $x$, the behavior of $f^*(x)$ can be arbitrary.

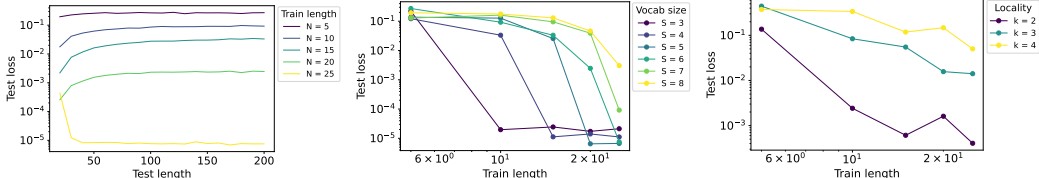

Figure 4: Experiments on the in-context $k$-gram task. **Left:** Test loss as a function of test length and train length, for fixed $k$ and $S$. For each fixed train length, as test length increases, the test loss plateaus at a finite value. **Middle:** Final test loss as a function of train length and $S$, for fixed $k$. The value the test loss plateaus at decreases monotonically with training length, and increases with $S$. **Right:** Final test loss as a function of train length and $k$, for fixed $S$. The value the test loss plateaus at increases monotonically with $k$.

ModPTask: The vocabulary is $\Sigma = \{0, 1\}$. Given a period $p$ and index $k$, the output is the average of all tokens in positions which are $k \mod p$:

$$f^*(x_{1:T}) = \frac{\sum_{t=1}^{T} \mathbf{1}(x_t = 1, t \equiv k \mod p)}{\sum_{t=1}^{T} \mathbf{1}(t \equiv k \mod p)}.$$

One observes that $f^*$ is expressible by a limit transformer with $\Delta = p$ and $L = \Theta(1)$.

We train depth 1 transformers (consisting of a single self-attention layer followed by an MLP layer) on SimpleTask for varying frequencies $\omega$ and ModPTask for varying periods $p$. For a fixed training length $N$, we train models on sequences of length $T \leq N$, and compute the test loss on sequences of length $T' \geq N$. More details on the experimental methodology are presented in Appendix C; sketches for both constructions are provided in Appendix C.1.

Results for SimpleTask and ModPTask are presented in Figure 1 and Figure 2 respectively. In the leftmost panes of both figures, we observe that the test loss plateaus as the test length increases. In the rightmost panes of both figures, we observe that the value at which the test loss plateaus at decreases monotonically with the training length. This provides qualitative support for the conclusions of Theorem 4.1, in particular that (i) given a target accuracy $\varepsilon$, tasks expressible by a one-layer limit transformer have a finite $N$ such that a model which fits the task on sequences up to length $N$ acheives $\varepsilon$ error on sequences of all length and (ii) the value of this $N$ increases monotonically as $\varepsilon$ increases. Moreover, the rightmost pane in Figure 1 shows that $N$ scales with the parameter norm $L$, while the rightmost pane in Figure 2 shows that $N$ scales with the periodicity parameter $\Delta$.

The proof of Theorem 4.1 relies on the "hardmax" attention behavior discussed in Section 3.2. To check the validity of this assumption, trained on the ModPTask with $p = 5$ for varying training lengths, and compute the post-softmax attention probabilities on a batch of test sequences. In Figure 3, we observe that the positions not equal to $k \mod p$ receive near zero attention probabilities while those in positions equal to $k \mod p$ receive nearly the same attention probability (the dashed black line). This provides evidence that, for large enough training length, the models are indeed operating in the hardmax regime.

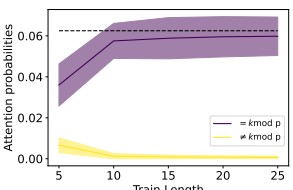

Figure 3: For the ModPTask, the softmax attention approximates uniform attention on all positions $\equiv k \mod p$.

**Two-layer Transformers.** We next provide empirical support for the conclusions of Theorem 5.2. We train depth 2 transformers on the *in-context $k$-gram* synthetic task, as defined in Definition 5.4. Additional experimental details are given in Appendix C. Results are presented in Figure 4. In the leftmost pane, we again observe that test loss plateaus as test length increases. Both the middle and rightmost plots show that as the training length increases, the limiting test loss decreases. Moreover, the middle plot shows the value of this limiting test loss increases with the alphabet size $S$ (when we fix $k = 2$), while the rightmost plot shows that it increases with $k$ (when we fix $S = 2$). This matches the qualitative dependence of the complexity measure $C(f)$ on both $S$ and $\tau$.

## 7 CONCLUSION

In this paper, we provided quantitative bounds on the training length required for LG to occur, in settings including finite- and infinite-precision attention, one- and two-layer transformers, and $\ell_\infty$ and average error control. Our results show that this minimum training length scales with the parameter norms of the transformer, the periodicity $\Delta$, locality $\tau$, alphabet size $|\Sigma|$, and inverse error $\varepsilon^{-1}$. Unifying our analyses is the high level argument that LG occurs whenever the forward pass of a transformer on a longer string can be "simulated" by that of a shorter string contained in the training set. Qualitative support for the derived scalings are presented in Section 6.

One interesting direction of future work is to extend our results to transformers with larger depth. In particular, it would be interesting to relate the minimum training length $N$ to other notions of complexity such as the length of the corresponding C-RASP program. Moreover, it would be interesting to extend our average-case analysis in Theorem 4.2 to broader classes of distributions over sequences. Finally, it is an important question to characterize how different positional embedding schemes, which empirically improve LG, affect the minimum training length $N$.

## ETHICS STATEMENT

This paper presents work whose goal is to advance the field of Machine Learning. There are many potential societal consequences of our work, none which we feel must be specifically highlighted here.

**LLM Usage:** We used LLMs to check for grammatical errors in the paper.

## REPRODUCIBILITY STATEMENT

Experimental details are provided in Appendix C, and code is attached to the submission.

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

# A  OMITTED PROOFS FROM SECTION 4

In all of the proofs and discussions for this section, we will assume that all sequences are of length at least $2^{p/\min\{\gamma(f),\gamma(g)\}}$. As discussed in Section 3.2, this means that attention will operate as an argmax over tokens with maximal logits, and therefore the computations of the transformer will be performed over a subset of tokens determined by (token, position mod $\Delta$) in the $\tau$-prefix as well as potentially some tokens in the $\tau$-suffix. As, such, it will be useful to make the following definition.

**Definition A.1.** Define the *token counting function* $n : \Sigma \times \mathbb{Z} \times \Sigma^* \to \mathbb{Z}_{\geq 0}$ by

$$n(s, i, x) = \#\{j \in 1, \dots, |x| - \tau \mid x_j = s \text{ and } j \equiv i \pmod{\Delta}\}.$$

That is, $n(s, i, x)$ counts the number of times that token $s$ appears in a position which is $i \pmod{\Delta}$ in $x$ before the $\tau$-suffix. In the case where there are no positional embedding vectors, we similarly define

$$n(s, x) = \#\{j \in 1, \dots, |x| - \tau \mid x_j = s\}.$$

The subset of (token, position mod $\Delta$) pairs which enter the hard attention mechanism is determined by the final token and its positional embedding vector. Thus, in the constructions which follow, it will be important that the original long sequence $x$ ($|x| = T$) is simulated by a shorter string $z$ ($|z| = N$) such that $x_T = z_T$ and $T \equiv N \pmod{\Delta}$. This will ensure that the hard attention mechanism considers the same (token, positional embedding) pairs for both strings, and therefore that the output of the limit transformer can be approximated by preserving the ratios of these quantities.

With these concerns in mind, consider a single-head, single-layer limit transformer $f$ and an input string $x$ with $|x| \geq 2^{\gamma(f)}$. Let $A(x) \subseteq \Sigma \times [\Delta]$ be the set of (token, position mod $\Delta$) pairs in the $\tau$-prefix attended to by $f$ when parsing the final token $x_T$, and let $A^\tau(x)$ be the set of (token, position) pairs attended to in the $\tau$-suffix. Then the internal state of $f$ immediately after the attention layer is given by

$$\tilde{f}(x) = \frac{\sum_{(s,i)\in A(x)} n(s, i, x) \boldsymbol{V}_f(\boldsymbol{E}_s^f + \boldsymbol{p}_i) + \sum_{(s,i)\in A^\tau(x)} \boldsymbol{V}_f(\boldsymbol{E}_s^f + \boldsymbol{p}_i)}{\sum_{(s,i)\in A(x)} n(s, i, x) + |A^\tau(x)|}, \tag{2}$$

and the full computation is given by

$$f(x) = \boldsymbol{U}_f\left((\boldsymbol{E}_{x_T}^f + \tilde{f}(x)) + \boldsymbol{B}_f\psi_f(\boldsymbol{A}_f(\boldsymbol{E}_{x_T}^f + \tilde{f}(x)) + \boldsymbol{b}_f)\right). \tag{3}$$

## A.1  HELPER LEMMAS

For both of the finite-precision theorems, the following lemma relating $\tilde{f}$ and $f$ will be useful.

**Lemma A.2.** *Let $x$ and $z$ be two sequences with $|x| = T$ and $|z| = N$, and suppose that the final tokens are equal: $x_T = z_N$. Then $\|f(x) - f(z)\| \leq L_f^{\mathrm{MLP}}\|\tilde{f}(x) - \tilde{f}(z)\|$, where*

$$L_f^{\mathrm{MLP}} = \|\boldsymbol{U}_f\|(1 + \|\boldsymbol{A}_f\|\|\boldsymbol{B}_f\|\|\psi_f\|)$$

*is a bound on the Lipschitz constant of the transformer MLP.*

*Proof.* We can write

$$f(x) = \boldsymbol{U}_f\left((\boldsymbol{E}_{x_T}^f + \tilde{f}(x)) + \boldsymbol{B}_f\psi_f(\boldsymbol{A}_f(\boldsymbol{E}_{x_T}^f + \tilde{f}(x)) + \boldsymbol{b}_f)\right). \tag{4}$$

Because $x_T = z_N$, we have $\boldsymbol{E}_{x_T}^f = \boldsymbol{E}_{z_N}^f$. Straightforward applications of the triangle inequality and the submultiplicative inequality for operator norms then yield the desired result. $\square$

The following lemma bounds the norm of the output of a limit transformer in terms of the norms of the weight matrices and activation function.

**Lemma A.3.** *Define the following quantities:*

$$L_f^{\mathrm{MLP}} = \|\boldsymbol{U}_f\|(1 + \|\boldsymbol{A}_f\|\|\boldsymbol{B}_f\|\|\psi_f\|),$$

$$M_f^V = \max_{s\in\Sigma,\, i\in[\Delta]} \|\boldsymbol{V}_f(\boldsymbol{E}_s^f + \boldsymbol{p}_i)\|,$$

$$M_f^E = \max_{s \in \Sigma, i \in [\Delta]} \|\boldsymbol{E}_s^f + \boldsymbol{p}_i\|.$$

*Then setting*

$$M_f = L_f^{\mathrm{MLP}}(M_f^E + M_f^V + \|b_f\|),$$

*we have $\|f(x)\| \le M_f$ for all $x$.*

*Proof.* The proof is nearly identical to that of Lemma A.2. We begin by observing that $\tilde{f}(x)$ is a convex combination of terms of the form $\boldsymbol{V}_f(\boldsymbol{E}_s^f + \boldsymbol{p}_i)$, so in particular $\|\tilde{f}(x)\| \le M_f^V$. The results follows by using the expression for $f(x)$ in terms of $\tilde{f}(x)$ given in equation (4), the triangle inequality, and submultiplicativity of the operator norm. $\square$

### A.2 PROOF OF THEOREM 4.1

In this section, we give the proof of Theorem 4.1. We will first prove another helper lemma which will aid in our simulation string constructions.

**Lemma A.4.** *Let $\{p_i\}_{i=1}^n$ be an arbitrary finite probability distribution. For any integer $N \ge 1$, there exist nonnegative integers $\{m_i\}_{i=1}^n$ such that $\sum_{i=1}^n m_i = N$ and $|m_i/N - p_i| \le 1/N$ for all $i = 1, \ldots, n$. In particular, this also implies that the $m_i$ satisfy $|p_iN - m_i| \le 1$ for all $i$.*

*Proof.* Define $\tilde{m}_i = \lfloor p_iN \rfloor$ and let $R = N - \sum_{i=1}^n \tilde{m}_i$. Note that $0 \le R \le n$. For $i = 1, \ldots, R$, define $m_i = \tilde{m}_i + 1$ and for $i = R+1, \ldots, n$, define $m_i = \tilde{m}_i$. This choice of $m_i$ can easily be seen to have the desired properties. $\square$

We are now ready to prove Theorem 4.1, which we restate for convenience.

**Theorem 4.1.** *There exists an $N = O\left(\max\left\{2^{p/\gamma}, \frac{L^2\Delta^7|\Sigma|^6\tau^2}{\varepsilon^2}\right\}\right)$ such that $\|f(x) - g(x)\| \le \varepsilon$ for all $|x| \le N$ implies that $\|f(x) - g(x)\| = O(\varepsilon)$ for any sequence $x$.*

*Proof.* We will first give a proof assuming there are no positional embedding vectors $\boldsymbol{p}_i$. We will then show how to easily adapt the result to the case of positional embedding vectors.

Consider two limit transformers $f$ and $g$ and an input string $x$. Let $P_f$ be the positions attended to by $f$ and $P_g$ be the positions attended to by $g$ in the $\tau$-prefix of $x$ and assume WLOG that $|P_f| \le |P_g|$. Let $A_f = A_f(x) = \{x_i \mid i \in P_f\}$ be the set of tokens which $f$ attends to and define $A_g = A_g(x)$ similarly.

We construct the auxiliary string $z$ as follows. The $\tau$-suffix of $z$ is always equal to the $\tau$-suffix of $x$; in particular, this ensures that the final tokens of $x$ and $z$ are equal.

If $|P_f|, |P_g| \le 1/\varepsilon$, then the attention pattern in the $\tau$-prefix of $x$ can be directly recreated simultaneously for $f$ and $g$ using at most $2/\varepsilon$ tokens by just copying the union of the tokens in attention for $f$ and $g$ into $z$. In this case, by formulas (2) and (3), we will have $f(x) = f(z)$ and $g(x) = g(z)$ and therefore $\|f(x) - g(x)\| = \|f(z) - g(z)\| \le \varepsilon$. Thus, we will assume that at least $|P_g| \ge 1/\varepsilon$.

We first recreate the attention pattern of $f$. To simplify notation, let $n_s = n(s, x)$ and $m_s = n(s, z)$. If $|P_f| \le 1/\varepsilon$, then we simply set $z_{1:|P_f|} = x_{P_f}$ (i.e., we set the first $|P_f|$ tokens of $z$ equal to the attention pattern of $f$ on the $\tau$-prefix of $x$). The tokens which we will add later do not belong to $A_f$; thus, we will clearly have $f(x) = f(z)$. Thus, in the following construction, we will assume that $|P_f| \ge 1/\varepsilon$.

We first proceed with a slightly more fine-grained construction of Lemma A.4. For each $s \in A_f$, define

$$\tilde{m}_s = \left\lfloor \frac{\lceil 1/\varepsilon \rceil}{|P_f|} n_s \right\rfloor$$

and let

$$R = \left\lfloor \sum_{s \in A_f \cap A_g} \frac{\lceil 1/\varepsilon \rceil}{|P_f|} n_s - \sum_{s \in A_f \cap A_g} \tilde{m}_s \right\rfloor.$$

We must have $R \leq |A_f \cap A_g|$, so choose $I \subseteq A_f \cap A_g$ with $|I| = R$ and define $m_s = \tilde{m}_s + 1$ for $s \in I$ and $m_s = \tilde{m}_s$ for $s \in (A_f \cap A_g) \setminus I$. This defines $m_s$ for all $s \in A_f \cap A_g$ with the following two properties:

$$\left| m_s - \frac{\lceil 1/\varepsilon \rceil}{|P_f|} n_s \right| \leq 1 \quad \forall s \in A_f \cap A_g, \qquad 0 \leq \frac{\lceil 1/\varepsilon \rceil}{|P_f|} |P_f \cap P_g| - \sum_{s \in A_f \cap A_g} m_s < 1. \quad (5)$$

We then turn to $A_f \setminus A_g$ and define

$$R' = \left\lfloor \sum_{s \in A_f} \frac{\lceil 1/\varepsilon \rceil}{|P_f|} n_s - \sum_{s \in A_f \setminus A_g} \tilde{m}_s - \sum_{s \in A_f \cap A_g} m_s \right\rfloor.$$

It is clear that $0 \leq R' \leq |A_f \setminus A_g|$, so we can similarly choose $J \subseteq A_f \setminus A_g$ with $|J| = R'$ and define $m_s = \tilde{m}_s + 1$ for $s \in J$ and $m_s = \tilde{m}_s$ for $s \in (A_f \setminus A_g) \setminus J$. The $m_s$ defined in this way have the property that

$$\left| m_s - \frac{\lceil 1/\varepsilon \rceil}{|P_f|} n_s \right| \leq 1 \quad \forall s \in A_f \setminus A_g, \qquad \left| \frac{\lceil 1/\varepsilon \rceil}{|P_f|} |P_f \setminus P_g| - \sum_{s \in A_f \setminus A_g} m_s \right| \leq 1. \quad (6)$$

Combining results (5) and (6), we have the additional result that

$$\left| \lceil 1/\varepsilon \rceil - \sum_{s \in A_f} m_s \right| \leq \left| \frac{\lceil 1/\varepsilon \rceil}{|P_f|} |P_f \cap P_g| - \sum_{s \in A_f \cap A_g} m_s \right| + \left| \frac{\lceil 1/\varepsilon \rceil}{|P_f|} |P_f \setminus P_g| - \sum_{s \in A_f \setminus A_g} m_s \right| \leq 2. \quad (7)$$

In particular, combining the results of inequalities (5), (6), and (7), we can conclude that the $m_s$ satisfy

$$\left| \frac{m_s}{\sum_{s' \in A_f} m_{s'}} - \frac{n_s}{|P_f|} \right| = O(\varepsilon). \quad (8)$$

We can use these inequalities to bound the difference between $f(x)$ and $f(z)$. Define

$$\tilde{f}^{\setminus \tau}(x) = \frac{\sum_{s \in A_f} n_s \boldsymbol{V}_f \boldsymbol{E}_s^f}{\sum_{s' \in A_f} n_{s'}}$$

to be the internal state of $f$ immediately after the attention layer, ignoring the $\tau$-suffix. Letting $P_f^\tau$ be the set of positions $f$ attends to in the $\tau$-suffix of $x$, observe that we can write

$$\tilde{f}(x) = \tilde{f}^{\setminus \tau}(x) \cdot \frac{\sum_{s' \in A_f} n_{s'}}{\sum_{s' \in A_f} n_{s'} + |P_f^\tau|} + \frac{\sum_{i \in P_f^\tau} \boldsymbol{V}_f \boldsymbol{E}_{x_i}^f}{\sum_{s' \in A_f} n_{s'} + |P_f^\tau|}.$$

It therefore follows that

$$\|\tilde{f}(x) - \tilde{f}^{\setminus \tau}(x)\| \leq \left| 1 - \frac{1}{1 + \frac{|P_f^\tau|}{\sum_{s' \in A_f} n_{s'}}} \right| \|\tilde{f}^{\setminus \tau}(x)\| + \frac{\tau M_f^V}{\sum_{s' \in A_f} n_{s'}} \quad (9)$$

$$\leq \frac{2\tau}{\sum_{s' \in A_f} n_{s'}} \cdot M_f^V + \frac{\tau M_f^V}{\sum_{s' \in A_f} n_{s'}} \quad (10)$$

$$\leq 3 M_f^V \tau \varepsilon. \quad (11)$$

Inequalities (9) and (10) both use the fact that $|P_f^\tau| \leq \tau$ and $\|\boldsymbol{V}_f \boldsymbol{E}_s^f\| \leq M_f^V$. Inequality (10) additionally uses that

$$\frac{1}{1 + \frac{|P_f^\tau|}{\sum_{s' \in A_f} n_{s'}}} \geq 1 - \frac{2|P_f^\tau|}{\sum_{s' \in A_f} n_{s'}}$$

provided that $|P_f^\tau|/\sum_{s'\in A_f} n_{s'} \leq 1/2$. Since $|P_f| = \sum_{s\in A_f} n_s > 1/\varepsilon$ and $|P_f^\tau| \leq \tau$, this inequality holds for $\varepsilon$ small enough. The final inequality (35) simply uses the fact that $\sum_{s\in A_f} n_s = |P_f| > 1/\varepsilon$.

If we then define

$$\tilde{f}^{\backslash\tau}(z) = \frac{\sum_{s\in A_f} m_s \boldsymbol{V}_f \boldsymbol{E}_s^f}{\sum_{s'\in A_f} m_{s'}},$$

a similar calculation will then yield

$$\|\tilde{f}(z) - \tilde{f}^{\backslash\tau}(z)\| \leq \frac{3\tau M_f^V}{\sum_{s\in A_f} m_s} \leq \frac{3\tau M_f^V}{\lceil 1/\varepsilon\rceil - 2} \leq 4M_f^V \tau\varepsilon$$

for $\varepsilon$ small enough. By the triangle inequality, we then have

$$\|\tilde{f}(x) - \tilde{f}(z)\| \leq \|\tilde{f}(x) - \tilde{f}^{\backslash\tau}(x)\| + \|\tilde{f}(z) - \tilde{f}^{\backslash\tau}(z)\| + \|\tilde{f}^{\backslash\tau}(x) - \tilde{f}^{\backslash\tau}(z)\|$$
$$= \|\tilde{f}^{\backslash\tau}(x) - \tilde{f}^{\backslash\tau}(z)\| + O(M_f^V \tau\varepsilon). \tag{12}$$

We can now bound $\|\tilde{f}^{\backslash\tau}(x) - \tilde{f}^{\backslash\tau}(z)\|$. We have

$$\|\tilde{f}^{\backslash\tau}(x) - \tilde{f}^{\backslash\tau}(z)\| = \left\| \frac{\sum_{s\in A_f} n_s \boldsymbol{V}_f \boldsymbol{E}_s^f}{|P_f|} - \frac{\sum_{s\in A_f} m_s \boldsymbol{V}_f \boldsymbol{E}_s^f}{\sum_{s'\in A_f} m_{s'}} \right\|$$

$$\leq M_f^V \sum_{s\in A_f} \left| \frac{n_s}{|P_f|} - \frac{m_s}{\sum_{s'\in A_f} m_{s'}} \right|$$

$$= O(M_f^V |\Sigma|\varepsilon). \tag{13}$$

Equation 13 uses the bound from (8) and the fact that $|A_f| \leq |\Sigma|$. Plugging (13) into (12), we obtain

$$\|\tilde{f}(x) - \tilde{f}(z)\| = O(M_f^V (|\Sigma| + \tau)\varepsilon).$$

Applying Lemma A.2, we then have

$$\|f(x) - f(z)\| = O(L_f^{\mathrm{MLP}} M_f^V (|\Sigma| + \tau)\varepsilon). \tag{14}$$

We will refer to the portion of $z$ which has been defined up to now as the *f-prefix* of $z$.

It remains to extend $z$ so that it can simulate the behavior of $g$ *without* adding any tokens in $A_f$ so as to preserve the previous calculations. There are now two cases depending on the size of $P_f \cap P_g$. First, suppose that $|P_f \cap P_g|/|P_g| \leq \varepsilon$. By Lemma A.4, there exist $\tilde{m}_s$ such that

$$\left| \tilde{m}_s - \frac{\lceil 1/\varepsilon^2\rceil}{|P_g|} n_s \right| \leq 1 \quad \forall s \in A_g, \qquad \sum_{s\in A_g} \tilde{m}_s = \lceil 1/\varepsilon^2\rceil.$$

We now define $m_s = \tilde{m}_s$ for $s \in A_g \setminus A_f$. Combined with the earlier definitions of $m_s$ for $s \in A_f \cap A_g$ above, this defines $m_s$ for all $s \in A_g$. Furthermore, we have

$$\sum_{s\in A_g\setminus A_f} m_s = \lceil 1/\varepsilon^2\rceil - \sum_{s\in A_f\cap A_g} \tilde{m}_s$$

$$\geq \lceil 1/\varepsilon^2\rceil - \sum_{s\in A_f\cap A_g} \left( \frac{\lceil 1/\varepsilon^2\rceil}{|P_g|} n_s + 1 \right)$$

$$= \lceil 1/\varepsilon^2\rceil - \frac{|P_f \cap P_g|}{|P_g|} \lceil 1/\varepsilon^2\rceil - |A_f \cap A_g|$$

$$\geq \lceil 1/\varepsilon^2\rceil(1 - \varepsilon) - |\Sigma| \tag{15}$$

$$\geq \lceil 1/\varepsilon^2\rceil(1 - 2\varepsilon)$$

for $\varepsilon$ small enough. (Here, (15) uses the assumption that $|P_f \cap P_g|/|P_g| \leq \varepsilon$.) Thus, using similar logic as the derivation for inequality (8), we obtain

$$\left| \frac{m_s}{\sum_{s' \in S_g \setminus S_f} m_{s'}} - \frac{n_s}{|P_g|} \right| = O(\varepsilon).$$

We now show that the terms in the $\tau$ suffix *and* in $A_f \cap A_g$ do not contribute much to $g$. Define

$$\tilde{g}^{\setminus \tau, f}(x) = \sum_{s \in A_g \setminus A_f} \frac{n_s \boldsymbol{V}_g \boldsymbol{E}_s^g}{|P_g|}.$$

We have the following bound:

$$\|\tilde{g}(x) - \tilde{g}^{\setminus \tau, f}(x)\| = \left\| \frac{\sum_{s \in A_g} n_s \boldsymbol{V}_g \boldsymbol{E}_s^g + \sum_{i \in P_g^\tau} \boldsymbol{V}_g \boldsymbol{E}_{x_i}^g}{|P_g| + |P_g^\tau|} - \sum_{s \in A_g \setminus A_f} \frac{n_s \boldsymbol{V}_g \boldsymbol{E}_s^g}{|P_g|} \right\|$$

$$\leq \frac{\sum_{i \in P_g^\tau} \|\boldsymbol{V}_g \boldsymbol{E}_{x_i}^g\|}{|P_g| + |P_g^\tau|} + \frac{\sum_{s \in A_g \cap A_f} n_s \|\boldsymbol{V}_g \boldsymbol{E}_{x_i}^g\|}{|P_g| + |P_g^\tau|} + \frac{\sum_{s \in A_g \setminus A_f} n_s \|\boldsymbol{V}_g \boldsymbol{E}_s^g\|}{|P_g|} \left| \frac{|P_g|}{|P_g| + |P_g^\tau|} - 1 \right|$$

$$\leq M_g^V \tau \varepsilon + \frac{M_g^V |P_f \cap P_g|}{|P_g|} + \frac{|P_g \setminus P_f| M_g^V}{|P_g|} \cdot \frac{2|P_g^\tau|}{|P_g|} \tag{16}$$

$$\leq M_g^V \tau \varepsilon + M_g^V \varepsilon + 2 M_g^V \tau \varepsilon$$

$$\leq 4 M_g^V \tau \varepsilon. \tag{17}$$

Inequality (16) used the fact that $|P_g| \geq 1/\varepsilon$, $|P_g^\tau| \leq \tau$, $\|\boldsymbol{V}_g \boldsymbol{E}_s^g\| \leq M_g^V$, and $|P_g|/(|P_g| + |P_g^\tau|) \geq 1 - 2|P_g^\tau|/|P_g|$ whenever $|P_g^\tau|/|P_g|$ is small enough (which it will be for small enough $\varepsilon$).

If we define

$$\tilde{g}^{\setminus \tau, f}(z) = \sum_{s \in A_g \setminus A_f} \frac{m_s \boldsymbol{V}_g \boldsymbol{E}_s^g}{\sum_{s' \in A_g \setminus A_f} m_{s'}},$$

then the same logic as used to derive (17) can be used to show that $\|\tilde{g}^{\setminus \tau, f}(z) - \tilde{g}(z)\| = O(M_g^V \varepsilon)$. This is because the critical facts that

$$\frac{\sum_{s \in A_g \cap A_f} m_s}{\sum_{s \in A_g \setminus A_f} m_s} = O(\varepsilon),$$

analogous to $|P_f \cap P_g|/|P_g|$; and

$$\frac{|P_g^\tau|}{\sum_{s \in A_g \setminus A_f} m_s} = O(\tau \varepsilon^2) = O(\varepsilon)$$

analogous to $|P_g^\tau|/|P_g| = O(\tau \varepsilon)$.

We can now use inequality (17), the triangle inequality, and Lemma A.2 to bound the error for $g$. We have

$$\|\tilde{g}(x) - \tilde{g}(z)\| \leq \|\tilde{g}(x) - \tilde{g}^{\setminus \tau, f}(x)\| + \|\tilde{g}^{\setminus \tau, f}(z) - \tilde{g}(z)\| + \|\tilde{g}^{\setminus \tau, f}(x) - \tilde{g}^{\setminus \tau, f}(z)\|$$

$$\leq O(M_g^V \tau \varepsilon) + \sum_{s \in A_g \setminus A_f} \left| \frac{m_s}{\sum_{s' \in A_g \setminus A_f} m_{s'}} - \frac{n_s}{|P_g|} \right| M_g^V$$

$$= O(M_g^V (\tau + |\Sigma|) \varepsilon).$$

Lemma A.2 then gives $\|g(x) - g(z)\| = O(L_g^{\mathrm{MLP}} M_g^V (\tau + |\Sigma|)\varepsilon)$. This completes the case when $|P_f \cap P_g|/|P_g| \leq \varepsilon$.

Otherwise we have $|P_f \cap P_g|/|P_g| > \varepsilon$. In this case, we have $|P_g| < |P_f \cap P_g|/\varepsilon \leq |P_f|/\varepsilon$. We now consider two further subcases. Again if $|P_f| \leq 1/\varepsilon$, we then have $|P_g| \leq 1/\varepsilon^2$. Thus, there are at most $1/\varepsilon + 1/\varepsilon^2 + \tau = O(1/\varepsilon^2)$ tokens in the union of the attention patterns of $f$ and $g$ on $x$. Thus, by setting $z$ equal to the collection of all tokens in this union of attention patterns, we have $f(x) = f(z)$, $g(x) = g(z)$, $\|f(x) - g(x)\| = \|f(z) - g(z)\| = O(\varepsilon)$, and $|z| = O(1/\varepsilon^2)$ as desired. Thus, we may assume that $|P_f| > 1/\varepsilon$.

Let $s^* = \operatorname{argmax}_{s \in A_f} n_s$. Note that since $|P_f| \geq 1/\varepsilon$ and $|A_f| \leq |\Sigma|$, we must have $n_{s^*} \geq |P_f|/|\Sigma|$. For $s \in A_g \setminus A_f$, we first define $\tilde{m}_s$ by

$$\tilde{m}_s = \left\lfloor \frac{m_{s^*}}{n_{s^*}} \cdot n_s \right\rfloor.$$

Again similar to Lemma A.4, we define

$$R = \lfloor \sum_{s \in A_g \setminus A_f} \frac{m_{s^*}}{n_{s^*}} n_s - \sum_{s \in A_g \setminus A_f} m_s \rfloor.$$

We again have $R \leq |A_g \setminus A_f|$, so we can choose $I \subseteq A_g \setminus A_f$, $|I| = R$, and define $m_s = \tilde{m}_s + 1$ for $s \in I$ and $m_s = \tilde{m}_s$ for $s \in (A_g \setminus A_f) \setminus I$. In this way, we have

$$\left| m_s - \frac{m_{s^*}}{n_{s^*}} n_s \right| \leq 1 \quad \forall s \in A_g \setminus A_f, \qquad \left| \sum_{s \in A_g \setminus A_f} \frac{m_{s^*}}{n_{s^*}} n_s - \sum_{s \in A_g \setminus A_f} m_s \right| \leq 1. \tag{18}$$

Note that in addition, we have

$$\sum_{s \in A_g \setminus A_f} m_s \leq \sum_{s \in A_g \setminus A_f} \frac{m_{s^*}}{n_{s^*}} n_s + 1 \tag{19}$$

$$\leq \sum_{s \in A_g \setminus A_f} \frac{\frac{\lceil 1/\varepsilon \rceil}{|P_f|} n_{s^*} + 1}{n_{s^*}} n_s + 1 \tag{20}$$

$$\leq \frac{\lceil 1/\varepsilon \rceil}{|P_f|} |P_g \setminus P_f| + \frac{|\Sigma|}{n_{s^*}} + 1 \tag{21}$$

$$\leq \lceil 1/\varepsilon \rceil \cdot (1/\varepsilon) + |\Sigma|^2/\varepsilon + 1 \tag{22}$$

$$= O(1/\varepsilon^2). \tag{23}$$

In particular, this implies that this construction can be completed by adding at most $O(1/\varepsilon^2)$ tokens to $z$, so in all cases the length of $z$ is $O(1/\varepsilon^2)$ as desired.

We now proceed to bound the approximation error. We first want to extend the bound in (18) to all of $A_g$. To this end, we have

$$\left| \sum_{s \in A_g} \frac{m_{s*}}{n_{s*}} n_s - \sum_{s \in A_g} m_s \right| \leq \left| \sum_{s \in A_g \setminus A_f} \frac{m_{s*}}{n_{s*}} n_s - \sum_{s \in A_g \setminus A_f} m_s \right| + \left| \sum_{s \in A_g \cap A_f} \frac{m_{s*}}{n_{s*}} n_s - \sum_{s \in A_g \cap A_f} m_s \right|$$

$$\leq 1 + \frac{m_{s*}}{n_{s*}} |P_g \cap P_f| - \frac{\lceil 1/\varepsilon \rceil}{|P_f|} |P_g \cap P_f| + 1 \tag{24}$$

$$\leq 2 + \left( \frac{\frac{\lceil 1/\varepsilon \rceil}{|P_f|} n_{s*} + 1}{n_{s*}} - \frac{\lceil 1/\varepsilon \rceil}{|P_f|} \right) |P_g \cap P_f| \tag{25}$$

$$\leq 2 + \frac{|P_g \cap P_f|}{n_{s*}}$$

$$\leq 2 + |\Sigma|. \tag{26}$$

Inequality (24) uses (18) and (5); (25) again uses (5); and (26) uses the fact that $n_{s*} \geq |P_f|/|\Sigma|$.

We can now bound the error of the *ratio* of $m_s$ over all of $A_g$. We have

$$\frac{m_s}{\sum_{s' \in A_g} m_{s'}} \geq \frac{\frac{m_{s*}}{n_{s*}} n_s - 1}{\frac{m_{s*}}{n_{s*}} \sum_{s' \in A_g} n_{s'} + 2 + |\Sigma|} \tag{27}$$

$$\geq \frac{n_s}{\sum_{s' \in A_g} n_{s'} + \frac{2 + |\Sigma|}{m_{s*}/n_{s*}}} - \frac{1}{\frac{m_{s*}}{n_{s*}} \sum_{s' \in A_g} n_{s'}}$$

$$\geq \frac{n_s}{\sum_{s' \in A_g} n_{s'}} - O\left( \frac{\frac{|\Sigma|}{m_{s*}/n_{s*}}}{\sum_{s' \in A_g} n_{s'}} \right) - \frac{1}{m_{s*}} \tag{28}$$

$$\geq \frac{n_s}{\sum_{s' \in A_g} n_{s'}} - O\left( \frac{|\Sigma|}{m_{s*}} \right)$$

$$\geq \frac{n_s}{\sum_{s' \in A_g} n_{s'}} - O(|\Sigma|^2 \varepsilon). \tag{29}$$

Inequality (27) uses (5) and (26); (28) uses $(\sum_{s' \in A_g} n_{s'})/n_{s*} \geq 1$; and (29) again uses (5) and $n_{s*} \geq |P_f|/|\Sigma|$ to conclude $m_{s*} = \Omega(|\Sigma|/\varepsilon)$. In a similar fashion, it can be shown that

$$\frac{m_s}{\sum_{s' \in A_g} m_{s'}} \leq \frac{n_s}{\sum_{s' \in A_g} n_{s'}} + O(|\Sigma|^2 \varepsilon).$$

Now we compare $g(x)$ and $g(z)$. As before, the effect of the $\tau$-prefix contributes at most $O(L_g^{\mathrm{MLP}} M_g^V \tau \varepsilon)$ to $\|g(x) - g(z)\|$, so we have

$$\|g(x) - g(z)\| \leq L_g^{\mathrm{MLP}} \sum_{s \in S_g} \left| \frac{m_s}{\sum_{s' \in S_g} m_{s'}} - \frac{n_s}{\sum_{s' \in S_g} n_{s'}} \right| M_g^V + O(L_g^{\mathrm{MLP}} M_g^V \tau \varepsilon)$$

$$= O\left( L_g^{\mathrm{MLP}} M_g^V (|\Sigma|^3 + \tau) \varepsilon \right).$$

Let $M_f = L_f^{\mathrm{MLP}} M_f^V$ and similarly for $g$. In every case, we have constructed $z$ such that $\|f(x) - f(z)\| = O(M_f(|\Sigma| + \tau)\varepsilon)$ and $\|g(x) - g(z)\| = O(M_g(|\Sigma|^3 + \tau)\varepsilon)$, and the length of $z$ is $O(1/\varepsilon^2)$. Making the crude bound $|\Sigma|^3 + \tau \leq |\Sigma|^3 \tau$ for convenience, we therefore have

$$\|f(x) - g(x)\| = O((M_f + M_g)|\Sigma|^3 \tau \varepsilon)$$

whenever $\|f(z) - g(z)\| \leq \varepsilon$ for all inputs $|z| \leq N$, and $N = O(1/\varepsilon^2)$. Thus, by substituting $\varepsilon \mapsto \frac{\varepsilon}{(M_f + M_g)|\Sigma|^3 \tau}$, we have $\|f(x) - g(x)\| = O(\varepsilon)$ for any string $x$ provided that $f$ and $g$ differ by at most $\varepsilon$ on inputs up to a length

$$N = O\left(\frac{(M_f + M_g)^2 |\Sigma|^6 \tau^2}{\varepsilon^2}\right).$$

**Including positional embedding vectors**  The setting with positional embedding vectors can be reduced to the general vocabulary case at the cost of increasing $|\Sigma| \to |\Sigma|\Delta$ and an additional factor of $\Delta$ by considering each possible (token, position mod $\Delta$) combination as its own token without positional embedding vectors. (This increases the vocabulary size from $|\Sigma|$ to $|\Sigma|\Delta$.) The construction without positional embedding vectors can then be used considering this expanded vocabulary; however, placing a "token" in the expanded vocabulary $\Sigma \times [\Delta]$ may require placing up to $\Delta$ true tokens to ensure that the positional embedding is correct. Thus, this construction requires at most an additional factor of $\Delta$ tokens. This gives a final bound

$$N = O\left(\frac{(M_f + M_g)^2 \Delta^7 |\Sigma|^6 \tau^2}{\varepsilon^2}\right).$$

As discussed in the beginning of the section, it will also be critical that $|z| \equiv |x| \pmod{\Delta}$; this can always be accomplished by padding $z$ with at most $\Delta$ additional tokens, which does not change the asymptotic length bound. $\qquad\square$

### A.3  PROOF OF THEOREM 4.2

**Lemma A.5.** *Let $x \in \Sigma^T$ and suppose that its constituent tokens $x_i$ are drawn i.i.d. from a categorical distribution, where $\mathbb{P}(x_i = s) = p_s$ for each $s \in \Sigma$. Then with probability at least $1 - \rho$, we have that*

$$\left|\frac{n(s, i, x)}{T/\Delta} - p_s\right| \leq \delta$$

*for all $(s, i)$ simultaneously provided that $T \geq \Delta \delta^{-2} \log \frac{2|\Sigma|\Delta}{\rho}$. We say that $x \in \mathrm{bulk}_T$ when the above inequality holds for all $(s, i) \in \Sigma \times [\Delta]$ simultaneously.*

*Proof.* Fix $i$ and consider the subset of positions $j \equiv i \pmod{\Delta}$ and let $T' = T/\Delta$ be the length of each of these subsequences. (We will ignore the fact that this may not be an integer as it is neither interesting nor important.) By Hoeffding's inequality, we have that $|n(s, i, x) - p_s T'| > c\sqrt{T'}$ with probability at most $2e^{-c^2}$. Setting $c = \sqrt{\log \frac{2|\Sigma|\Delta}{\rho}}$ and taking a union bound over $(s, i) \in \Sigma \times [\Delta]$, we see that

$$\left|\frac{n(s, x)}{T'} - p_s\right| \leq \sqrt{\frac{\log \frac{2|\Sigma|\Delta}{\rho}}{T'}}$$

with probability at least $1 - \rho$. Setting $\delta = \sqrt{\frac{\log \frac{2|\Sigma|\Delta}{\rho}}{T'}}$ and solving for $T = \Delta T'$ yields the desired result. $\qquad\square$

**Lemma A.6.** *Let $A(x) \subseteq \Sigma \times [\Delta]$ be the set of (token, position mod $\Delta$) pairs in the $\tau$-prefix attended to by $f$ when parsing the final token $x_T$. Furthermore, suppose $A(x) \neq \emptyset$, i.e., some tokens in the $\tau$-prefix enter hard attention. Define*

$$\tilde{f}^{\backslash \tau}(x) = \frac{\sum_{(s,i) \in A(x)} n(s, i, x) \boldsymbol{V}_f(\boldsymbol{E}_s^f + \boldsymbol{p}_i)}{\sum_{(s', i', x) \in A(x)} n(s', i', x)}$$

*to be the internal state of $f$ immediately after the attention layer, ignoring the $\tau$-suffix. Furthermore, define*

$$\bar{\tilde{f}}(x) = \frac{\sum_{(s,i) \in A(x)} p_s \boldsymbol{V}_f(\boldsymbol{E}_s^f + \boldsymbol{p}_i)}{\sum_{(s', i') \in A(x)} p_{s'}}.$$

*Finally suppose that $\min_{s \in \Sigma} p_s \geq \gamma$ and that $\delta$ is small enough such that $|\Sigma|\Delta\delta/\gamma \leq 1/2$. Then for any $x \in \text{bulk}_T$, we have that*

$$\|\tilde{f}^{\backslash \tau}(x) - \bar{\bar{f}}(x)\| \leq \frac{3M_f^V(|\Sigma|\Delta)^2\delta}{\gamma},$$

*where $M_f^V = \max_{s \in \Sigma, i \in [\Delta]} \|\boldsymbol{V}_f(\boldsymbol{E}_s^f + \boldsymbol{p}_i)\|$.*

*Proof.* Let $A = A(x)$. Observe that

$$\|\tilde{f}^{\backslash \tau}(x) - \bar{\bar{f}}(x)\| \leq \sum_{(s,i) \in A} \left| \frac{n(s,i,x)}{\sum_{(s',i') \in A} n(s',i',x)} - \frac{p_s}{\sum_{(s',i') \in A} p_{s'}} \right| M_f^V. \tag{30}$$

We will proceed by bounding the terms in this summation. Let $T' = T/\Delta$. Observe that for $x \in \text{bulk}_T$, we have the following:

$$\frac{n(s,i,x)}{\sum_{(s',i') \in A} n(s',i',x)} \leq \frac{(p_s + \delta)T'}{\sum_{(s',i') \in A}(p_{s'} - \delta)T'}$$

$$\leq \left( \frac{p_s}{\sum_{(s',i') \in A} p_{s'}} + \frac{\delta}{\sum_{(s',i') \in A} p_{s'}} \right) \frac{\sum_{(s',i') \in A} p_{s'}}{\sum_{(s',i') \in A} p_{s'} - \delta|\Sigma|\Delta} \tag{31}$$

$$\leq \left( \frac{p_s}{\sum_{(s',i') \in A} p_{s'}} + \frac{\delta}{\gamma} \right) \left( 1 + \frac{2|\Sigma|\Delta\delta}{\gamma} \right) \tag{32}$$

$$\leq \frac{p_s}{\sum_{(s',i') \in A} p_{s'}} + \frac{3|\Sigma|\Delta\delta}{\gamma}.$$

Inequality (31) holds because $|A| \leq |\Sigma|\Delta$. Inequality (32) holds because $\sum_{s' \in A} p_{s'} \geq \gamma$ (since $A$ is nonempty and all $p_{s'} \geq \gamma$) and $1/(1 - |\Sigma|\Delta\delta/\gamma) \leq 1 + 2|\Sigma|\Delta\delta/\gamma$ when $|\Sigma|\Delta\delta/\gamma \leq 1/2$. A similar argument with $\delta \mapsto -\delta$ and the inequalities reversed also shows that

$$\frac{n(s,x)}{\sum_{(s',i') \in A} n(s',i',x)} \geq \frac{p_s}{\sum_{(s',i') \in A} p_{s'}} - \frac{3|\Sigma|\Delta\delta}{\gamma}.$$

We can therefore bound the terms in (30) and we obtain

$$\|\tilde{f}^{\backslash \tau}(x) - \bar{\bar{f}}(x)\| \leq \sum_{s \in A} \frac{3|\Sigma|\Delta\delta}{\gamma} M_f^V \leq \frac{3(|\Sigma|\Delta)^2 M_f^V \delta}{\gamma}$$

as desired. $\square$

**Lemma A.7.** *Let $A(x)$ be defined as in Lemma A.6 and again suppose $A(x) \neq \emptyset$. Let $A^\tau(x)$ be the set of (token, position) pairs attended to in the $\tau$-suffix. Define*

$$\tilde{f}(x) = \frac{\sum_{(s,i) \in A(x)} n(s,i,x)\boldsymbol{V}_f(\boldsymbol{E}_s^f + \boldsymbol{p}_i) + \sum_{(s,i) \in A^\tau} \boldsymbol{V}_f(\boldsymbol{E}_s^f + \boldsymbol{p}_i)}{\sum_{(s,i) \in A(x)} n(s,i,x) + |A^\tau|}$$

*to be the internal state of $f$ immediately after the attention layer, this time not ignoring the $\tau$-suffix. Then we have*

$$\|\tilde{f}(x) - \tilde{f}^{\backslash \tau}(x)\| \leq \frac{3\tau\Delta M_f^V}{(\gamma - \delta)T}$$

*provided that $x \in \text{bulk}_T$.*

*Proof.* We denote $A = A(x)$ and $A^\tau = A^\tau(x)$. Observe that we can write

$$\tilde{f}(x) = \tilde{f}^{\backslash \tau}(x) \cdot \frac{\sum_{(s,i) \in A} n(s,i,x)}{\sum_{(s,i) \in A} n(s,i,x) + |A^\tau|} + \frac{\sum_{(s,i) \in A^\tau} \boldsymbol{V}_f(\boldsymbol{E}_s^f + \boldsymbol{p}_i)}{\sum_{(s,i) \in A} n(s,i,x) + |A^\tau|}.$$

It therefore follows that

$$\|\tilde{f}(x) - \tilde{f}^{\backslash\tau}(x)\| \leq \left|1 - \frac{1}{1 + \frac{|A^\tau|}{\sum_{(s,i)\in A} n(s,i,x)}}\right| \|\tilde{f}^{\backslash\tau}(x)\| + \frac{\tau M_f^V}{\sum_{(s,i)\in A} n(s,i,x)} \tag{33}$$

$$\leq \frac{2\tau}{\sum_{(s,i)\in A} n(s,i,x)} \cdot M_f^V + \frac{\tau M_f^V}{\sum_{(s,i)\in A} n(s,i,x)} \tag{34}$$

$$\leq \frac{3\tau M_f^V}{(\gamma - \delta)T/\Delta}. \tag{35}$$

Inequalities (33) and (34) both use the fact that $|A^\tau| \leq \tau$ and $\|\boldsymbol{V}_f(\boldsymbol{E}_s^f + \boldsymbol{p}_i)\| \leq M_f^V$. Inequality (34) additionally uses that $1/(1 + |A^\tau|/\sum_{(s,i)\in A} n(s,i,x)) \geq 1 - 2|A^\tau|/\sum_{(s,i)\in A} n(s,i,x)$ provided that $|A^\tau|/\sum_{(s,i)\in A} n(s,i,x) \leq 1/2$. The final inequality (35) uses the fact that $A \neq \emptyset$; that $x \in \text{bulk}_T$ so $n(s,i,x) \geq (p_s - \delta)T/\Delta$; and that $p_s \geq \gamma$. $\qquad\square$

**Lemma A.8.** *Suppose that $x \in \text{bulk}_T$, $z \in \text{bulk}_N$, $x_{T-\tau+1:T} = z_{N-\tau+1:N}$ and $N \equiv T \pmod{\Delta}$ with $N \leq T$, and $\min_{s\in\Sigma} p_s \geq \gamma$. Then*

$$\|f(x) - f(z)\| \leq 6M_f^V L_f^{\text{MLP}} \left(\frac{(|\Sigma|\Delta)^2\delta}{\gamma} + \frac{\tau\Delta}{(\gamma - \delta)N}\right).$$

*Here, $L_f^{\text{MLP}}$ is the bound on the MLP Lipschitz constant from Lemma A.2*

*Proof.* Observe that since $x$ and $z$ share a common $\tau$-suffix (and therefore a common final token) as well as a common positional embedding vector on the final token, $A(x) = A(z)$ and $A^\tau(x) = A^\tau(z)$. We may now consider two cases. If $A(x) = A(z) = \emptyset$, then $f(x) = f(z)$ exactly (all of the calculations are performed on the shared $\tau$-suffix) and the desired inequality holds trivially.

Otherwise, we may assume that $A(x) = A(z) \neq \emptyset$. We may then apply Lemmas A.6 and A.7. We have

$$\|\tilde{f}(x) - \bar{\bar{f}}(x)\| \leq \|\tilde{f}(x) - \tilde{f}^{\backslash\tau}(x)\| + \|\tilde{f}^{\backslash\tau}(x) - \bar{\bar{f}}(x)\| \leq \frac{3\tau\Delta M_f^V}{(\gamma - \delta)T} + \frac{3M_f^V(|\Sigma|\Delta)^2\delta}{\gamma}. \tag{36}$$

The analogous inequality holds for $z$ with $T$ replaced by $N$. Since $\bar{\bar{f}}(x)$ depends on $x$ only via $A(x)$, we have $\bar{\bar{f}}(x) = \bar{\bar{f}}(z)$. Thus we can again apply the triangle inequality to write $\|\tilde{f}(x) - \tilde{f}(z)\| \leq \|\tilde{f}(x) - \bar{\bar{f}}(x)\| + \|\tilde{f}(z) - \bar{\bar{f}}(z)\|$. Applying inequality (36) to each of these terms and using the fact that $N \leq T$, we have

$$\|\tilde{f}(x) - \tilde{f}(z)\| \leq 6M_f^V \left(\frac{\tau\Delta}{(\gamma - \delta)N} + \frac{(|\Sigma|\Delta)^2\delta}{\gamma}\right).$$

We can then directly apply Lemma A.2 to obtain the final result. $\qquad\square$

**Lemma A.9.** *Let $\{p_s\}_{s\in\Sigma} \sim \text{Dirichlet}((\alpha_s)_{s\in\Sigma})$ be drawn from a Dirichlet distribution with parameters $\alpha_s$. Define $\alpha^* = \sum_{s\in\Sigma} \alpha_s$ and $\alpha_0 = \min_{s\in\Sigma} \alpha_s$. Then we have*

$$\mathbb{P}(\exists s \in \Sigma : p_s < \gamma) \leq \frac{2|\Sigma|}{\alpha_0} 4^{\alpha^*} \gamma^{\alpha_0}.$$

*Proof.* Rather than dealing with the more complex joint distribution of the $p_s$, we will bound the marginals and apply a union bound. The marginals of the Dirichlet distribution are $p_s \sim \text{Beta}(\alpha_s, \alpha^* - \alpha_s)$, so it suffices to provide a lower tail bound for the beta distribution.

Let $x \sim \text{Beta}(\alpha, \beta)$, so $x$ has density $f(x) = \frac{1}{B(\alpha,\beta)} x^{\alpha-1}(1 - x)^{\beta-1}$, where $B(\alpha, \beta) = \int_0^1 x^{\alpha-1}(1-x)^{\beta-1} dx$ is the beta function. We first give a lower bound on $B(\alpha, \beta)$. When $\alpha, \beta > 1$,

we have

$$
\begin{aligned}
B(\alpha, \beta) &\geq \int_{1/4}^{3/4} x^{\alpha-1}(1-x)^{\beta-1}\, dx \\
&\geq \frac{1}{2} \cdot \left(\frac{1}{4}\right)^{\alpha-1}\left(\frac{1}{4}\right)^{\beta-1} \\
&= \frac{1}{2^{2\alpha+2\beta-3}} \\
&\geq \frac{1}{4^{\alpha+\beta}}.
\end{aligned}
$$

When $\alpha \leq 1$, we have

$$
B(\alpha, \beta) \geq \int_0^1 (1-x)^{\beta-1}\, dx = \frac{1}{\beta}.
$$

Similarly, when $\beta \leq 1$, we have

$$
B(\alpha, \beta) \geq \int_0^1 x^{\alpha-1}\, dx = \frac{1}{\alpha}.
$$

In particular, since $4^{\alpha+\beta} \geq \alpha, \beta$ for $\alpha, \beta > 0$, we have that

$$
B(\alpha, \beta) \geq \min\{\alpha^{-1}, \beta^{-1}, 4^{-(\alpha+\beta)}\} = 4^{-(\alpha+\beta)}.
$$

With this inequality, we can now establish the following bound for $t \leq 1/2$:

$$
\begin{aligned}
\mathbb{P}(x \leq t) &= \frac{1}{B(\alpha, \beta)} \int_0^t x^{\alpha-1}(1-x)^{\beta-1}\, dx \\
&\leq \frac{1}{B(\alpha, \beta)} \int_0^t x^{\alpha-1}(1-x)^{-1}\, dx \\
&\leq \frac{1}{B(\alpha, \beta)} \int_0^t x^{\alpha-1}(1-t)^{-1}\, dx \\
&= \frac{t^{\alpha}}{\alpha(1-t)B(\alpha, \beta)} \\
&\leq \frac{2}{\alpha} \cdot 4^{\alpha+\beta} t^{\alpha}.
\end{aligned}
$$

Now that we have established the tail bound for a general beta distribution, we can return to the original goal of bounding the Dirichlet. The marginal beta distribution for each $p_s$ has $\alpha = \alpha_s$ and $\beta = \alpha^* - \alpha_s$. Thus, by a union bound, we have

$$
\begin{aligned}
\mathbb{P}(\exists s \,:\, p_s < \gamma) &\leq \sum_{s \in \Sigma} \mathbb{P}(p_s < \gamma) \\
&\leq \sum_{s \in \Sigma} \frac{2}{\alpha_s} \cdot 4^{\alpha_s + \alpha^* - \alpha_s} \gamma^{\alpha_s} \\
&\leq \frac{2|\Sigma|}{\alpha_0} 4^{\alpha^*} \gamma^{\alpha_0},
\end{aligned}
$$

as desired. $\qquad\square$

**Lemma A.10.** *Suppose that $T \geq N \geq \Delta\delta^{-2}\log\frac{2|\Sigma|\Delta}{\rho}$ and $\min_s p_s \geq \gamma$. Then we have*

$$
\|f - g\|_{T, \mathcal{P}} = O\left((M_f + M_g)\left(\rho + \frac{(|\Sigma|\Delta)^2\delta}{\gamma} + \frac{\tau\Delta}{(\gamma-\delta)N}\right) + \|f - g\|_{N', \mathcal{P}}\right).
$$

*Proof.* Given an integer $N$, define the simulation map $\mathrm{sim}_N(x) = x_{T-N'+1:T} =$ the last $N'$ tokens of $x$, where $N \leq N' < N + \Delta$ is chosen such that $N' \equiv T \pmod{\Delta}$. Note that using this definition

and for $N \geq \tau$, we have $x_{T-\tau+1:T} = (\mathrm{sim}_N(x))_{N-\tau+1:\tau}$, i.e., the $\tau$-suffixes coincide. Furthermore, by definition, $N' = |\mathrm{sim}_N(x)| \equiv T \pmod{\Delta}$ and the final tokens match, so $A(\mathrm{sim}_N(x)) = A(x)$.

By Lemma A.5 and a union bound, $x \in \mathrm{bulk}_T$ and $\mathrm{sim}_N(x) \in \mathrm{bulk}_{N'}$ simultaneously with probability at least $1 - 2\rho$ provided that $T \geq N \geq \Delta\delta^{-2}\log\frac{2|\Sigma|\Delta}{\rho}$. The token independence means that $\sum_{|x|=T, \mathrm{sim}_N(x)=z} \mathbb{P}(x) = \mathbb{P}(z)$, so we have

$$\sum_{|x|=T} \mathbb{P}(x)\|f(x) - g(x)\| \leq \sum_{\substack{|x|=T \\ x \notin \mathrm{bulk}_T \text{ or} \\ \mathrm{sim}_N(x) \notin \mathrm{bulk}_{N'}}} \mathbb{P}(x) \cdot (M_f + M_g) + \sum_{\substack{|x|=T \\ x \in \mathrm{bulk}_T \text{ and} \\ \mathrm{sim}_N(x) \in \mathrm{bulk}_{N'}}} \mathbb{P}(x)\|f(x) - g(x)\|$$

$$\leq 2\rho(M_f + M_g) + \sum_{z \in \mathrm{bulk}_{N'}} \left( \sum_{\substack{x \in \mathrm{bulk}_T \\ \mathrm{sim}_N(x)=z}} \mathbb{P}(x) \right) (\|f(z) - g(z)\| + \|f(x) - f(z)\| + \|g(x) - g(z)\|)$$

$$\leq 2\rho(M_f + M_g) + \sum_{z \in \mathrm{bulk}_{N'}} \mathbb{P}(z)(\|f(z) - g(z)\| + \|f(x) - f(z)\| + \|g(x) - g(z)\|)$$

$$\leq 2\rho(M_f + M_g) + \|f - g\|_{N',\mathcal{P}} + 6(M_f^V L_f^{\mathrm{MLP}} + M_g^V L_g^{\mathrm{MLP}}) \left( \frac{(|\Sigma|\Delta)^2\delta}{\gamma} + \frac{\tau\Delta}{(\gamma - \delta)N} \right) \quad (37)$$

$$= O\left( (M_f + M_g)\left( \rho + \frac{(|\Sigma|\Delta)^2\delta}{\gamma} + \frac{\tau\Delta}{(\gamma - \delta)N} \right) + \|f - g\|_{N',\mathcal{P}} \right). \quad (38)$$

Inequality (37) follows from Lemma A.8. Inequality (38) uses the fact that $M_f^V L_f^{\mathrm{MLP}} \leq M_f$ and similarly for $g$. $\qquad\square$

We are now ready to prove Theorem 4.2, which we restate here for convenience.

**Theorem 4.2.** *For any probability distribution $\mathcal{P} = (p_s)_{s \in \Sigma}$ over the token vocabulary $\Sigma$, define*

$$\|f - g\|_{n,\mathcal{P}} = \sum_{|x|=n} \mathbb{P}_{\mathcal{P}}(x)\|f(x) - g(x)\|,$$

*where $\mathbb{P}_{\mathcal{P}}(x) = \prod_{i=1}^{|x|} p_{x_i}$ is the probability of the sequence $x$ when the tokens are drawn i.i.d. from $\mathcal{P}$. Let $\mathcal{P} = (p_s)_{s \in \Sigma} \sim \mathrm{Dir}((\alpha_s)_{s \in \Sigma})$ be drawn from a Dirichlet distribution, and define*

$$\|f - g\|_n = \mathbb{E}_{\mathcal{P} \sim \mathrm{Dir}((\alpha_s)_{s \in \Sigma})}[\|f - g\|_{n,\mathcal{P}}].$$

*Let $\alpha_0 = \min_{s \in \Sigma} \alpha_s$. Then there exists*

$$N_0 = O\left( \max\left\{ 2^{p/\gamma}, \frac{16^{\frac{\alpha^*}{\alpha_0}} L^{2+2\alpha_0^{-1}}|\Sigma|^{4+2\alpha_0^{-1}}\Delta^5}{\alpha_0^{2\alpha_0^{-1}}\varepsilon^{2+2\alpha_0^{-1}}} \log\frac{|\Sigma|\Delta L}{\varepsilon} \right\} \right) = \widetilde{O}(\varepsilon^{-2-2\alpha_0^{-1}})$$

*such that if $\|f - g\|_N \leq \varepsilon$ for all $N \leq N_0$, we have that $\|f - g\|_T = O(\varepsilon^{1/2})$ for any $T$.*

*Proof.* By Markov's inequality, $\mathbb{E}_{\mathcal{P} \sim \mathrm{Dir}(\alpha_s)_{s \in \Sigma}}\|f - g\|_{N',\mathcal{P}} \leq \varepsilon$ implies that $\|f - g\|_{N',\mathcal{P}} > \eta$ with probability at most $\varepsilon/\eta$. When $\mathcal{P}$ is such that $\|f - g\|_{N',\mathcal{P}} > \eta$, we can use the bound $\|f - g\|_{T,\mathcal{P}} \leq M_f + M_g$.

By Lemma A.9, $\min_s p_s < \gamma$ with probability at most $\frac{2|\Sigma|}{\alpha_0} 4^{\alpha^*} \gamma^{\alpha_0}$. On this event, we can again bound $\|f(x) - g(x)\|_{T,\mathcal{P}} \leq M_f + M_g$.

Conditional on $\min_s p_s \geq \gamma$ and $\|f - g\|_{N',\mathcal{P}} \leq \eta$, we can use the bound from Lemma A.10.

Thus, by marginalizing $\mathcal{P}$ over the previous three cases, we have that

$$\mathbb{E}_{\{p_s\}}\|f - g\|_{1,T} = O\left( (M_f + M_g)\left( \frac{|\Sigma|}{\alpha_0} 4^{\alpha^*} \gamma^{\alpha_0} + \frac{\varepsilon}{\eta} + \rho + \frac{(|\Sigma|\Delta)^2\delta}{\gamma} + \frac{\tau\Delta}{(\gamma - \delta)N} \right) + \eta \right) \quad (39)$$

provided that $N \geq \Delta \delta^{-2} \log \frac{2|\Sigma|\Delta}{\rho}$. To make the entire bound $O(\varepsilon^{1/2})$, we choose the following:

$$\eta = \varepsilon^{1/2}, \qquad \rho = \frac{\varepsilon^{1/2}}{M_f + M_g}, \qquad \gamma = \left( \frac{\alpha_0 \varepsilon^{1/2}}{4^{\alpha^*} |\Sigma| (M_f + M_g)} \right)^{\alpha_0^{-1}},$$

$$\delta = \frac{\varepsilon^{1/2} \gamma}{(M_f + M_g)(|\Sigma|\Delta)^2} = \frac{\alpha_0^{\alpha_0^{-1}} \varepsilon^{\frac{1}{2}(1 + \alpha_0^{-1})}}{4^{\frac{\alpha^*}{\alpha_0}} (M_f + M_g)^{1 + \alpha_0^{-1}} |\Sigma|^{2 + \alpha_0^{-1}} \Delta^2}.$$

Note that with these settings, we indeed have $\gamma > \delta$ and furthermore $\tau\Delta/((\gamma - \delta)N) = O(\varepsilon^{1 + \alpha_0^{-1}/2}) = o(\varepsilon^{1/2})$. All other terms in (39) are $O(\varepsilon^{1/2})$. Thus, we arrive at an error of $O(\varepsilon^{1/2})$ with

$$N = O\left( \frac{16^{\frac{\alpha^*}{\alpha_0}} (M_f + M_g)^{2 + 2\alpha_0^{-1}} |\Sigma|^{4 + 2\alpha_0^{-1}} \Delta^5}{\alpha_0^{2\alpha_0^{-1}} \varepsilon^{1 + \alpha_0^{-1}}} \log \frac{|\Sigma|\Delta(M_f + M_g)}{\varepsilon} \right)$$

as desired. We make several remarks. First, we actually required that $\|f - g\|_{N'} \leq \varepsilon$, but since $N' < N + \Delta$ this does not change the final asymptotic bound on sequence length. Second, it is interesting that up to leading order terms in $\varepsilon^{-1}$, $\tau$ does not enter the bound.

A final remark on the proof is that if we strengthen the assumption to $\|f - g\|_{N', \mathcal{P}} \leq \varepsilon$ *conditionally* on $\mathcal{P}$ with $\min_s p_s > \gamma$, the resulting error can scale as $\varepsilon$ rather than $\varepsilon^{1/2}$, albeit with a larger required $N_0$. In this case, (39) becomes

$$\mathbb{E}_{\{p_s\}} \|f - g\|_{1,T} = O\left( (M_f + M_g) \left( \frac{|\Sigma|}{\alpha_0} 4^{\alpha^*} \gamma^{\alpha_0} + \rho + \frac{(|\Sigma|\Delta)^2 \delta}{\gamma} + \frac{\tau\Delta}{(\gamma - \delta)N} \right) + \varepsilon \right). \quad (40)$$

Setting $\rho$, $\gamma$, and $\delta$ according to

$$\rho = \frac{\varepsilon}{M_f + M_g}, \qquad \gamma = \left( \frac{\varepsilon}{4^{\alpha^*} \frac{|\Sigma|}{\alpha_0} (M_f + M_g)} \right)^{\alpha_0^{-1}},$$

$$\delta = \frac{\varepsilon \gamma}{(M_f + M_g)(|\Sigma|\Delta)^2} = \frac{\alpha_0^{\alpha_0^{-1}} \varepsilon^{1 + \alpha_0^{-1}}}{4^{\frac{\alpha^*}{\alpha_0}} (M_f + M_g)^{1 + \alpha_0^{-1}} |\Sigma|^{2 + \alpha_0^{-1}} \Delta^2},$$

inequality (40) is $O(\varepsilon)$ with

$$N = O\left( \frac{16^{\frac{\alpha^*}{\alpha_0}} (M_f + M_g)^{2 + 2\alpha_0^{-1}} |\Sigma|^{4 + 2\alpha_0^{-1}} \Delta^5}{\alpha_0^{2\alpha_0^{-1}} \varepsilon^{2 + 2\alpha_0^{-1}}} \log \frac{|\Sigma|\Delta(M_f + M_g)}{\varepsilon} \right)$$

$$= O(\varepsilon^{-(2 + 2\alpha_0^{-1})} \log \varepsilon^{-1}).$$

$\square$

# B  OMITTED PROOFS FROM SECTION 5

**Notation.** We will assume WLOG that $\Sigma = [S]$. For a string $x \in [S]^{|x|}$, define $\mu(x)$ to be the empirical frequencies of the tokens in $x$, i.e $\mu(x) := \frac{1}{|x|} \sum_{i=1}^{|x|} \mathbf{e}_{x_i}$, where $\mathbf{e}_j \in \mathbb{R}^S$ is the $j$th standard basis element. Moreover, let $x_{\leq i}$ denote the substring of $x$ containing the first $i$ tokens, and for a set $\mathcal{A}$, $x_{\mathcal{A}}$ the substring of $x$ containing only those indices in $\mathcal{A}$. Finally, for integers $a < b$, define $[a : b]$ to be the set of integers $\{a, a+1, \ldots, b-1, b\}$.

## B.1  PROOF OF KEY SIMULATION LEMMA

In this section, we prove Lemma 5.3, which we restate below for convenience.

**Lemma 5.3.** *Let $p : [S]^{\tau+1} \times \Delta^S \to \mathbb{R}^m$ be a fixed function, which is L Lipschitz in its second argument and uniformly bounded by G. Then, there exists a subset $\mathcal{I} \subset [T]$ such that, if $z = x_{\mathcal{I}}$, then $||\mathcal{I}| - n| \le \tau + 1 + n^{1/3}$ and*

$$\left\| \frac{1}{T} \sum_{t=1}^{T} p(x_{t-\tau:t}, \mu(x_{\le t})) - \frac{1}{|z|} \sum_{t=1}^{|z|} p(z_{t-\tau:t}, \mu(z_{\le t})) \right\| \lesssim \frac{(G+L)(\tau+1)}{n^{1/3}}.$$

*Proof.* Our proof proceeds via the probabilistic method. Let us sample $\mathcal{I}$ as follows. Let $p = n/T$, and let $q \in (0, 1)$ be a parameter to be chosen later. Let us define a Markov chain $j_1, \dots, j_T$ on the state space $\{0, 1\}$, with the following transition probabilities:

$$\mathbb{P}(j_{t+1} = 0 \mid j_t = 0) = 1 - r, \quad \mathbb{P}(j_{t+1} = 1 \mid j_t = 0) = r$$
$$\mathbb{P}(j_{t+1} = 0 \mid j_t = 1) = q, \qquad \mathbb{P}(j_{t+1} = 1 \mid j_t = 1) = 1 - q$$

Letting $r := \frac{pq}{1-p}$, the stationary distribution is $\mathbb{P}(j_t = 1) = p$.

We will let the subset $\mathcal{I}$ be $\mathcal{I} := \{i \mid j_i = 1\} \cup [T - \tau : T]$

**Computing the variance of $|\mathcal{I}|$.** The first step is to compute the variance of $|\mathcal{I}|$. By definition, $\mathbb{E}|\mathcal{I}| = (T - \tau - 1) \cdot p + (\tau + 1) = n + (\tau + 1)(1 - p)$.

Since the $k$-step transition kernel satisfies

$$\mathbb{P}[j_{t+k} = 1 \mid j_t = 1] = (1 - q - r)^k (1 - p) + p,$$

we have that

$$\mathbb{E}(|\mathcal{I}|)^2 = \mathbb{E} \left( \sum_{t=1}^{T} j_t \right)^2$$

$$= \sum_{i,i'=1}^{T} \mathbb{E}[j_i j_{i'}]$$

$$= (\tau + 1)^2 + 2(\tau + 1)(T - \tau - 1)p + \sum_{i,i'=1}^{T-\tau-1} \left( (1 - q - r)^{|i-i'|}(1 - p)p + p^2 \right)$$

$$\le (\tau + 1)^2 + 2(\tau + 1)(T - \tau - 1)p + (T + \tau - 1)^2 p^2 + 2(T - \tau - 1)(1 - p)p \sum_{i=0}^{\infty} (1 - q - r)^i$$

$$\le (\mathbb{E}|\mathcal{I}|)^2 + 2T \frac{(1-p)p}{q+r}$$

$$\le (\mathbb{E}|\mathcal{I}|)^2 + \frac{2Tp}{q}$$

$$= (\mathbb{E}|\mathcal{I}|)^2 + \frac{2n}{q}.$$

Therefore $\mathbb{E}(|\mathcal{I}| - \mathbb{E}|\mathcal{I}|)^2 \le 2n/q$.

**Decomposing the original expression.** Next, we bound the quantity

$$\mathbb{E} \left\| \frac{1}{T} \sum_{t=1}^{T} p(x_{t-\tau:t}, \mu(x_{\le t})) - \frac{1}{n} \sum_{t=1}^{|z|} p(z_{t-\tau:t}, \mu(z_{\le t})) \right\|.$$

Define $\mathcal{I}_{gap}$ to be the set of indices in $\mathcal{I}$ such that some index in $\{i - \tau, \dots, i - 1\}$ is not in $\mathcal{I}$, i.e

$$\mathcal{I}_{gap} = \{i \in \mathcal{I} \mid \exists t \in [\tau] : i - t \notin \mathcal{I}\}.$$

We can write, denoting $\mathcal{I} = \{i_1, \ldots, i_{\mathcal{I}}\}$ with $i_1 < i_2 < \cdots < i_{|\mathcal{I}|}$,

$$\frac{1}{n}\sum_{t=1}^{|z|} p(z_{t-\tau:t}, \mu(z_{\leq t})) = \frac{1}{n}\sum_{t=1}^{|z|} p(z_{t-\tau:t}, \mu(z_{\leq t})) \cdot \mathbf{1}(i_t \in I_{gap}) + \frac{1}{n}\sum_{t=1}^{|z|} p(z_{t-\tau:t}, \mu(z_{\leq t})) \cdot \mathbf{1}(i_t \notin I_{gap})$$

$$= \frac{1}{n}\sum_{t=1}^{|z|} p(z_{t-\tau:t}, \mu(z_{\leq t})) \cdot \mathbf{1}(i_t \in I_{gap}) + \frac{1}{n}\sum_{t=1}^{|z|} p(x_{i_t-\tau:i_t}, \mu(z_{\leq t})) \cdot \mathbf{1}(i_t \notin I_{gap})$$

$$= \frac{1}{n}\sum_{t=1}^{|z|} p(z_{t-\tau:t}, \mu(z_{\leq t})) \cdot \mathbf{1}(i_t \in I_{gap}) + \frac{1}{n}\sum_{t=1}^{T} p(x_{t-\tau:t}, \mu(x_{[t]\cap\mathcal{I}})) \cdot \mathbf{1}([t-\tau:t] \subset \mathcal{I}).$$

Therefore we can decompose

$$\mathbb{E}\left\|\frac{1}{T}\sum_{t=1}^{T} p(x_{t-\tau:t}, \mu(x_{\leq t})) - \frac{1}{n}\sum_{t=1}^{|z|} p(z_{t-\tau:t}, \mu(z_{\leq t}))\right\|$$

$$\leq \underbrace{\mathbb{E}\left\|\frac{1}{T}\sum_{t=1}^{T} p(x_{t-\tau:t}, \mu(x_{\leq t})) - \frac{1}{n}\sum_{t=1}^{T} p(x_{t-\tau:t}, \mu(x_{\leq t})) \cdot \mathbf{1}([t-\tau:t] \subset \mathcal{I})\right\|}_{\text{(I)}}$$

$$+ \underbrace{\mathbb{E}\left\|\frac{1}{n}\sum_{t=1}^{T} p(x_{t-\tau:t}, \mu(x_{\leq t})) \cdot \mathbf{1}([t-\tau:t] \subset \mathcal{I}) - \frac{1}{n}\sum_{t=1}^{T} p(x_{t-\tau:t}, \mu(x_{[t]\cap\mathcal{I}})) \cdot \mathbf{1}([t-\tau:t] \subset \mathcal{I})\right\|}_{\text{(II)}}$$

$$+ \underbrace{\frac{G\mathbb{E}|\mathcal{I}_{gap}|}{n}}_{\text{(III)}}.$$

**Bounding (I):** Let us begin by defining the random variable

$$Z_t = p(x_{t-\tau:t}, \mu(x_{\leq t}))(1 - \frac{T}{n}\mathbf{1}([t-\tau:t] \subset \mathcal{I})).$$

The first term is then

$$\text{(I)} = \frac{1}{T}\mathbb{E}\left\|\sum_{i=1}^{T} Z_i\right\|$$

$$\leq \frac{1}{T}\sum_{i=1}^{\tau}\|Z_i\| + \frac{1}{T}\sum_{i=T-\tau}^{T}\|Z_i\| + \frac{1}{T}\mathbb{E}\left\|\sum_{i=1}^{T-\tau-1} Z_i\right\|$$

$$\leq \frac{G(2\tau + 1)}{n} + \frac{1}{T}\left(\mathbb{E}\left\|\sum_{\tau+1}^{T-\tau-1} Z_i\right\|^2\right)^{1/2}$$

$$= \frac{G(2\tau + 1)}{n} + \frac{1}{T}\left(\mathbb{E}\sum_{t=\tau+1}^{T-\tau-1}\|Z_i\|^2 + \sum_{i \neq j}\mathbb{E}\langle Z_i, Z_j\rangle\right)^{1/2}$$

First, see that

$$\mathbb{E}\|Z_t\|^2 \leq G^2\mathbb{E}\left[\left(1 - \frac{T}{n}\mathbf{1}([t-\tau:t] \subset \mathcal{I})\right)^2\right] \leq G^2\left(1 - \frac{2T}{n}p(1-q)^\tau + \frac{T^2}{n^2}p(1-q)^\tau\right)$$

$$\leq \frac{G^2T^2p}{n^2}$$

$$= \frac{G^2T}{n}.$$

Next, we have that

$$|\mathbb{E}\langle Z_i, Z_j\rangle| \le G^2\left|\mathbb{E}\left[\left(1 - \frac{T}{n}\mathbf{1}([i-\tau:i]\subset\mathcal{I})\right)\left(1 - \frac{T}{n}\mathbf{1}([j-\tau:j]\subset\mathcal{I})\right)\right]\right|$$

$$= G^2\left|1 - \frac{2T}{n}\cdot p(1-q)^\tau + \frac{T^2}{n^2}\mathbb{P}([i-\tau:i]\subset\mathcal{I}, [j-\tau:j]\subset\mathcal{I})\right|$$

$$= G^2\left|1 - 2(1-q)^\tau + p^{-2}\cdot\mathbb{P}([i-\tau:i]\subset\mathcal{I}, [j-\tau:j]\subset\mathcal{I})\right|$$

Let's assume that $i < j$. First, consider the case where $j \ge i + \tau$. Then

$$\mathbb{P}([i-\tau:i]\subset\mathcal{I}, [j-\tau:j]\subset\mathcal{I}) = p(1-q)^\tau\cdot\mathbb{P}(j-\tau\in\mathcal{I}\mid i\in\mathcal{I})\cdot(1-q)^\tau$$

$$= p(1-q)^{2\tau}\left(p + (1-p)(1-q-r)^{j-i-\tau}\right),$$

and thus

$$|\mathbb{E}\langle Z_i, Z_j\rangle| \le G^2\left|1 - 2(1-q)^\tau + (1-q)^{2\tau} + p^{-1}(1-p)(1-q)^{2\tau}(1-q-r)^{j-i-\tau}\right|$$

$$\le G^2\left((1-(1-q)^\tau)^2 + p^{-1}(1-q)^{j-i+\tau}\right)$$

$$\le G^2\left(\tau^2 q^2 + p^{-1}(1-q)^{j-i+\tau}\right)$$

Next, for $j < i + \tau$, we have that

$$\mathbb{P}([i-\tau:i]\subset\mathcal{I}, [j-\tau:j]\subset\mathcal{I}) = \mathbb{P}([i-\tau:j]\subset\mathcal{I})$$

$$= p(1-q)^{j-i+\tau},$$

and thus

$$|\mathbb{E}\langle Z_i, Z_j\rangle| \le G^2\left|1 - 2(1-q)^\tau + p^{-1}(1-q)^{j-i+\tau}\right|$$

$$\le G^2 p^{-1}(1-q)^{j-i+\tau}.$$

Altogether, **(I)** can be bounded as

$$\textbf{(I)} \le \frac{G(2\tau+1)}{n} + \frac{1}{T}\left(\frac{G^2 T^2}{n} + T^2 G^2\tau^2 q^2 + 2G^2 p^{-1}T\sum_{k>0}(1-q)^{k+\tau}\right)^{1/2}$$

$$\le \frac{G(2\tau+1)}{n} + \frac{1}{T}\left(\frac{G^2 T^2}{n} + T^2 G^2\tau^2 q^2 + 2G^2 T^2 n^{-1}q^{-1}\right)^{1/2}$$

$$\lesssim \frac{G(\tau+1)}{n} + \frac{G}{\sqrt{n}} + G\tau q + \frac{G}{\sqrt{nq}}.$$

**Bounding (II):** Let's next consider the **(II)** term. Since $p$ is $L$-Lipschitz in its second argument, we have that

$$\textbf{(II)} = \mathbb{E}\left\|\frac{1}{n}\sum_{t=1}^T p(x_{t-\tau:t}, \mu(x_{\le t}))\cdot\mathbf{1}([t-\tau:t]\subset\mathcal{I}) - \frac{1}{n}\sum_{t=1}^T p(x_{t-\tau:t}, \mu(x_{[t]\cap\mathcal{I}}))\cdot\mathbf{1}([t-\tau:t]\subset\mathcal{I})\right\|$$

$$\le \frac{L}{n}\sum_{t=1}^T\mathbb{E}\left[\left\|\mu(x_{\le t}) - \mu(x_{[t]\cap\mathcal{I}})\right\|\cdot\mathbf{1}([t-\tau:t]\subset\mathcal{I})\right]$$

Let's compute the $t$th term in this sum, for $t \in [\tau+1, T-\tau-1]$ (for $t \le \tau$, the quantity is trivially zero, and for $t \ge T - \tau$ we can bound it by $O(1)$). We have that

$$\mathbb{E}\left[\left\|\mu(x_{\le t}) - \mu(x_{[t]\cap\mathcal{I}})\right\|\cdot\mathbf{1}([t-\tau:t]\subset\mathcal{I})\right]$$

$$= \mathbb{E}\left[\left\|\mu(x_{\le t}) - \frac{\sum_{i=1}^t e_{x_i}\mathbf{1}(i\in\mathcal{I})}{|\mathcal{I}\cap[t]|}\right\|\cdot\mathbf{1}([t-\tau:t]\subset\mathcal{I})\right]$$

$$= p(1-q)^\tau\mathbb{E}\left[\left\|\frac{\mu(x_{\le t})\cdot|\mathcal{I}\cap[t]| - \sum_{i=1}^t e_{x_i}\mathbf{1}(i\in\mathcal{I})}{|\mathcal{I}\cap[t]|}\right\| \mid \mathbf{1}([t-\tau:t]\subset\mathcal{I})\right].$$

The denominator is $|\mathcal{I} \cap [t]| = \sum_{i=1}^{t} \mathbf{1}(i \in \mathcal{I})$. We first bound its conditional expectation:

$$\mathbb{E}[|\mathcal{I} \cap [t]| \mid \mathbf{1}([t - \tau : t] \subset \mathcal{I})] = \tau + 1 + \sum_{i=1}^{t-\tau-1} \mathbb{P}(i \in \mathcal{I} \mid t - \tau \in \mathcal{I})$$

$$= \tau + 1 + (t - \tau - 1)p + (1 - p) \sum_{i=1}^{t-\tau-1} (1 - q - r)^i$$

$$\geq pt.$$

Next, we can bound the conditional variance of the denominator:

$$\mathrm{Var}(|\mathcal{I} \cap [t]| \mid \mathbf{1}([t - \tau : t] \subset \mathcal{I}))$$
$$= \mathrm{Var}(|\mathcal{I} \cap [t - \tau - 1]| \mid \mathbf{1}([t - \tau : t] \subset \mathcal{I}))$$
$$= \sum_{i,j=1}^{t-\tau-1} \mathrm{Cov}(i \in \mathcal{I}, j \in \mathcal{I} \mid t - \tau \in \mathcal{I})$$
$$= \sum_{i,j=1}^{t-\tau-1} \mathbb{P}(j \in \mathcal{I} \mid i \in \mathcal{I})\mathbb{P}(i \in \mathcal{I} \mid t - \tau \in \mathcal{I}) - \mathbb{P}(j \in \mathcal{I} \mid t - \tau \in \mathcal{I})\mathbb{P}(i \in \mathcal{I} \mid t - \tau \in \mathcal{I})$$
$$= \sum_{i,j=1}^{t-\tau-1} (1 - p)\big((1 - q - r)^{i-j} - (1 - q - r)^{t-\tau-j}\big)\big((1 - q - r)^{t-\tau-i}(1 - p) + p\big)$$
$$\leq \sum_{i,j=1}^{t-\tau-1} (1 - q)^{i-j}\big(p + (1 - p)(1 - q)^{t-\tau-i}\big)$$
$$= \sum_{i=1}^{t-\tau-1} \big(p + (1 - p)(1 - q)^{t-\tau-i}\big) \sum_{j=1}^{i}(1 - q)^{i-j}$$
$$\leq q^{-1} \sum_{i=1}^{t-\tau-1} \big(p + (1 - p)(1 - q)^{t-\tau-i}\big)$$
$$\leq pq^{-1}(t - \tau - 1) + q^{-2}$$

Altogether, by Chebyshev's inequality, we can upper bound the conditional probability that the denominator is too small:

$$\mathbb{P}(|\mathcal{I} \cap [t]| \leq \tfrac{1}{2}pt \mid t - \tau \in \mathcal{I}) \leq \mathbb{P}(|\mathcal{I} \cap [t]| \leq \tfrac{1}{2}\mathbb{E}[|\mathcal{I} \cap [t]| \mid \mathbf{1}([t - \tau : t] \subset \mathcal{I})] \mid [t - \tau : t] \subset \mathcal{I})$$

$$\leq \frac{4\mathrm{Var}(|\mathcal{I} \cap [t]| \mid \mathbf{1}([t - \tau : t] \subset \mathcal{I}))}{\mathbb{E}[|\mathcal{I} \cap [t]| \mid \mathbf{1}([t - \tau : t] \subset \mathcal{I})]^2}$$

$$\leq \frac{pq^{-1}t + q^{-2}}{p^2 t^2}$$

$$= \frac{1}{pqt} + \frac{1}{p^2 q^2 t^2}$$

$$\lesssim \frac{1}{pqt} \wedge 1,$$

where the last inequality follows from the fact that the probability must be bounded by 1.

Altogether, the $t$th term in the sum is

$$\mathbb{E}\big[\big\|\mu(x_{\leq t}) - \mu(x_{[t] \cap \mathcal{I}})\big\| \cdot \mathbf{1}([t - \tau : t] \subset \mathcal{I})\big]$$

$$\lesssim \frac{\mathbb{E}\big[\big\|\mu(x_{\leq t}) \cdot |\mathcal{I} \cap [t]| - \sum_{i=1}^{t} e_{x_i}\mathbf{1}(i \in \mathcal{I})\big\| \mid \mathbf{1}([t - \tau : t] \subset \mathcal{I})\big]}{t} + \left(\frac{1}{qt} \wedge p\right).$$

The numerator in the above expression can be written as

$$\mathbb{E}\left[\left\|\sum_{i=1}^{t}(\mathbf{1}(i \in \mathcal{I}) - p)(e_{x_i} - \mu(x_{\leq t}))\right\| \mid \mathbf{1}([t - \tau : t] \subset \mathcal{I})\right] = \mathbb{E}\left[\left\|\sum_{i=1}^{t} Z_i\right\| \mid \mathbf{1}([t - \tau : t] \subset \mathcal{I})\right],$$

where $Z_i := (\mathbf{1}(i \in \mathcal{I}) - p)(e_{x_i} - \mu(x_{\leq t}))$. For $i, j < t - \tau$, we have the bounds (assuming WLOG $j < i$)

$$|\mathbb{E}[\langle Z_i, Z_j \rangle \mid \mathbf{1}([t - \tau : t] \subset \mathcal{I})]|$$
$$\lesssim |\mathbb{E}[(\mathbf{1}(i \in \mathcal{I}) - p)(\mathbf{1}(j \in \mathcal{I}) - p) \mid \mathbf{1}([t - \tau : t] \subset \mathcal{I})]|$$
$$= \big((\mathbb{P}(j \in \mathcal{I} \mid i \in \mathcal{I}) - p)\mathbb{P}(i \in \mathcal{I} \mid t - \tau \in \mathcal{I}) - p\mathbb{P}(j \in \mathcal{I} \mid t - \tau \in \mathcal{I}) + p^2\big)$$
$$= \big((1 - p)(1 - q - r)^{i-j}\big((1 - q - r)^{t-\tau-i}(1 - p) + p\big) - p\big((1 - q - r)^{t-\tau-j}(1 - p) + p\big) + p^2\big)$$
$$= \big(p(1 - p)(1 - q - r)^{i-j} + (1 - p)^2(1 - q - r)^{t-\tau-j} - p(1 - p)(1 - q - r)^{t-\tau-j}\big)$$
$$\leq p(1 - q)^{i-j}.$$

Therefore,

$$\mathbb{E}\left[\left\|\sum_{i=1}^{t} Z_i\right\| \mid \mathbf{1}([t - \tau : t] \subset \mathcal{I})\right] \leq \tau + 1 + \mathbb{E}\left[\left\|\sum_{i=1}^{t-\tau-1} Z_i\right\|^2 \mid \mathbf{1}([t - \tau : t] \subset \mathcal{I})\right]^{1/2}$$

$$\leq \tau + 1 + \left(tp + p\sum_{i \neq j}(1 - q)^{i-j}\right)^{1/2}$$

$$\lesssim \tau + 1 + \sqrt{pt/q}.$$

Putting everything together, the $t$th term in the sum can be bounded by

$$\mathbb{E}\big[\|\mu(x_{\leq t}) - \mu(x_{[t] \cap \mathcal{I}})\| \cdot \mathbf{1}([t - \tau : t] \subset \mathcal{I})\big] \lesssim t^{-1}(\tau + 1) + t^{-1/2}p^{1/2}q^{-1/2} + \left(\frac{1}{qt} \wedge p\right)$$

$$\lesssim \left(t^{-1}(\tau + 1) + t^{-1/2}p^{1/2}q^{-1/2} + \frac{1}{qt}\right) \wedge p,$$

where the last line uses the fact that the entire expression can be trivially bounded by $O(p)$. Plugging back into the original expression for (**II**), this term can thus be upper bounded as

$$\textbf{(II)} \lesssim \frac{L}{n} \sum_{t=1}^{T} \left(t^{-1}(\tau + 1) + t^{-1/2}p^{1/2}q^{-1/2} + \frac{1}{qt}\right) \wedge p + \frac{L\tau}{n}$$

$$\lesssim \frac{L}{n} \cdot \left(\sqrt{Tp/q} + \sum_{t=1}^{T} \frac{\tau + 1 + q^{-1}}{t} \wedge p\right) + \frac{L\tau}{n}$$

$$\lesssim \frac{L}{n} \cdot \left(\sqrt{Tp/q} + \tau + 1 + q^{-1} + \log\big(Tp/(\tau + 1 + q^{-1})\big)\right)$$

$$\lesssim \frac{L}{\sqrt{nq}} + \frac{L(\tau + q^{-1})}{n} + \frac{L\log n}{n}.$$

**Bounding (III):** Finally, for fixed $t \in [\tau + 1 : T - \tau - 1]$, we have $\mathbb{P}(i \in \mathcal{I}_{gap}) = p - p(1 - q)^{\tau} \lesssim pq\tau$. Therefore we can bound **(III)** as

$$\textbf{(III)} \leq \frac{G\mathbb{E}|\mathcal{I}_{gap}|}{n} \leq \frac{G(Tpq\tau + 2\tau)}{n} \leq G\tau(q + 2/n).$$

**Putting everything together.** Altogether, we have that

$$\mathbb{E}\left\|\frac{1}{T}\sum_{t=1}^{T} p(x_{t-\tau:t}, \mu(x_{\leq t})) - \frac{1}{n}\sum_{t=1}^{|z|} p(z_{t-\tau:t}, \mu(z_{\leq t}))\right\| \lesssim \frac{(G + L)(\tau + 1)}{n} + \frac{G + L}{\sqrt{nq}} + G\tau q + \frac{L\log n}{n}$$

$$\leq \frac{(G + L)(\tau + 1)}{n^{1/3}},$$

where the last inequality follows from choosing $q = n^{-1/3}$.

By the probabilistic method, there exists $\mathcal{I}$ such that $||\mathcal{I}| - \mathbb{E}|\mathcal{I}|| \leq 2n^{1/3} \implies ||\mathcal{I}| - n| \leq (\tau + 1 + 2n^{1/3})$ and

$$\left\| \frac{1}{T} \sum_{t=1}^{T} p(x_{t-\tau:t}, \mu(x_{\leq t})) - \frac{1}{n} \sum_{t=1}^{|z|} p(z_{t-\tau:t}, \mu(z_{\leq t})) \right\| \lesssim \frac{(G+L)(\tau+1)}{n^{1/3}}.$$

For this choice of $\mathcal{I}$, we have that

$$\left\| \frac{1}{T} \sum_{t=1}^{T} p(x_{t-\tau:t}, \mu(x_{\leq t})) - \frac{1}{|z|} \sum_{t=1}^{|z|} p(z_{t-\tau:t}, \mu(z_{\leq t})) \right\|$$

$$\lesssim \frac{(G+L)(\tau+1)}{n^{1/3}} + \left\| \frac{1}{|z|} \sum_{t=1}^{|z|} p(z_{t-\tau:t}, \mu(z_{\leq t})) \right\| \left| 1 - \frac{|z|}{n} \right|$$

$$\lesssim \frac{(G+L)(\tau+1)}{n^{1/3}},$$

as desired. $\qquad\qquad\qquad\qquad\qquad\qquad\qquad\qquad\qquad\qquad\qquad\qquad\qquad\qquad$ $\square$

## B.2 PROOF OF THEOREM 5.2

The proof begins by showing that the output of the first layer of attention at position $i$ can only depend on the histogram of the first $i$ tokens $\mu(x_{\leq i})$, along with the $\tau$-prefix of $x_{\leq i}$.

**Lemma B.1.** *Let $f$ be a fixed transformer with key, query and value matrices $\{(\boldsymbol{K}_{1,h}, \boldsymbol{Q}_{1,h}, \boldsymbol{V}_{1,h})\}_{h \in [H]} \cup \{(\boldsymbol{K}_{2,1}, \boldsymbol{Q}_{2,1}, \boldsymbol{V}_{2,1})\}$, MLP weights $\{(\boldsymbol{A}_l, \boldsymbol{B}_l)\}_{l \in \{1,2\}}$, embeddings $\|\boldsymbol{E}_s\| \leq 1$, and unembedding $\boldsymbol{U}$. There exists a function $q_f : [S]^{\tau+1} \times \Delta^S \times \mathbb{N} \to \mathbb{R}^d$ such that*

$$\boldsymbol{y}_i^{(1)} = q_f(x_{i-\tau:i}, \mu(x_{\leq i}), i)$$

*Moreover, $f$ satisfies*

$$q_f(w, \mu, i) \lesssim \left( 1 + \sum_{h=1}^{H} \|\boldsymbol{V}_{1,h}\|_{op} \right) (1 + \|\boldsymbol{B}_1\|_{op} \|\boldsymbol{A}_1\|_{op}) =: G_f$$

$$|q_f(w, \mu, i) - q_f(w, \mu, j)| \lesssim \left( 1 + \|\boldsymbol{B}_1\|_{op} \|\boldsymbol{A}_1\|_{op} \right) (\tau^2 + 1) \min(i,j)^{-\gamma} \sum_{h=1}^{H} \exp\left( 4 \left\| \boldsymbol{K}_{1,h}^{\top} \boldsymbol{Q}_{1,h} \right\|_{op} \right) =: H_f \min(i,j)^{-\gamma}$$

$$\|\nabla_\mu q_f(w, \mu, i)\|_{op} \leq 2S \left( \sum_{h=1}^{H} \|\boldsymbol{V}_{1,h}\|_{op} \exp\left( 4 \left\| \boldsymbol{K}_{1,h}^{\top} \boldsymbol{Q}_{1,h} \right\|_{op} \right) \right) \left( 1 + \|\boldsymbol{B}_1\|_{op} \|\boldsymbol{A}_1\|_{op} \right) =: L_f$$

*Proof.* Recall that the first layer self-attention logits are

$$a_{i,j}^{(1,h)} = \boldsymbol{E}_{x_j}^{\top} \boldsymbol{K}_{1,h}^{\top} \boldsymbol{Q}_{1,h} \boldsymbol{E}_{x_i} + \log i \cdot \phi_{1,h}(j,i)$$

and thus we can rewrite $\boldsymbol{Y}_i^{(1)}$ as

$$\boldsymbol{Y}_i^{(1)}$$

$$= \boldsymbol{E}_{x_i} + \sum_{h=1}^{H} \frac{\sum_{j=1}^{i} \exp\left(a_{i,j}^{(1,h)}\right) \boldsymbol{V}_{1,h} \boldsymbol{E}_{x_j}}{\sum_{j=1}^{i} \exp\left(a_{i,j}^{(1,h)}\right)}$$

$$= \boldsymbol{E}_{x_i} + \sum_{h=1}^{H} \frac{\sum_{j=1}^{i} \exp\left(\boldsymbol{E}_{x_j}^{\top} \boldsymbol{K}_{1,h}^{\top} \boldsymbol{Q}_{1,h} \boldsymbol{E}_{x_i}\right) \cdot i^{\phi_{1,h}(j,i)} \cdot \boldsymbol{V}_{1,h} \boldsymbol{E}_{x_j}}{\sum_{j=1}^{i} \exp\left(\boldsymbol{E}_{x_j}^{\top} \boldsymbol{K}_{1,h}^{\top} \boldsymbol{Q}_{1,h} \boldsymbol{E}_{x_i}\right) \cdot i^{\phi_{1,h}(j,i)}}$$

$$= \boldsymbol{E}_{x_i} +$$

$$\sum_{h=1}^{H} \frac{\sum_{s \in [S]} i \exp\left(\boldsymbol{E}_s^{\top} \boldsymbol{K}_{1,h}^{\top} \boldsymbol{Q}_{1,h} \boldsymbol{E}_{x_i}\right) \boldsymbol{V}_{1,h} \boldsymbol{E}_s \mu(x_{\leq i})_s + \sum_{j=i-\tau}^{i} \left(i^{\phi_{1,h}(j,i)} - 1\right) \exp\left(\boldsymbol{E}_{x_j}^{\top} \boldsymbol{K}_{1,h}^{\top} \boldsymbol{Q}_{1,h} \boldsymbol{E}_{x_i}\right) \boldsymbol{V}_{1,h} \boldsymbol{E}_{x_j}}{\sum_{s \in [S]} i \exp\left(\boldsymbol{E}_s^{\top} \boldsymbol{K}_{1,h}^{\top} \boldsymbol{Q}_{1,h} \boldsymbol{E}_{x_i}\right) \mu(x_{\leq i})_s + \sum_{j=i-\tau}^{i} \left(i^{\phi_{1,h}(j,i)} - 1\right) \exp\left(\boldsymbol{E}_{x_j}^{\top} \boldsymbol{K}_{1,h}^{\top} \boldsymbol{Q}_{1,h} \boldsymbol{E}_{x_i}\right)}$$

$$= \boldsymbol{E}_{x_i} + \sum_{h=1}^{H} \frac{N_h}{D_h},$$

where for each $h$, we have

$$N_h(x) := \sum_{s \in [S]} i^{1-\gamma_h} \exp\left(\boldsymbol{E}_s^{\top} \boldsymbol{K}_{1,h}^{\top} \boldsymbol{Q}_{1,h} \boldsymbol{E}_{x_i}\right) \boldsymbol{V}_{1,h} \boldsymbol{E}_s \mu(x_{\leq i})_s$$

$$+ \sum_{j=i-\tau}^{i} \left(i^{\phi_{1,h}(j,i)-\gamma_h} - i^{-\gamma_h}\right) \exp\left(\boldsymbol{E}_{x_j}^{\top} \boldsymbol{K}_{1,h}^{\top} \boldsymbol{Q}_{1,h} \boldsymbol{E}_{x_i}\right) \boldsymbol{V}_{1,h} \boldsymbol{E}_{x_j}$$

$$D_h(x) := \sum_{s \in [S]} i^{1-\gamma_h} \exp\left(\boldsymbol{E}_s^{\top} \boldsymbol{K}_{1,h}^{\top} \boldsymbol{Q}_{1,h} \boldsymbol{E}_{x_i}\right) \mu(x_{\leq i})_s + \sum_{j=i-\tau}^{i} \left(i^{\phi_{1,h}(j,i)-\gamma_h} - i^{-\gamma_h}\right) \exp\left(\boldsymbol{E}_{x_j}^{\top} \boldsymbol{K}_{1,h}^{\top} \boldsymbol{Q}_{1,h} \boldsymbol{E}_{x_i}\right)$$

$$\gamma_h := \max(1, \max_{0 \leq t \leq \tau} \phi_{1,h}(i-t,i)) = \max \mathcal{P}_h$$

Therefore we can write $\boldsymbol{Y}_i^{(1)} = q_{SA}(x_{i-\tau:i}, \mu(x_{\leq i}), i)$, where for $w_{0:\tau} \in [S]^{\tau+1}, \mu \in \Delta^S$, $q_{SA}(w, \mu, i)$ is given by

$$q_{SA}(w, \mu, i) = E_{w_\tau} + \sum_{h=1}^{H} \frac{N_h(w, \mu, i)}{D_h(w, \mu, i)},$$

where

$$N_h(w, \mu, i) := \sum_{s \in [S]} i^{1-\gamma_h} \exp\left(\boldsymbol{E}_s^{\top} \boldsymbol{K}_{1,h}^{\top} \boldsymbol{Q}_{1,h} \boldsymbol{E}_{w_\tau}\right) \boldsymbol{V}_{1,h} \boldsymbol{E}_s \mu_s$$

$$+ \sum_{t=0}^{\tau} \left(i^{\phi_{1,h}(i-t,i)-\gamma_h} - i^{-\gamma_h}\right) \exp\left(\boldsymbol{E}_{w_t}^{\top} \boldsymbol{K}_{1,h}^{\top} \boldsymbol{Q}_{1,h} \boldsymbol{E}_{w_\tau}\right) \boldsymbol{V}_{1,h} \boldsymbol{E}_{w_t}$$

$$D_h(w, \mu, i) := \sum_{s \in [S]} i^{1-\gamma_h} \exp\left(\boldsymbol{E}_s^{\top} \boldsymbol{K}_{1,h}^{\top} \boldsymbol{Q}_{1,h} \boldsymbol{E}_{w_\tau}\right) \mu_s + \sum_{t=0}^{\tau} \left(i^{\phi_{1,h}(i-\tau,i)-\gamma_h} - i^{-\gamma_h}\right) \exp\left(\boldsymbol{E}_{w_t}^{\top} \boldsymbol{K}_{1,h}^{\top} \boldsymbol{Q}_{1,h} \boldsymbol{E}_{w_\tau}\right)$$

Each $N_h(w, \mu, i)/D_h(w, \mu, i)$ term in the above sum is of the form of the expression in Lemma B.2.

First, we see that each denominator can be lower bounded by $\exp\left(-\left\|\boldsymbol{K}_{1,h}^{\top} \boldsymbol{Q}_{1,h}\right\|_{op}\right)$. Moreover, we have

$$\sum_{k} \|A_k\| \lesssim (\tau+1)\|\boldsymbol{V}_{1,h}\|_{op} \exp\left(\left\|\boldsymbol{K}_{1,h}^{\top} \boldsymbol{Q}_{1,h}\right\|_{op}\right)$$

$$\sum_{k} |B_k| \lesssim (\tau+1) \exp\left(\left\|\boldsymbol{K}_{1,h}^{\top} \boldsymbol{Q}_{1,h}\right\|_{op}\right)$$

Altogether, since $\gamma := \gamma(f) = \min_{h \in H} (\gamma_h - \max\{p \in \mathcal{P}_h : p \neq \gamma_h\})$, we can bound

$$|q_{SA}(w, \mu, i) - q_{SA}(w, \mu, j)| \lesssim (\tau^2 + 1)j^{-\gamma} \sum_{h=1}^{H} \exp\left(4\left\|\boldsymbol{K}_{1,h}^\top \boldsymbol{Q}_{1,h}\right\|_{op}\right).$$

Next, see that $\boldsymbol{y}_i^{(1)} = q_{MLP}(\boldsymbol{Y}_i^{(1)})$, where

$$q_{MLP}(x) = x + \boldsymbol{B}_1 \psi_1(\boldsymbol{A}_1 x + \boldsymbol{b}_1).$$

Since $\psi_1$ is 1-Lipschitz, $q_{MLP}$ is $1 + \|\boldsymbol{B}_1\|_{op}\|\boldsymbol{A}_1\|_{op}$ Lipschitz. Altogether, since $q_f(w, \mu, i) = q_{MLP}(q_{SA}(w, \mu, i))$, we have

$$|q_f(w, \mu, i) - q_f(w, \mu, j)| \lesssim \left(1 + \|\boldsymbol{B}_1\|_{op}\|\boldsymbol{A}_1\|_{op}\right)(\tau^2 + 1)j^{-\gamma} \exp\left(4\left\|\boldsymbol{K}_{1,h}^\top \boldsymbol{Q}_{1,h}\right\|_{op}\right).$$

Next, for the uniform bound, observe that $\left\|\boldsymbol{Y}_i^{(1)}\right\| \leq 1 + \sum_{h=1}^{H} \|\boldsymbol{V}_{1,h}\|_{op}$, and therefore $\left\|\boldsymbol{y}_i^{(1)}\right\| \leq \left(1 + \sum_{h=1}^{H} \|\boldsymbol{V}_{1,h}\|_{op}\right)(1 + \|\boldsymbol{B}_1\|_{op}\|\boldsymbol{A}_1\|_{op})$.

Finally, we compute the Lipschitz constant with respect to $\mu$.

Let $M_h \in \mathbb{R}^{d \times S}$ be the matrix with the $s$th column being $\exp\left(\boldsymbol{E}_s^\top \boldsymbol{K}_{1,h}^\top \boldsymbol{Q}_{1,h} \boldsymbol{E}_{w_\tau}\right) \boldsymbol{V}_{1,h} \boldsymbol{E}_s$, let $b_h$ be the vector with $s$th entry $\exp\left(\boldsymbol{E}_s^\top \boldsymbol{K}_{1,h}^\top \boldsymbol{Q}_{1,h} \boldsymbol{E}_{w_\tau}\right)$, and let $C_h = \sum_{t=0}^{\tau} \left(i^{\phi_{1,h}(i-\tau,i)-\gamma_h} - i^{-\gamma_h}\right) \exp\left(\boldsymbol{E}_{w_t}^\top \boldsymbol{K}_{1,h}^\top \boldsymbol{Q}_{1,h} \boldsymbol{E}_{w_\tau}\right) > 0$. We have that

$$\nabla_\mu q_{SA}(w, \mu, i) = \sum_{h=1}^{H} \frac{i^{1-\gamma_h} M_h}{i^{1-\gamma_h} \langle b_h, \mu \rangle + C_h} - \frac{i^{2(1-\gamma_h)} M_h \mu b_j^\top}{(i^{1-\gamma_h} \langle b_h, \mu \rangle + C_h)^2},$$

and since $\langle b_h, \mu \rangle \geq \exp\left(-\left\|\boldsymbol{K}_{1,h}^\top \boldsymbol{Q}_{1,h}\right\|_{op}\right)$ and

$$\|M_h\|_{op} \leq \sqrt{S}\|\boldsymbol{V}_{1,h}\|_{op} \exp\left(\left\|\boldsymbol{K}_{1,h}^\top \boldsymbol{Q}_{1,h}\right\|_{op}\right) \quad \text{and} \quad \|b_h\| \leq \sqrt{S} \exp\left(\left\|\boldsymbol{K}_{1,h}^\top \boldsymbol{Q}_{1,h}\right\|_{op}\right),$$

we have that

$$\|\nabla_\mu q_{SA}(w, \mu, i)\|_{op} \leq \sum_{h=1}^{H} 2S\|\boldsymbol{V}_{1,h}\|_{op} \exp\left(4\left\|\boldsymbol{K}_{1,h}^\top \boldsymbol{Q}_{1,h}\right\|_{op}\right).$$

Altogether,

$$\|\nabla_\mu q_f(w, \mu, i)\|_{op} \leq 2S\left(\sum_{h=1}^{H} \|\boldsymbol{V}_{1,h}\|_{op} \exp\left(4\left\|\boldsymbol{K}_{1,h}^\top \boldsymbol{Q}_{1,h}\right\|_{op}\right)\right)\left(1 + \|\boldsymbol{B}_1\|_{op}\|\boldsymbol{A}_1\|_{op}\right).$$

$\square$

In order to prove the main theorem, it suffices to apply the key simulation lemma Lemma 5.3.

*Proof of Theorem 5.2.* Let $f, g \in \mathcal{F}_\tau$ be two transformers. In the forward pass of $f$, the second layer logits are given by

$$a_{i,j}^{(2,1)} = \left(\boldsymbol{y}_j^{(1)}\right)^\top \boldsymbol{K}_{2,1}^\top \boldsymbol{Q}_{2,1} \boldsymbol{y}_i^{(1)} = q_f(x_{j-\tau:j}, \mu(x_{\leq j}), j)^\top \boldsymbol{K}_{2,1}^\top \boldsymbol{Q}_{2,1} q_f(x_{i-\tau:i}, \mu(x_{\leq i}), i),$$

and therefore

$$\boldsymbol{Y}_T^{(2)} = q_f(x_{T-\tau:T}, \mu(x_{\leq T}), T)$$
$$+ \boldsymbol{V}_{2,1} \sum_{j=1}^{T} \frac{\exp\left(q_f(x_{j-\tau:j}, \mu(x_{\leq j}), j)^\top \boldsymbol{K}_{2,1}^\top \boldsymbol{Q}_{2,1} q_f(x_{T-\tau:T}, \mu(x_{\leq T}), T)\right) q_f(x_{j-\tau:j}, \mu(x_{\leq j}), j)}{\sum_{j=1}^{T} \exp\left(q_f(x_{j-\tau:j}, \mu(x_{\leq j}), j)^\top \boldsymbol{K}_{2,1}^\top \boldsymbol{Q}_{2,1} q_f(x_{T-\tau:T}, \mu(x_{\leq T}), T)\right)}.$$

The analogous expression holds for the second layer logits in the forward pass of $g$.

Let us define the following sequence of functions:

$$p_0(w,\mu) = e_{w-1}$$
$$p_{f,1}(w,\mu) = \exp\Big(q_f(w,\mu,T)^\top \boldsymbol{K}_{2,1}^\top \boldsymbol{Q}_{2,1} q_f(x_{T-\tau,T},\mu(x),T)\Big)$$
$$p_{f,2}(w,\mu) = p_{f,1}(w,\mu)q_f(w,\mu),$$

along with the analogous $p_{g,1}, p_{g,2}$ for the transformer $g$. We first see that $\|p_0\| \le 1$ and $p_0$ is constant in $\mu$.

Next, we have that we can uniformly bound $p_{f,1}$ by

$$|p_{f,1}(w,\mu)| \le \exp\left(G_f^2 \left\|\boldsymbol{K}_{2,1}^\top \boldsymbol{Q}_{2,1}\right\|_{op}\right),$$

and the Lipschitz bound

$$\nabla_\mu p_{f,1}(w,\mu) = \exp\Big(q_f(w,\mu,T)^\top \boldsymbol{K}_{2,1}^\top \boldsymbol{Q}_{2,1} q_f(x_{T-\tau,T},\mu(x),T)\Big)\nabla_\mu q_f(w,\mu,T)\boldsymbol{K}_{2,1}^\top \boldsymbol{Q}_{2,1} q_f(x_{T-\tau,T},\mu(x),T)$$
$$\implies \|\nabla_\mu p_{f,1}(w,\mu)\| \le \exp\left(G_f^2 \left\|\boldsymbol{K}_{2,1}^\top \boldsymbol{Q}_{2,1}\right\|_{op}\right)\left\|\boldsymbol{K}_{2,1}^\top \boldsymbol{Q}_{2,1}\right\|_{op} G_f L_f.$$

Finally, for $p_{f,2}$ we have the uniform bound

$$|p_{f,2}(w,\mu)| \le \exp\left(G_f^2 \left\|\boldsymbol{K}_{2,1}^\top \boldsymbol{Q}_{2,1}\right\|_{op}\right)G_f$$

and the Lipschitz bound

$$\nabla_\mu p_{f,2}(w,\mu) = p_{f,1}(w,\mu)\nabla_\mu q_f(w,\mu,T) + \nabla_\mu p_1(w,\mu)q_f(w,\mu)$$
$$\implies \|\nabla_\mu p_{f,2}(w,\mu)\| \le \exp\left(G_f^2 \left\|\boldsymbol{K}_{2,1}^\top \boldsymbol{Q}_{2,1}\right\|_{op}\right)\left\|\boldsymbol{K}_{2,1}^\top \boldsymbol{Q}_{2,1}\right\|_{op} G_f^2 L_f.$$

Define the quantity $M_f$ as

$$M_f := \exp\left(G_f^2 \left\|\boldsymbol{K}_{2,1}^\top \boldsymbol{Q}_{2,1}\right\|_{op}\right)\left\|\boldsymbol{K}_{2,1}^\top \boldsymbol{Q}_{2,1}\right\|_{op} G_f^2 L_f,$$

and analogously for $M_g$.

Let us define the function $p$ by

$$p(w,\mu) = \begin{bmatrix} M_f M_g \cdot p_0(w,\mu) \\ G_f M_g \cdot p_{f,1}(w,\mu) \\ M_g \cdot p_{f,2}(w,\mu) \\ M_f G_g \cdot p_{g,1}(w,\mu) \\ M_f \cdot p_{g,2}(w,\mu) \end{bmatrix}$$

$p$ is both uniformly bounded by and has a Lipschitz constant of $M_f M_g$. Therefore by Lemma 5.3, we have the following bounds:

$$\left\| \frac{1}{T} \sum_{j=1}^{T} p_0(x_{j-\tau:j}, \mu(x_{\leq j})) - \frac{1}{|z|} \sum_{j=1}^{|z|} p_0(z_{j-\tau:j}, \mu(z_{\leq j})) \right\| \lesssim \frac{\tau+1}{n^{1/3}}$$

$$\left| \frac{1}{T} \sum_{j=1}^{T} p_{f,1}(x_{j-\tau:j}, \mu(x_{\leq j})) - \frac{1}{|z|} \sum_{j=1}^{|z|} p_{f,1}(z_{j-\tau:j}, \mu(z_{\leq j})) \right| \lesssim \frac{M_f G_f^{-1}(\tau+1)}{n^{1/3}}$$

$$\left\| \frac{1}{T} \sum_{j=1}^{T} p_{f,2}(x_{j-\tau:j}, \mu(x_{\leq j})) - \frac{1}{|z|} \sum_{j=1}^{|z|} p_{f,2}(z_{j-\tau:j}, \mu(z_{\leq j})) \right\| \lesssim \frac{M_f(\tau+1)}{n^{1/3}}$$

$$\left| \frac{1}{T} \sum_{j=1}^{T} p_{g,1}(x_{j-\tau:j}, \mu(x_{\leq j})) - \frac{1}{|z|} \sum_{j=1}^{|z|} p_{g,1}(z_{j-\tau:j}, \mu(z_{\leq j})) \right| \lesssim \frac{M_g G_g^{-1}(\tau+1)}{n^{1/3}}$$

$$\left\| \frac{1}{T} \sum_{j=1}^{T} p_{g,2}(x_{j-\tau:j}, \mu(x_{\leq j})) - \frac{1}{|z|} \sum_{j=1}^{|z|} p_{g,2}(z_{j-\tau:j}, \mu(z_{\leq j})) \right\| \lesssim \frac{M_g(\tau+1)}{n^{1/3}}.$$

Let's first look at $p_0$. Observe that

$$\frac{1}{T} \sum_{j=1}^{T} p_0(x_{j-\tau:j}, \mu(x_{\leq j})) = \frac{1}{T} \sum_{j=1}^{T} e_{x_j} = \mu(x),$$

and therefore

$$\|\mu(x) - \mu(z)\| = \left\| \frac{1}{T} \sum_{j=1}^{T} p_0(x_{j-\tau:j}, \mu(x_{\leq j})) - \frac{1}{|z|} \sum_{j=1}^{|z|} p_0(z_{j-\tau:j}, \mu(z_{\leq j})) \right\| \lesssim \frac{\tau+1}{n^{1/3}}.$$

Next let's look at $p_{f,1}$. We have that

$$\left| \frac{1}{T} \sum_{j=1}^{T} \exp\Big( q_f(x_{j-\tau:j}, \mu(x_{\leq j}), j)^\top \boldsymbol{K}_{2,1}^\top \boldsymbol{Q}_{2,1} q_f(x_{T-\tau:T}, \mu(x_{\leq T}), T) \Big) - \frac{1}{T} \sum_{j=1}^{T} p_{f,1}(x_{j-\tau:j}, \mu(x_{\leq j})) \right|$$

$$\leq \frac{1}{T} \sum_{j=1}^{T} \left| \exp\Big( q_f(x_{j-\tau:j}, \mu(x_{\leq j}), j)^\top \boldsymbol{K}_{2,1}^\top \boldsymbol{Q}_{2,1} q_f(x_{T-\tau:T}, \mu(x_{\leq T}), T) \Big) \right.$$

$$\left. - \exp\Big( q_f(x_{j-\tau:j}, \mu(x_{\leq j}), T)^\top \boldsymbol{K}_{2,1}^\top \boldsymbol{Q}_{2,1} q_f(x_{T-\tau:T}, \mu(x_{\leq T}), T) \Big) \right|$$

$$\leq \frac{1}{T} \sum_{j=1}^{T} \exp\Big( G_f^2 \big\| \boldsymbol{K}_{2,1}^\top \boldsymbol{Q}_{2,1} \big\|_{op} \Big) \big\| \boldsymbol{K}_{2,1}^\top \boldsymbol{Q}_{2,1} \big\|_{op} G_f \cdot H_f j^{-\gamma(f)}$$

$$= \exp\Big( G_f^2 \big\| \boldsymbol{K}_{2,1}^\top \boldsymbol{Q}_{2,1} \big\|_{op} \Big) \big\| \boldsymbol{K}_{2,1}^\top \boldsymbol{Q}_{2,1} \big\|_{op} G_f \cdot H_f \xi_{\gamma(f)}(T),$$

where we're letting $\xi_\gamma(T) := \frac{1}{T} \sum_{j=1}^{T} j^{-\gamma}$

Next, note that

$$
\left| \frac{1}{|z|} \sum_{j=1}^{|z|} \exp\Big( q_f(z_{j-\tau:j}, \mu(z_{\leq j}), j)^\top \boldsymbol{K}_{2,1}^\top \boldsymbol{Q}_{2,1} q_f(z_{|z|-\tau:|z|}, \mu(z), |z|) \Big) - \frac{1}{|z|} \sum_{j=1}^{|z|} p_{f,1}(z_{j-\tau:j}, \mu(z_{\leq j})) \right|
$$

$$
\leq \frac{1}{|z|} \sum_{j=1}^{|z|} \exp\Big( G_f^2 \big\| \boldsymbol{K}_{2,1}^\top \boldsymbol{Q}_{2,1} \big\|_{op} \Big) \Big| q_f(z_{j-\tau:j}, \mu(z_{\leq j}), j)^\top \boldsymbol{K}_{2,1}^\top \boldsymbol{Q}_{2,1} q_f(z_{|z|-\tau:|z|}, \mu(z), |z|)
$$

$$
\qquad - q_f(z_{j-\tau:j}, \mu(z_{\leq j}), T)^\top \boldsymbol{K}_{2,1}^\top \boldsymbol{Q}_{2,1} q_f(x_{|z|-\tau:|z|}, \mu(x), T) \Big|
$$

$$
\leq \frac{1}{|z|} \sum_{j=1}^{|z|} \exp\Big( G_f^2 \big\| \boldsymbol{K}_{2,1}^\top \boldsymbol{Q}_{2,1} \big\|_{op} \Big) G_f \big\| \boldsymbol{K}_{2,1}^\top \boldsymbol{Q}_{2,1} \big\|_{op} \Big( \| q_f(z_{j-\tau:j}, \mu(z_{\leq j}), j) - q_f(z_{j-\tau:j}, \mu(z_{\leq j}), T) \|
$$

$$
\qquad + \| q_f(z_{|z|-\tau:|z|}, \mu(z), |z|) - q_f(x_{|z|-\tau:|z|}, \mu(x), T) \| \Big)
$$

$$
\leq \frac{1}{|z|} \sum_{j=1}^{|z|} \exp\Big( G_f^2 \big\| \boldsymbol{K}_{2,1}^\top \boldsymbol{Q}_{2,1} \big\|_{op} \Big) G_f \big\| \boldsymbol{K}_{2,1}^\top \boldsymbol{Q}_{2,1} \big\|_{op} \Big( H_f j^{-\gamma(f)} + H_f |z|^{-\gamma(f)} + L_f \| \mu(z) - \mu(x) \| \Big)
$$

$$
\lesssim \exp\Big( G_f^2 \big\| \boldsymbol{K}_{2,1}^\top \boldsymbol{Q}_{2,1} \big\|_{op} \Big) G_f \big\| \boldsymbol{K}_{2,1}^\top \boldsymbol{Q}_{2,1} \big\|_{op} \Big( H_f \xi_{\gamma(f)}(|z|) + L_f(\tau+1) n^{-1/3} \Big)
$$

Altogether,

$$
\left| \frac{1}{T} \sum_{j=1}^{T} \exp\Big( q_f(x_{j-\tau:j}, \mu(x_{\leq j}), j)^\top \boldsymbol{K}_{2,1}^\top \boldsymbol{Q}_{2,1} q_f(x_{T-\tau:T}, \mu(x_{\leq T}), T) \Big) \right.
$$

$$
\left. - \frac{1}{|z|} \sum_{j=1}^{|z|} \exp\Big( q_f(z_{j-\tau:j}, \mu(z_{\leq j}), j)^\top \boldsymbol{K}_{2,1}^\top \boldsymbol{Q}_{2,1} q_f(z_{|z|-\tau:|z|}, \mu(z), |z|) \Big) \right|
$$

$$
\lesssim \exp\Big( G_f^2 \big\| \boldsymbol{K}_{2,1}^\top \boldsymbol{Q}_{2,1} \big\|_{op} \Big) G_f \big\| \boldsymbol{K}_{2,1}^\top \boldsymbol{Q}_{2,1} \big\|_{op} \Big( H_f \xi_{\gamma(f)}(|z|) + L_f(\tau+1) n^{-1/3} \Big)
$$

$$
\qquad + \left| \frac{1}{T} \sum_{j=1}^{T} p_{f,1}(x_{j-\tau:j}, \mu(x_{\leq j})) - \frac{1}{|z|} \sum_{j=1}^{|z|} p_{f,1}(z_{j-\tau:j}, \mu(z_{\leq j})) \right|
$$

$$
\lesssim \frac{\exp\Big( G_f^2 \big\| \boldsymbol{K}_{2,1}^\top \boldsymbol{Q}_{2,1} \big\|_{op} \Big) L_f(\tau+1)}{n^{1/3}}
$$

$$
\lesssim \exp\Big( G_f^2 \big\| \boldsymbol{K}_{2,1}^\top \boldsymbol{Q}_{2,1} \big\|_{op} \Big) G_f \big\| \boldsymbol{K}_{2,1}^\top \boldsymbol{Q}_{2,1} \big\|_{op} \Big( H_f \xi_{\gamma(f)}(n) + L_f(\tau+1) n^{-1/3} \Big).
$$

Similarly, we can bound the numerators by

$$
\left| \frac{1}{T} \sum_{j=1}^{T} \exp\Big( q_f(x_{j-\tau:j}, \mu(x_{\leq j}), j)^\top \boldsymbol{K}_{2,1}^\top \boldsymbol{Q}_{2,1} q_f(x_{T-\tau:T}, \mu(x_{\leq T}), T) \Big) q_f(x_{j-\tau:j}, \mu(x_{\leq j}), j) \right.
$$

$$
\left. - \frac{1}{|z|} \sum_{j=1}^{|z|} \exp\Big( q_f(z_{j-\tau:j}, \mu(z_{\leq j}), j)^\top \boldsymbol{K}_{2,1}^\top \boldsymbol{Q}_{2,1} q_f(z_{|z|-\tau:|z|}, \mu(z), |z|) \Big) q_f(z_{j-\tau:j}, \mu(z_{\leq j}), j) \right|
$$

$$
\lesssim \exp\Big( G_f^2 \big\| \boldsymbol{K}_{2,1}^\top \boldsymbol{Q}_{2,1} \big\|_{op} \Big) G_f^2 \big\| \boldsymbol{K}_{2,1}^\top \boldsymbol{Q}_{2,1} \big\|_{op} \Big( H_f \xi_{\gamma(f)}(n) + L_f(\tau+1) n^{-1/3} \Big)
$$

Finally, we bound

$$
\big\| q_f(x_{T-\tau:T}, \mu(x), T) - q_f(z_{|z|-\tau:|z|}, \mu(z), |z|) \big\| \leq L_f \| \mu(x) - \mu(z) \| + H_f |z|^{-\gamma(f)}
$$

$$
\lesssim L_f(\tau+1) n^{-1/3} + H_f n^{-\gamma(f)}.
$$

Altogether, we can relate $\boldsymbol{Y}_T^{(2)}(x)$ and $\boldsymbol{Y}_{|z|}^{(2)}(z)$ by

$$\left\|\boldsymbol{Y}_T^{(2)}(x) - \boldsymbol{Y}_{|z|}^{(2)}(z)\right\| \lesssim \exp\left(4G_f^2\left\|\boldsymbol{K}_{2,1}^\top\boldsymbol{Q}_{2,1}\right\|_{op}\right)G_f^2\left\|\boldsymbol{K}_{2,1}^\top\boldsymbol{Q}_{2,1}\right\|_{op}\left(1 + \|\boldsymbol{V}_2\|_{op}\right)\left(H_f\xi_{\gamma(f)}(n) + L_f(\tau+1)n^{-1/3}\right).$$

Finally, we have that

$$\|f(x) - f(z)\| \leq \|\boldsymbol{U}\|\left(1 + \|\boldsymbol{B}_2\|_{op}\|\boldsymbol{A}_2\|_{op}\right)\left\|\boldsymbol{Y}_T^{(2)}(x) - \boldsymbol{Y}_{|z|}^{(2)}(z)\right\|.$$

Plugging in the expressions for $G_f, L_f, H_f$, and noting that $\xi_\gamma(n) \lesssim n^{-(1/3\wedge\gamma)}$, yields

$$\|f(x) - f(z)\| \lesssim C(f)n^{-(1/3\wedge\gamma(f))},$$

where

$$C(f) := \exp\left(C\left(1 + \sum_{h=1}^H \|\boldsymbol{V}_{1,h}\|_{op}\right)^2(1 + \|\boldsymbol{B}_1\|_{op}\|\boldsymbol{A}_1\|_{op})^2\left\|\boldsymbol{K}_{2,1}^\top\boldsymbol{Q}_{2,1}\right\|_{op}\right)\left(1 + \|\boldsymbol{V}_2\|_{op}\right)$$

$$\times \left(\sum_{h=1}^H \|\boldsymbol{V}_{1,h}\|_{op}\exp\left(4\left\|\boldsymbol{K}_{1,h}^\top\boldsymbol{Q}_{1,h}\right\|_{op}\right)\right)\left(1 + \|\boldsymbol{B}_2\|_{op}\|\boldsymbol{A}_2\|_{op}\right)\|\boldsymbol{U}\|_{op}(\tau^2 + 1)S.$$

Repeating the above argument for the transformer $g$, we have that

$$\|g(x) - g(z)\| \lesssim C(g)n^{-(1/3\wedge\gamma(g))}.$$

Combining these together implies the desired result. $\qquad\square$

### B.2.1 HELPER LEMMAS

**Lemma B.2.** *Let $f(i)$ be of the form*

$$f(i) = \frac{\sum_k A_k i^{-\gamma_k}}{\sum_k B_k i^{-\gamma_k}},$$

*where $A_k \in \mathbb{R}^d$ and $\sum_k B_k i^{-\gamma_k} \geq \delta$ for all $i$. Assume that $0 = \gamma_1 < \gamma_2 < \cdots < \gamma_K$. Then, for $j < i$,*

$$\|f(i) - f(j)\| \leq \delta^{-2}j^{-\gamma_2}\left(\sum_k \|A_k\|\right)\left(\sum_k |B_k|\right)$$

*Proof.* One can write

$$\|f(i) - f(j)\| = \frac{\left\|(\sum_k A_k i^{-\gamma_k})(\sum_k B_k j^{-\gamma_k}) - (\sum_k A_k j^{-\gamma_k})(\sum_k B_k i^{-\gamma_k})\right\|}{\left|(\sum_k B_k i^{-\gamma_k})(\sum_k B_k j^{-\gamma_k})\right|}$$

$$\leq \delta^{-2}\left\|\sum_{l\neq k}(A_lB_k - A_kB_l)i^{-\gamma_l}j^{-\gamma_k}\right\|$$

$$\leq \delta^{-2}j^{-\gamma_2}\left(\sum_k \|A_k\|\right)\left(\sum_k |B_k|\right)$$

$\qquad\square$

### B.3 IN-CONTEXT $k$-GRAM CONSTRUCTION

Below, we sketch the in-context $k$-gram construction, which closely follows Construction 2 in Nichani et al. (2024).

In the first layer, the $h$-th head will attend fully to the $(i - h)$-th token; this is done by setting $\phi_{1,h}(i - h, i)$ to be large, and the rest of the entries of $\phi$, along with $\boldsymbol{K}_{1,h}$ and $\boldsymbol{Q}_{1,h}$, equal to 0. By

choosing an embedding dimension of $d \geq (\tau + 1)S$, the value matrices $\boldsymbol{V}_{1,h}$ can be chosen such that $\boldsymbol{Y}_i^{(1)} = \boldsymbol{E}_{x_i} \oplus \boldsymbol{E}_{x_{i-1}} \oplus \cdots \oplus \boldsymbol{E}_{x_{i-\tau}}$; this is accomplished via $\boldsymbol{V}_{1,h}$ being a block identity matrix, which thus satisfies $\|\boldsymbol{V}_{1,h}\|_{op} = 1$. The first layer MLP is then set to identically zero, so that $\boldsymbol{y}^{(1)} = \boldsymbol{Y}^{(1)}$

In the second layer, $\boldsymbol{K}_{2,1}^\top \boldsymbol{Q}_{2,1}$ is equal to $\beta$ times another block-identity matrix, which compares the $\boldsymbol{E}_{x_{i-1}} \oplus \cdots \oplus \boldsymbol{E}_{x_{i-\tau}}$ subspace of $\boldsymbol{y}_i^{(1)}$ to the $\boldsymbol{E}_{x_T} \oplus \cdots \oplus \boldsymbol{E}_{x_{T-\tau+1}}$ subspace of $\boldsymbol{y}_T^{(1)}$. This places an attention weight of $e^{\tau\beta}$ on each token $i$ with $x_{i-\tau:i-1} = x_{T-\tau+1:T}$, and a weight of at most $e^{\tau(\beta-1)}$ on all other tokens. Finally, the second value matrix $\boldsymbol{V}_{2,1}$ copies from the $\boldsymbol{E}_{x_i}$ subspace of $\boldsymbol{y}_i$, while the second layer MLP is also zero.

Altogether, the output of the transformer is

$$f(x_{1:T})_s = \frac{e^{\beta\tau} \sum_i \mathbf{1}(x_{i-\tau:i-1} = x_{T-\tau+1:T}, x_i = s) + E_{N,s}}{e^{\beta\tau} \sum_i \mathbf{1}(x_{i-\tau:i-1} = x_{T-\tau+1:T}) + E_D},$$

where $\|E_N\|, |E_D| \leq T e^{\beta(\tau-1)}$. Therefore

$$\|f(x_{1:T}) - f^*(x_{1:T})\| \lesssim \frac{T}{e^\beta \sum_i \mathbf{1}(x_{i-\tau:i-1} = x_{T-\tau+1:T})}$$

On a "typical" sequence $x$, $\sum_i \mathbf{1}(x_{i-\tau:i-1} = x_{T-\tau+1:T}) = \Theta(T)$, in which case

$$\|f(x_{1:T}) - f^*(x_{1:T})\| \lesssim e^{-\beta} \leq \varepsilon$$

whenever $\beta \asymp \log(1/\varepsilon)$. Therefore $\left\|\boldsymbol{K}_{2,1}^\top \boldsymbol{Q}_{2,1}\right\|_{op} = \Theta(\log(1/\varepsilon))$. Plugging in, this yields a complexity measure of

$$C(f) = \exp\left(C k^2 \log(1/\varepsilon)\right) k^2 S = \varepsilon^{-\Theta(k^2)}.$$

## C    EXPERIMENTAL METHODOLOGY

**Data Generation:**

- SimpleTask: Each sequence $x_{1:T}$ is generated by first sampling a probability vector $\mathbf{p} \in \mathbb{R}^3$ uniformly at random over the simplex, then sampling each $x_i$ i.i.d, where $x_i = s$ with probability $\mathbf{p}_s$. This ensures that $\mathrm{Var}(f^*) = \Theta(1)$. We vary $\omega$ between 2 and 5.5 in intervals of 0.5.

- ModPTask: Each sequence $x_{1:T}$ is generated by first generating $q_0, \ldots, q_{p-1}$ i.i.d uniformly from $[0, 1]$. Then, each $x_i$ is sampled from $\mathrm{Bernoulli}(p_k)$, where $k \equiv i \mod p$. This ensures that $\mathrm{Var}(f^*) = \Theta(1)$, and also that attending to incorrect positions mod $p$ cannot help the model. We vary $\Delta$ from 3 to 8

- *In-context $k$-gram:* The data generation follows that of Nichani et al. (2024). Each sequence $x_{1:T}$ is generated by first sampling a $k$-wise transition tensor $\pi \in [S]^k$, where for any $z_{1:k-1}$ the distribution $\pi(\cdot \mid z_{1:k-1})$ is sampled uniformly at random over the simplex in $S$ dimensions. Next, $x_{1:k-1}$ are sampled uniformly at random. Finally, for $i \geq k$, we sample $x_i \sim \pi(\cdot \mid x_{i-k:i-1})$. To ensure that $x_{T-k+1:T}$ occurs at least once in the sequence, we randomly select an index $i \in [k : T-1]$, and replace $x_{i-k+1:i}$ with $x_{T-k+1:T}$. We fix $k = 2$ and vary $S$ from 3 to 8, and also fix $S = 2$ and vary $k$ from 2 to 4.

**Training Procedure:**

- Single-layer transformers: The model architecture is one layer of a single self-attention head followed by an MLP. The embedding dimension is $d = 16$ and the MLP width is 256. We use the $\mu$P initialization Yang et al. (2022), and train using the Adam optimizer with learning rate $\eta = 10^{-2}/d$ for the hidden layers and $\eta = 10^{-2}$ for the embedding layers. We train all of the models using online SGD (sampling a fresh batch of size 1024 at each step), until the training loss crosses below $10^{-5}$. All results are averaged over 8 random seeds.

- Two-layer transformers: The model architecture is a two-layer transformer, with $k-1$ heads in the first layer and one head in the second layer. The embedding dimension is either $d = 32$ (when $k = 2$ is fixed and $S$ ranges from 3 to 8) or $d = 16$ (when $S = 2$ and $k$ ranges from 2 to 4). We use the $\mu$P initialization and train using the Adam optimizer with learning rate $\eta = 3 \cdot 10^{-2}/d$ for the hidden layers and $\eta = 10^{-2}$ for the embedding layers, on a fresh batch of size 1024 at each step for $2^{15}$ steps. The $k = 2$ results are averaged over 8 random seeds, while the $S = 2$ results are averaged over 14 random seeds.

## C.1 EXPRESSIVITY OF SYNTHETIC TASKS

We sketch the constructions for each of the synthetic tasks in Section 6.

**SimpleTask:** Set $\boldsymbol{p}_i = 0$, and let $\boldsymbol{E}_0, \boldsymbol{E}_1, \boldsymbol{E}_2$ be orthogonal. Choose $\boldsymbol{K}, \boldsymbol{Q}$ so that $a_{i,j} = \infty$ when $j = 0, 1$ and $a_{i,j} = 0$ when $j = 2$. The attention probabilities will then be uniform over all 0 and 1 tokens, and thus the output of self-attention becomes $\boldsymbol{Y}_T = \boldsymbol{E}_{x_T} + \frac{c_0(x)}{c_0(x)+c_1(x)}\boldsymbol{V}\boldsymbol{E}_0 + \frac{c_1(x)}{c_0(x)+c_1(x)}\boldsymbol{V}\boldsymbol{E}_1$. We can then set $\boldsymbol{V}\boldsymbol{E}_0 = -\boldsymbol{V}\boldsymbol{E}_1$. It suffices to approximate the one-dimensional function $z \mapsto \sin(\omega z)$ with an MLP; it is well known (Barron, 1993) that this can be done with weight norms $\Theta(\omega)$, as desired.

**ModPTask:** Let $\{\boldsymbol{q}_i\}_{i \in [\Delta]}$ be some fixed set of orthogonal embeddings, and let $\boldsymbol{p}_i$ be equal to $\boldsymbol{q}_j$, where $i \neq j \mod p$. These are periodic embeddings with periodicity $\Delta = p$. Choose $\boldsymbol{K}, \boldsymbol{Q}$ so that $a_{i,j}$ equals $\infty$ if $j \equiv k \mod p$ and 0 otherwise. The attention probabilities will then be uniform over all positions which are $k \mod p$. Choosing $\boldsymbol{V}$ so that $\boldsymbol{V}\boldsymbol{q}_j = 0$ for all $j$, the output of self-attention becomes $\boldsymbol{Y}_T = \boldsymbol{y}_T + f^*(x_{1:T})\boldsymbol{V}\boldsymbol{E}_1 + (1 - f^*(x_{1:T}))\boldsymbol{V}\boldsymbol{E}_0$. Choosing the readout layer appropriately, we can ensure that $T(x)_T = f^*(x_{1:T})$, as desired.

