# OpenReview forum: "Quantitative Bounds for Length Generalization in Transformers"
_ICLR.cc/2026/Conference — ICLR 2026 Oral_

### Official Review · Reviewer_dyQR · 2025-10-25

**Soundness:** 3
**Presentation:** 2
**Contribution:** 3
**Rating:** 6
**Confidence:** 2

**Summary:**

The paper provides quantitative bounds on the required training sequence length for transformer models to achieve length generalization by proving that this occurs when the model's behavior on longer sequences can be "simulated" by its behavior on the shorter sequences it was trained on.

**Strengths:**

1. The paper tackles an interesting and important aspect of length generalization, namely the minimum training length to enable it, given a task.
2. The paper takes care of the finite-precision arithmetic that is used in practice, something often overlooked in theoretical work.
3. Overall the paper is generally well-written and explained. Intuition and remarks are provided after most mathematical definitions and results which makes it easy to follow the reasoning and the motivation.

**Weaknesses:**

1. I am missing a comprehensive discussion on the limitations of the paper’s analysis, especially on all the technical assumptions and simplifications on both the class of tasks that is considered and on the architectural modifications.
2. The introduction of the $\phi_{l,h}(j,i)$ terms in the definition of the limit transformer appears to be a significant difference from standard attention. And contrary to what is said on line 133, it seems that they can only capture dependencies between the positions, not the content. At any rate, the paper would benefit from explaining why allowing for more expressive dependencies than the bilinear interaction of classic attention is negligible for the properties of the hypothesis class.
3. The definition of translation invariance is also a bit strange as nothing prevents positional information to also flow through the $\mathbf y_j \mathbf K^\top \mathbf Q \mathbf y_i$ term.  Perhaps it would be neater if you separate the input embeddings and the positional encodings as two separate components rather than adding them together.
4. The experiments and the corresponding plots can be explained more clearly and in more detail. The subplots are missing captions which could clarify what is being plotted and the first subplot is missing indication on the legend as to what the different lines are showing.

**Questions:**

Typos:
- Line 133: “denote allow”.
- Line 441: “value the test loss” -> “value of the test loss”

---

> ### Author Response · Authors · 2025-11-20
>
> Thank you to the reviewers for your detailed review, and for your positive evaluation of our paper. We respond to your questions below.
>
> > I am missing a comprehensive discussion on the limitations of the paper’s analysis, especially on all the technical assumptions and simplifications on both the class of tasks that is considered and on the architectural modifications.
>
> The main technical limitations of our paper are the depth of the transformer (one or two layers) and the assumption that the target function obeys the $\Delta$-periodicity and $\tau$-local assumptions, which limits the class of tasks considered. The latter limitation is in some sense necessary, as without some restriction on the target function, it is impossible to identify it from sequences of finite length. The extension to deeper transformers is an interesting direction for future work. We believe that the methods we used for the two layer case should be extendable to more layers, though the complexity of the required calculations may increase rapidly; in essence, there are just many more high-order dependencies in the token statistics that we need to keep track of. We hope that the ideas we have presented in their current form still make a valuable step forward towards theoretical understanding of LG and we believe that generalizing our results to larger models is an interesting direction of future work. We give a more in-depth discussion of architectural modifications below.
>
> > The introduction of the \phi_l,h(j,i)  terms in the definition of the limit transformer appears to be a significant difference from standard attention. And contrary to what is said on line 133, it seems that they can only capture dependencies between the positions, not the content. At any rate, the paper would benefit from explaining why allowing for more expressive dependencies than the bilinear interaction of classic attention is negligible for the properties of the hypothesis class.
>
> You are correct that the $\phi$ terms only capture positional information and not the content (i.e., specific tokens) at these positions. Our statement “The functions $\phi_{l,h}(j, i)$ allow for modifications to the attention pattern which cannot be captured by positional embedding vectors alone” (emphasis added) on line 133 just means that the $\phi$ terms allow us to include additional positional information. The information about the content will still come from the token embeddings.
>
> Regarding the expressivity of the setting in the paper vs. the purely bilinear setup, we actually view this as a non-negligible modification which increases the complexity of the problem. However, this should be viewed as a feature of our setting, rather than a drawback. First, the case without the additional positional information $\phi$ is a special case of our setting. This can be seen because $\phi \equiv 0$ satisfies all of the conditions we have placed on $\phi$.
>
> The motivation for introducing the $\phi$, which stems from Huang et al. (ICLR 2025), is the following. If we were to allow for arbitrary absolute positional embeddings, then LG would be impossible, as there would be no way to predict the transformer behavior on unseen positions. The $\phi$ therefore act as an abstraction of relative positional embeddings, where the transformer can still use positional information through $\phi$ to attend to other positions locally, yet in a translational invariant manner which keeps the complexity of the transformer bounded even as the input sequence length grows to infinity. Huang et al. show that a number of basic computational primitives, for example the induction head, can indeed be expressed by limit transformers.
>
> In summary, the $\phi$s create a more expressive function class which permits more nuanced behavior for the transformer at long sequence lengths.

---

> > ### Author Response · Authors · 2025-11-20
> > **Official Comment by Authors (cont.)**
> >
> > > The definition of translation invariance is also a bit strange as nothing prevents positional information to also flow through the y_j K^T Q y_i term. Perhaps it would be neater if you separate the input embeddings and the positional encodings as two separate components rather than adding them together.
> >
> > This is a good observation and it is true that the positional embedding can affect the key/query dot product. In the original framing of the limit transformer proposed by Huang et al. (ICLR 2025), they made stronger additional assumptions which essentially prevented this from happening, ensuring translation invariance both in the positional embeddings themselves and the dot products. Our analysis shows that this stronger assumption is not necessary to achieve LG: in spite of the fact that the positional information can impact the key/query dot products, we still obtain our result. The case where positional information is prevented from impacting the transformer in this way should be strictly simpler. We also remark that the setup we use in the paper is closer to what is used in practice, as the positional embeddings are added before computing attention logits. Thus, we believe this setup to be preferable.
> >
> > > The experiments and the corresponding plots can be explained more clearly and in more detail. The subplots are missing captions which could clarify what is being plotted and the first subplot is missing indication on the legend as to what the different lines are showing.
> >
> > Thank you for drawing our attention to this. In the left panel of Fig. 1, the number in the legend corresponds to the training sequence length. This is indicated in the caption for the figure but not in the legend itself; we have updated the figure to make this clearer. The left panel shows that the plateau of the test loss decreases as the training length increases, which is expected.

---

> > > ### Comment · Reviewer_dyQR · 2025-11-22
> > >
> > > Thank you for your detailed response.
> > >
> > > I am still a bit confused about the introduction of the $\phi$ terms. On one hand you say that they are not necessary as can be all set to 0. But then you say that these terms are necessary if one wants to capture relative positional information when the input sequence length grows to infinity. Therefore, it is not clear to me whether your results are only valid for models that do incorporate these additional terms or not.

---

> > > > ### Author Response · Authors · 2025-11-25
> > > >
> > > > Our apologies for the confusion. To be clear, our results apply whether or not the $\phi$ terms are included. This is because not including the $\phi$ terms is equivalent to setting $\phi_{l, h}(j,i)=0$ for all $i,j,l,h$, and this definition of $\phi$ satisfies the conditions of our theorems. Thus, our results apply whether or not the $\phi$ are included.
> > > >
> > > > The second point that the $\phi$ are necessary to include relative information for long sequences is a statement about the model of the problem itself, not our results. Essentially, what we are trying to say is that the problem itself is less interesting without the inclusion of the $\phi$ terms, because local information is washed out for long input sequences without these terms. So while the $\phi$ terms are a deviation from the standard transformer attention setup, they actually make for behavior which is more qualitatively similar to that of real transformers on long but not “nearly infinite” length sequences.
> > > >
> > > > We hope this has clarified the situation but we are happy to discuss further if it has not. The tl;dr is that all of our results are applicable whether or not the $\phi$s are included, but we think the problem itself is more interesting when they are included (and this view seems to be held by others in the theory community as well).

---

> > > > > ### Comment · Reviewer_dyQR · 2025-11-25
> > > > >
> > > > > Thank you for the clarification. I understand, I suppose that this is consistent with prior findings that transformers cannot naively scale to sequences of arbitrary length due to attention not being able to be sufficiently picky. And if one was to study LG for arbitrary lengths, they'd need to work around this. I appreciate your response and will increase my score accordingly.

---

### Official Review · Reviewer_77Hi · 2025-10-26

**Soundness:** 3
**Presentation:** 4
**Contribution:** 3
**Rating:** 8
**Confidence:** 4

**Summary:**

The authors provide the first non-asymptotic quantitative theory of length generalization (LG) in transformers, establishing explicit bounds on the minimum training sequence length $N$ required for a model trained on short sequences to generalize to arbitrarily long ones. In the finite-precision (hardmax) regime, Theorem 4.1 shows that for one-layer $\tau$-local, $\Delta$-periodic limit transformers, if two models agree on inputs up to length $N = O(\max{2^{p/\gamma}, L^{2}\Delta^{7}|\Sigma|^{6}\tau^{2}\varepsilon^{-2}})$, then they agree within $O(\varepsilon)$ on all longer sequences, with $L$ being a Lipshitz style upper bound, $\gamma$ the attention logit margin, and $|\Sigma|$ the vocabulary size. Theorem 4.2 extends this to average-case error control, showing that under regular input distributions, $|f-g|{T,P} = O(\varepsilon^{1/2})$ for all $T \ge N$, with $N$ scaling polynomially in $L, \Delta, \tau, |\Sigma|$. In the infinite-precision (softmax) setting, Theorem 5.2 introduces a complexity measure $C(f)$ containing exp and poly factors in the norms, and and provides the equivalent characterization of the length $N$ for LG in two-layer transformers, implying exponential dependence on the inverse positional margin, which although will be huge in general. All proofs rely on a unified simulation argument constructing a short sequence $z$ that preserves the sufficient statistics of a long sequence $x$ so that $f(x)\approx f(z)$. Empirical studies on synthetic tasks (SimpleTask, ModPTask, and in-context $k$-gram) validate these scalings: test loss plateaus at finite lengths, decreasing with training length. Collectively, the results establish that the capacity for LG in terms of the model parameters and the usual error tolerance, providing a rigorous quantitative framework linking transformer parameters to their extrapolation behavior.

Note the questions and concerns I have raised below.

**Strengths:**

1. The Dirichlet assumption in Theorem 4.2 is a good choice, as classical NLP literature has worked with it for decades in many practical applications. This makes the contribution of this work more practical

2. The authors' efforts in studying both finite and infinite precision and also providing bounds for L_{inf} and average case for finite setting is a good contribution to this line of research.

3. I found Lemma 5.3 to be a useful addition. While I did not read the proof end to end, this is a good supporting lemma of independent interest - assuming that it does not exist in the literature in a similar form yet.

**Weaknesses:**

1. I have a concern about the value of N - $2^{\frac{p}{\min\{\gamma(f), \gamma(g)\}}}$ . While I understand the requirements for the proofs, I am curious if the authors have some thoughts on adapting equation 2 on page 13 – for say cases where the logit differences can be binned and perhaps some covering number arguments can be used? Please note this is just a rough guess and I have not put proper thought behind this statement, and I am just curious about the ideas from the authors’ end.

2. In line 1024 of page 19, while the triviality of - |Σ|^3 + τ ≤ |Σ| ^ X τ , why was it even necessary to loosen the bound by such an extent. Furthermore, for a good number of practical datasets where |Σ| = O(τ), this unnecessarily loosens the bound. Again, curious about this situation.

**Questions:**

1. What exact upper bound for the Lipshitz constant are the authors using to support the argument - “This is unavoidable, as the Lipschitz constant of the first layer softmax scales exponentially” - line 366 on page 7. To my understanding, there have been various proposals for the upper bound such as [1-3], some of which not scaling exponentially.

2. Can the authors clarify the argument - “One observes that f ∗ is expressible by a one-layer limit transformer with no positional embeddings and L = Θ(ω).” from line 418? More specifically, how was this bound achieved for L and similarly in line 424 for the ModPTask?

3. The left subfigure in Figure 1 on page 8 is interesting – the smaller values of ω exhibit larger test loss across all lengths. I am curious to understand the authors’ discussion on this, because either using the bounds from this paper, or using standard generalization bounds, for smaller ω, a smaller test loss across all lengths is expected. Looking forward to this response in case I made a mistake in my understanding.



[1] Kim, H., Papamakarios, G., & Mnih, A. (2021). The Lipschitz Constant of Self-Attention. In Proc. ICML 2021, PMLR vol. 139

[2] Castin, V., Ablin, P., & Peyré, G. (2024). How Smooth Is Attention? (ICML 2024). Proceedings of Machine Learning Research vol. 235

[3] Wang, Y., Chauhan, J., Wang, W., & Hsieh, C.-J. (2023). Universality and Limitations of Prompt Tuning. In Advances in Neural Information Processing Systems 36 (NeurIPS 2023).

---

> ### Author Response · Authors · 2025-11-20
>
> Thank you to the reviewers for your detailed review, and for your positive evaluation of our paper. We respond to your questions below.
>
> > I have a concern about the value of N - 2^{p/min(\gamma(f), \gamma(g)}. While I understand the requirements for the proofs, I am curious if the authors have some thoughts on adapting equation 2 on page 13 – for say cases where the logit differences can be binned and perhaps some covering number arguments can be used? Please note this is just a rough guess and I have not put proper thought behind this statement, and I am just curious about the ideas from the authors’ end.
>
> If we have understood your suggestion correctly, an approach similar to the one you mentioned may have been considered by the Huang et al. (ICLR 2025) paper which introduced the limit transformer. The idea is that because the computation in equation (2) can be binned according to the finite precision of the transformer, there are essentially only a finite number of limit transformers which can be expressed given bounds on the problem-specific constants. Thus, there must be some finite training length N where if two limit transformers agree up to this point, then they will agree for all sequence lengths; this is because disagreement on some sequence means that the two limit transformers are not equal. We found that we could not easily modify this approach to give quantitative bounds on the length N, however, hence our alternate proof. We also hope that the constructive approach we have given may shed additional insight on how LG occurs. We agree that it would be interesting to see if a covering number approach could be used to derive a quantitative bound from the general method of Huang et al.
>
> > In line 1024 of page 19, while the triviality of - |Σ|^3 + τ ≤ |Σ| ^ X τ , why was it even necessary to loosen the bound by such an extent. Furthermore, for a good number of practical datasets where |Σ| = O(τ), this unnecessarily loosens the bound. Again, curious about this situation.
>
> You are right that this does loosen the bound when $|\Sigma|$ and $\tau$ are of similar size. It is not necessary for the analysis and all of the computations can be performed with the original $|\Sigma|^3 + \tau$ in place of $|\Sigma|^3 \tau$. We made this change primarily for aesthetic and pedagogical reasons. Since in the end we are mostly concerned with the qualitative behavior of the bound, rather than the exact polynomial powers for the problem-specific constants, we thought that presenting the bound in this form made it easier for a reader to parse with only very minor changes to the qualitative interpretation.
>
> > What exact upper bound for the Lipshitz constant are the authors using to support the argument - “This is unavoidable, as the Lipschitz constant of the first layer softmax scales exponentially” - line 366 on page 7. To my understanding, there have been various proposals for the upper bound such as [1-3], some of which not scaling exponentially.
>
> In the proof of Theorem 5.2, we treat self-attention as a mapping on the space of probability measures; in particular, we interpret self-attention as a function which takes in a probability distribution over [S] and outputs a vector in R^d. This allows us to have a unified bound independent of the sequence length T, which is necessary since we require LG for sequences of arbitrary length.
>
> The bounds in the papers you’ve referenced which do not scale exponentially, actually depend on the length of the input sequence. For example, [2, Theorem 3.3] has the Lipschitz constant growing with $\sqrt{n}$, where n is the sequence length. [2] also provides bounds independent of the sequence length in a “mean-field” regime, which also interprets self-attention as a function of a probability distribution and thus more closely resembles the setting in our paper. In this mean-field regime, [2, Theorem 3.5] provides a bound on the Lipschitz constant which does actually scale exponentially (and in Proposition 3.6 argues that this is tight).
>
> We will be sure to emphasize the subtlety of this point and add the references you’ve suggested.
>
> > Can the authors clarify the argument - “One observes that f ∗ is expressible by a one-layer limit transformer with no positional embeddings and L = Θ(ω).” from line 418? More specifically, how was this bound achieved for L and similarly in line 424 for the ModPTask?
>
> The sketch of the constructions for both SimpleTask and ModPTask are provided in Appendix C.1. The bound for L comes from the classical result in approximation theory that any one-dimensional $\omega$-Lipschitz function can be approximated by a two-layer neural network with weight norm $\Theta(\omega)$.

---

> > ### Author Response · Authors · 2025-11-20
> > **Official Comment by Authors (cont.)**
> >
> > > The left subfigure in Figure 1 on page 8 is interesting – the smaller values of ω exhibit larger test loss across all lengths. I am curious to understand the authors’ discussion on this, because either using the bounds from this paper, or using standard generalization bounds, for smaller ω, a smaller test loss across all lengths is expected. Looking forward to this response in case I made a mistake in my understanding.
> >
> > Thank you for drawing our attention to this, as we realize the figure is somewhat unclear. In the left panel of Fig. 1, the number in the legend corresponds to the training sequence length, rather than the weight $\omega$. This is indicated in the caption for the figure but not in the legend itself; we have updated the figure to make this clearer. The left panel shows that the plateau of the test loss decreases as the training length increases, which is expected. The right plot shows that the test loss plateaus at a value which generally increases with $\omega$.

---

> > > ### Comment · Reviewer_77Hi · 2025-11-21
> > > **Response to the authors**
> > >
> > > Thanks a lot for the responses. I often don't see such clear and concise clarifications, so this is great!
> > > I think given the paper's direction and current state; it will be good to see this work highlighted as a spotlight/oral in the conference. Thus, I am increasing the score to 10 and am willing to defend my score should the AC initiate a discussion.
> > > I hope the authors have a satisfactory discussion with the remaining reviewers.

---

### Official Review · Reviewer_96HU · 2025-11-01

**Soundness:** 3
**Presentation:** 3
**Contribution:** 3
**Rating:** 6
**Confidence:** 2

**Summary:**

The paper builds upon the theoretical constructions (Limit Transformer) and the conclusion in  Huang et al. (2024)  - that Limit Transformers can achieve Length Generalization if trained under sufficient length for at least (if not at most) class of problems that can be expressed by a C-RASP program. This paper attempts to go beyond that conclusion and also provide quantitative bounds for the required lengthts under different settings - such as infinite-precision vs finite-precision attention or different error controls, single-layer and 2-layer transformers.

The key level idea used for the proof is to construct a simulation map from strings of arbitrary length to strings of bounded length.

**Strengths:**

* Generally well written and well connected with the literatur.
* Provides theoretical proof to support quantitative bounds for required training length
* Provides some empirical support that error rate at which test loss plateus decreases with increasing training length.

**Weaknesses:**

Not any significant blocker to acceptance to my awareness. One could question the practical impact or limited scope, but it is targeted as a theoretical paper under learning theory - and seems to sufficiently fulfill its targeted goal. There are precedents of papers with similar scope getting accepted.

Besides this, one limitation is that the study is mainly tied to (periodic) absolute positional encodings (as far as I understand) - but the authors keep studies of other positional schemes for future work - which is fair enough.

**Questions:**

n/a

---

> ### Author Response · Authors · 2025-11-20
>
> Thank you to the reviewers for your detailed review, and for your positive evaluation of our paper. We respond to your questions below.
>
> >Besides this, one limitation is that the study is mainly tied to (periodic) absolute positional encodings (as far as I understand) - but the authors keep studies of other positional schemes for future work - which is fair enough.
>
> It is true that the positional embedding vectors we use are absolute and extending the analysis to include relative positional embedding vectors is an important direction for future work. We remark, however, that the translation invariance of the positional embedding information $\phi$ effectively makes it relative. This is because, for any fixed relative distance $k$ and any two indices $i, j$, we have:
> $$ \phi(i, i+k) = \phi(i + (j-i), i + k + (j-i)) = \phi(j, j+k). $$
> Thus, our results have some partial coverage of the relative positional encoding case and we expect similar techniques will be applicable.

---

> > ### Comment · Reviewer_96HU · 2025-11-22
> >
> > Yes, that makes sense.
> > I increased my overall score to 8, based on the overall reviews and rebuttals.

---

### Official Review · Reviewer_F5Ex · 2025-11-03

**Soundness:** 3
**Presentation:** 3
**Contribution:** 2
**Rating:** 8
**Confidence:** 2

**Summary:**

This paper focuses on the problem of length generalization (LG) of Transformers - i.e., the ability of a model to maintain its performance on unseen long sequences after being trained on short sequences - and for the first time gives quantitative bounds on the length of the training sequences required to achieve LG. The study covers a variety of scenarios, with the core logic that LG can be achieved when the internal behavior of long sequences can be “simulated” by short sequences in the training set. The paper clarifies the correlation between training length and model parameter paradigms, period Δ, locality T, etc. through theoretical derivation, and verifies the conclusions through synthetic tasks such as SimpleTask, ModPTask, etc., which provide theoretical references for scaling the length of training contexts of Large Language Models (LLMs).

**Strengths:**

- The length of the training sequence required for LG is quantified for the first time, addressing the key issue that previous studies have only demonstrated the “existence of a threshold” but have not clarified the “size of the threshold”.
- The conclusions are more broadly applicable by also considering a variety of variables such as the type of accuracy, the number of model layers, and the way the error is controlled. Covers many different types of theoretical scenarios and explores the quantitative issues regarding generalizability in a comprehensive and detailed manner.
- The boundaries are established by rigorous mathematical derivation and then verified by a synthetic task with high confidence in the conclusions. The theory is paid to the experiment, and the correctness of the theory is effectively verified through the results.

**Weaknesses:**

- Very interesting paper that gives a quantitative analysis of length generalizability, based on the theoretical framework of the “limit transformer”, and therefore lacks an analysis of the transformer scheme with relative positional coding, which is currently more used in various methods.
- The paper verifies the correctness of its theory on a small-scale transformer structure, and it is hoped that the paper will give further analysis on whether there are limitations in the analysis and experimentation of larger models.
- Are the complex model parameters on which quantitative analysis relies also greater in more complex models? Does this create bottlenecks in analysis for larger models? Hopefully the paper will give at least a qualitative judgment.
- Although I have the above doubts, I nevertheless think that the work in this paper is solid, informative, and importantly groundbreaking for the study of length generalizability, and I therefore hope that the authors will be able to give at least a qualitative judgment on the above doubts.

**Questions:**

see Weakness

---

> ### Author Response · Authors · 2025-11-20
>
> Thank you to the reviewers for your detailed review, and for your positive evaluation of our paper. We respond to your questions below.
>
> > Very interesting paper that gives a quantitative analysis of length generalizability, based on the theoretical framework of the “limit transformer”, and therefore lacks an analysis of the transformer scheme with relative positional coding, which is currently more used in various methods.
>
> It is true that the positional embedding vectors we use are absolute and extending the analysis to include relative positional embedding vectors is an important direction for future work. We remark, however, that the translation invariance of the positional embedding information $\phi$ effectively makes it relative. This is because, for any fixed relative distance $k$ and any two indices $i, j$, we have:
> $$ \phi(i, i+k) = \phi(i + (j-i), i + k + (j-i)) = \phi(j, j+k). $$
> Thus, our results have some partial coverage of the relative positional encoding case and we expect similar techniques will be applicable.
>
> > The paper verifies the correctness of its theory on a small-scale transformer structure, and it is hoped that the paper will give further analysis on whether there are limitations in the analysis and experimentation of larger models.
>
> The main limitation in the current theoretical analysis for larger models is the restriction to at most two transformer layers. We believe that the methods we used for the two layer case should be extendable to more layers, though the complexity of the required calculations may increase rapidly; in essence, there are just many more high-order dependencies in the token statistics that we need to keep track of. We hope that the ideas we have presented in their current form still make a valuable step forward towards theoretical understanding of LG and we think that generalizing our results to larger models is an interesting direction of future work.
>
> > Are the complex model parameters on which quantitative analysis relies also greater in more complex models? Does this create bottlenecks in analysis for larger models? Hopefully the paper will give at least a qualitative judgment.
>
> Strictly speaking, none of the parameters (vocabulary size $|\Sigma|$, periodicity $\Delta$, locality parameter $\tau$, weight norm bound $L$, minimum logit gap $1/\gamma$) necessarily have to increase for larger models, though it is possible that the precise dependence of the bounds on these parameters may increase as e.g. the depth of the model increases. In practice, more complex target functions (such as predicting the next token in code generation or natural language) might be expected to have larger values for some of these parameters. In general, larger values for these problem-specific constants don’t change the analysis provided that they do not also grow with the sequence length.

---

> > ### Comment · Reviewer_F5Ex · 2025-11-28
> >
> > This information has explained my doubts and I am appreciative of the work you all have done and I will keep my scoring the same.

---

### Meta-Review · Area_Chair_5YBH · 2026-01-13

**Summary:**

From the reviews, the main concerns among reviewers include: (i) the scope of the modeling assumptions relative to modern transformer practice, (ii) the restrictions of the theoretical setting (task class and depth), and (iii) the magnitude/tightness and interpretability of the resulting bounds, as well as some presentation clarity in the experiments.

- Positional schemes and modeling alignment: reviewers noted the analysis is “mainly tied to (periodic) absolute positional encodings” (Reviewer 96HU, Weaknesses) and “lacks an analysis … with relative positional coding” (Reviewer F5Ex, Weaknesses). Relatedly, Reviewer dyQR questioned the modeling choices, including that the “definition of translation invariance is also a bit strange” (Reviewer dyQR, Weaknesses).

- Task-class and architectural limitations: reviewers highlighted that the theory/experiments operate in restricted regimes, including “small-scale transformer structure” (Reviewer F5Ex, Weaknesses) and reliance on “technical assumptions and simplifications on … the class of tasks” (Reviewer dyQR, Weaknesses). Depth limitations were also emphasized in discussion: “restriction to at most two transformer layers” (Authors’ response to Reviewer F5Ex).

- Tightness and size of quantitative bounds: Reviewer 77Hi raised concerns about potentially very large terms in the required training length, explicitly flagging “a concern about the value of N … 2^{p/min(γ(f),γ(g))}” (Reviewer 77Hi, Weaknesses) and asked for clarification around the statement that exponential scaling in softmax Lipschitz behavior is “unavoidable” (Reviewer 77Hi, Questions). Reviewer 96HU also noted one could “question the practical impact or limited scope” (Reviewer 96HU, Weaknesses).

- Experimental/figure clarity: reviewers requested clearer plotting/annotation, e.g., “subplots are missing captions … [and] the … subplot is missing indication on the legend” (Reviewer dyQR, Weaknesses), and Reviewer 77Hi noted confusion about Fig. 1 interpretation (Reviewer 77Hi, Questions).

**Reviewer Concerns:**

Reviewer concerns addressed by the rebuttal / discussion:

- **Relative vs. absolute positional information**: Authors acknowledged the limitation (“extending the analysis … is an important direction for future work”) but argued partial coverage via translation invariance, stating it “effectively makes it relative” and giving the identity ϕ(j,j+k)=ϕ(i,i+k) (Authors’ responses to Reviewers F5Ex and 96HU: “translation invariance … effectively makes it relative.”).

- **Confusion about the role/necessity of ϕ-terms**: Reviewer dyQR explicitly asked whether results “are only valid for models that do incorporate these additional terms” (Reviewer dyQR, comment 22 Nov). Authors clarified unequivocally: “our results apply whether or not the ϕ terms are included,” since omitting them is “equivalent to setting ϕ…=0” (Authors’ reply 25 Nov). Reviewer dyQR confirmed: “Thank you for the clarification. I understand … and will increase my score accordingly.” (Reviewer dyQR, comment 26 Nov).

- **Softmax Lipschitz exponential scaling question**: Reviewer 77Hi asked what bound supports “Lipschitz constant … scales exponentially” (Reviewer 77Hi, Questions). Authors responded that they need a bound “independent of the sequence length T,” and that non-exponential bounds in cited works “depend on the length of the input sequence,” while mean-field bounds “do actually scale exponentially … and … this is tight” (Authors’ response to Reviewer 77Hi, citing [2, Theorem 3.5] and Proposition 3.6; and: “We will … add the references”).

- **Over-loosening of a polynomial bound**: Reviewer 77Hi questioned loosening |Σ|^3+τ to a more conservative form (Reviewer 77Hi, Weaknesses). Authors agreed it “does loosen the bound” and stated it was “not necessary for the analysis,” done “primarily for aesthetic and pedagogical reasons” (Authors’ response to Reviewer 77Hi).

- **Figure/legend clarity**: Reviewers asked for clearer legends/captions (Reviewer dyQR, Weaknesses; Reviewer 77Hi, Questions). Authors clarified that in Fig. 1 “the number in the legend corresponds to the training sequence length, rather than the weight ω,” and noted they “updated the figure” (Authors’ response to Reviewer 77Hi; similarly in response to Reviewer dyQR).

Still outstanding / only partially addressed:

- **Full treatment of modern positional schemes beyond the current abstraction**: while authors argue “partial coverage” via translation invariance, they still frame extending to relative positional embeddings as “an important direction for future work” (Authors’ responses to Reviewers F5Ex and 96HU). A complete analysis for standard relative-position mechanisms remains open.

- **Extension beyond 1–2 layers and implications for larger models**: reviewers asked about “limitations … of larger models” (Reviewer F5Ex, Weaknesses) and broader limitations discussion (Reviewer dyQR, Weaknesses). Authors identify the key restriction as “at most two transformer layers” and state extension is plausible but “complexity … may increase rapidly” due to “many more high-order dependencies” (Authors’ responses to Reviewers F5Ex and dyQR). This is a reasonable qualitative explanation, but the generalization to deeper architectures is not yet established.

- **Practical tightness / magnitude of bounds for realistic tasks**: Reviewer 96HU noted one could “question the practical impact or limited scope” (Reviewer 96HU, Weaknesses), and Reviewer 77Hi asked about reducing the conservativeness of N (e.g., “covering number arguments”) (Reviewer 77Hi, Weaknesses). Authors explain why they did not derive quantitative bounds from a covering-number style argument and leave it as an interesting direction (“could not easily modify … to give quantitative bounds,” “interesting to see if a covering number approach could be used”) (Authors’ response to Reviewer 77Hi). The discussion clarifies intent but does not materially tighten bounds or validate them on non-synthetic tasks.

**Reviewer Scores:**

- **Reviewer F5Ex**: Likely unchanged. The reviewer explicitly stated the rebuttal “explained my doubts” and “I will keep my scoring the same” (Reviewer F5Ex, comment 28 Nov).
  - **Expected score**: 8 (as in the review).

- **Reviewer 96HU**: The reviewer explicitly increased their score after rebuttal/discussion: “I increased my overall score to 8” (Reviewer 96HU, comment 22 Nov).
  - **Expected score**: 8 (up from the originally posted 6 in the review).

- **Reviewer 77Hi**: The reviewer explicitly increased their score after the authors’ clarifications: “I am increasing the score to 10 and am willing to defend my score” (Reviewer 77Hi, comment 21 Nov).
  - **Expected score**: 10 (up from the originally posted 8 in the review).

- **Reviewer dyQR**: The reviewer indicated they would raise their score after the clarification on ϕ-terms: “I appreciate your response and will increase my score accordingly” (Reviewer dyQR, comment 26 Nov). Since no updated numeric score is stated in the discussion, my best estimate is a modest increase from 6 to 8 (i.e., moving from marginally above threshold to a clearer accept), contingent on the reviewer’s typical calibration.
  - **Expected score**: 6 or 8.

---

### Decision · Program_Chairs · 2026-01-26

Accept (Oral)